# Tighter CMI-Based Generalization Bounds via Stochastic Projection and Quantization

**Milad Sefidgaran** [1], **Kimia Nadjahi** [2], **Abdellatif Zaidi** [1,3]

[1] Paris Research Center, Huawei Technologies France
[2] CNRS, ENS Paris, France [3] Université Gustave Eiffel, France
milad.sefidgaran2@huawei.com, kimia.nadjahi@ens.fr, abdellatif.zaidi@univ-eiffel.fr

## Abstract

In this paper, we leverage stochastic projection and lossy compression to establish new conditional mutual information (CMI) bounds on the generalization error of statistical learning algorithms. It is shown that these bounds are generally tighter than the existing ones. In particular, we prove that for certain problem instances for which existing MI and CMI bounds were recently shown in Attias et al. [2024] and Livni [2023] to become vacuous or fail to describe the right generalization behavior, our bounds yield suitable generalization guarantees of the order of $\mathcal{O}(1/\sqrt{n})$, where $n$ is the size of the training dataset. Furthermore, we use our bounds to investigate the problem of data "memorization" raised in those works, and which asserts that there are learning problem instances for which any learning algorithm that has good prediction there exist distributions under which the algorithm must "memorize" a big fraction of the training dataset. We show that for every learning algorithm, there exists an auxiliary algorithm that does *not* memorize and which yields comparable generalization error for any data distribution. In part, this shows that memorization is not necessary for good generalization.

## 1 Introduction

One of the major problems in statistical learning theory consists in understanding what really drives the generalization error of learning algorithms. That is, what makes an algorithm trained on a given dataset continue to perform well on unseen data samples. Historically, this fundamental question has been studied independently in various lines of work, using seemingly unconnected tools. This includes VC-dimension theory [1], Rademacher complexity approaches [2], stability-based analysis [3] and, more recently, intrinsic-dimension [4–8] and information-theoretic approaches [9–21]. It is only until recently that the above various approaches were shown to be possibly unified [22, 23] using a *variable-length* compressibility technique, which is rate-distortion-theoretic in nature.

In the context of statistical learning theory perhaps one can date back information-theoretic approaches to the PAC-Bayes bounds of McAllester [24, 25], which were then followed by various extensions and ramifications [26–39]. The mutual information (MI) bounds of [9] and [10] have the advantages to be relatively simpler comparatively and of offering somewhat clearer insights into the question of generalization. Roughly, such bounds suggest that a learning algorithm generalizes better as its output model reveals less information about the training data samples, where the amount of revealed information is measured in terms of the Shannon mutual information.

However, MI-based bounds are also known to sometimes take large (infinite) values and become vacuous, such as for continuous data and deterministic models. This shortcoming has been identified in a number of works, including [40, 41]. The issue was believed to be resolved by the introduction in [12] of the important framework of conditional mutual information (CMI). The CMI setting introduces a "super-sample" construction in which an auxiliary "ghost sample" is used in conjunction with the training sample; and a sequence of Bernoulli random variables determines which data samples among the super-sample were

used for the training. It is shown that a bound on the generalization error involves the mutual information between the Bernoulli random variables and the hypothesis (e.g., model parameters), conditionally given the super-sample [12, Theorem 2]. Because the entropy of Bernoulli random variables is bounded, the resulting bound is bounded. Many follow-up works have proposed extensions and improvements of the original CMI bounds, including using *randomized subset* and *individual sample* techniques, disintegration, and fast-rate variations in regimes in which the empirical risk is small – See [42] for more on this.

CMI-type bounds were largely believed to be exempt from the aforementioned limitations of MI bounds until it was recently reported that examples can be constructed for which the standard[1] CMI-based bound and its individual-sample variant fail [14, 43, 46]. The (counter-) examples of [46] are in the context of Stochastic Convex Optimization (SCO) problems; and those of [43] involve carefully constructed Convex-Lipschitz-bounded (CLB) and Convex-set-Strongly convex-Lipschitz (CSL) instance problems. These limitations were sometimes extrapolated to the extent of even questioning the utility of information-theoretic bounds for the analysis of the generalization error of statistical learning algorithms more generally [47]. In this context, we mention [23, Appendix A] in which it was shown that, when applied to the counter-example of [47], a lossy version of MI bounds yields generalization bounds that are of order $\mathcal{O}(1/n)$, instead of $\Omega(1)$ in the case of standard (lossless) MI bounds.[2] The idea of lossy compression was also used in [49].

In this paper, essentially, we show that the aforementioned limitations are in fact *not* inherent to the CMI framework; and, actually, the CMI framework can be adjusted slightly by the incorporation of a suitable stochastic projection and a suitable lossy compression to cope with those issues. Also, leveraging the utility of CMI and membership inference to study the problem of memorization and its relationship to generalization in machine learning, we use our results to revisit the necessity of memorization for SCO problems claimed in [43]. We show that memorization is *not* necessary for good generalization; and, as such, the result contributes to a better understanding of what role memorization plays in machine learning, a problem which is yet to be fully understood. Specifically, our contributions are as follows.

- We introduce stochastic projection in conjunction with lossy compression in the CMI framework, and we use them to establish a new CMI-based bound that is generally tighter than the CMI bounds of [12].

- We show that, in sharp contrast with classic CMI-based bounds which fail when applied to the aforementioned CLB, CSL and SCO problem instances of [43, 46] and may not even decay with the number of training samples, our new CMI bound yields meaningful results and decays with the number of training samples as $\mathcal{O}(1/\sqrt{n})$.

- By applying them to generalized linear stochastic (non-convex) optimization problems, in the appendices we demonstrate that our bounds remain non-vacuous even beyond the convex case previously studied in [50]. The generalization is shown to come at the expense of a slower decay with $n$ in our case; namely, $\mathcal{O}(1/\sqrt[4]{n})$ instead of $\mathcal{O}(1/\sqrt{n})$ if the functions are convex as in [50].

- We leverage the key ingredients of stochastic projection and lossy compression in the framework of CMI to study the "memorization" issue identified and studied in [43]. Specifically, [43] has demonstrated that, for a given problem instance and every $\varepsilon$-learner algorithm, there exists a data distribution under which the algorithm "memorizes" the training samples. We show that for any learning algorithm $\mathcal{A}$ that memorizes the training data, one can find (via stochastic projection and lossy compression) an alternate learning algorithm $\bar{\mathcal{A}}$ with comparable generalization error and that does *not* memorize the training data for any data distribution. In part, this means that memorization is *not* necessary for good generalization in SCO.

- In the appendices, we use our general bound to study the generalization error of subspace training algorithms. Specifically, we investigate the setting in which the training is performed using SGD or SGLD; and we derive new bounds based on the differential entropy of Gaussian mixture distributions. This entropy depends on the gradient difference for the training and test datasets, the noise power, the learning rate, and the uncertainty of the index of the training dataset within the super-dataset.

---

[1]The authors of [43] do not evaluate the performance of variants of CMI such as chained CMI [44], evaluated CMI and $f$-CMI [20, 21, 45] on their counter-example.

[2]The counterexample of [47] has also been addressed by Wang and Mao [48] using a different technique called "Sample-Conditioned Hypothesis Stability".

## 2 Notation and Background

Let $Z$ be some random variable with unknown distribution $\mu$ and taking values in some alphabet $\mathcal{Z}$. Let $S_n \triangleq (Z_1, \dots, Z_n) \in \mathcal{Z}^n$ be a set of $n$ data samples drawn uniformly from the distribution $\mu$, *i.e.,* $S_n \sim P_{S_n} = \mu^{\otimes n}$. In the framework of statistical learning, a (possibly) stochastic learning algorithm $\mathcal{A}: \mathcal{Z}^n \to \mathcal{W}$ takes the training dataset $S_n$ as input and returns a hypothesis $W \in \mathcal{W} \subseteq \mathbb{R}^D$. We assume that $\mathcal{A}$ is *randomized*, in the sense that its output $W \triangleq \mathcal{A}(S_n)$ is a random variable distributed according to $P_{W|S_n}$. We denote the distribution induced on $(S_n, W)$ as $P_{S_n, W} = P_{W|S_n} \otimes P_{S_n} = P_{W|S_n} \otimes \mu^{\otimes n}$.

For a given function $\ell: \mathcal{Z} \times \mathcal{W} \to \mathbb{R}$, the loss incurred by using a hypothesis $w \in \mathcal{W}$ for a sample $z$ is evaluated as $\ell(z, w)$. A statistical learning algorithm seeks to find a hypothesis $w$ whose *population risk* $\mathcal{R}(w) \triangleq \mathbb{E}_{Z \sim \mu}[\ell(Z, w)]$ is minimal. However, since the data distribution $\mu$ is unknown, direct computation of the population risk $\mathcal{R}(w)$ is not possible. Instead, one resorts to minimizing the *empirical risk* $\widehat{\mathcal{R}}(s_n, w) \triangleq \frac{1}{n} \sum_{i=1}^{n} \ell(z_i, w)$ or a regularized version of it. Throughout, if $s_n$ is known from the context, we will use the shorthand notation $\widehat{\mathcal{R}}_n(w) \equiv \widehat{\mathcal{R}}(s_n, w)$.

The *generalization error* induced by a specific choice of hypothesis $w \in \mathcal{W}$ and dataset $s_n$ is evaluated as

$$\mathrm{gen}(s_n, w) \triangleq \mathcal{R}(w) - \widehat{\mathcal{R}}_n(w);$$

and the expected *generalization error* of the learning algorithm $\mathcal{A}$ is obtained by taking the expectation over all possible choices of $(s_n, w)$, as

$$\mathrm{gen}(\mu, \mathcal{A}) \triangleq \mathbb{E}_{P_{S_n, W}}[\mathrm{gen}(S_n, W)] = \mathbb{E}_{P_{S_n, W}}[\mathcal{R}(W) - \widehat{\mathcal{R}}_n(W)].$$

### 2.1 Conditional Mutual Information Framework

Let $\tilde{\mathbf{S}} \in \mathcal{Z}^{n \times 2}$ be a super-sample composed of $2n$ data points $Z_{i,j}$ that are drawn uniformly from the distribution $\mu$, where $j \in \{0, 1\}$ and $i \in [n]$. Also, let $\mathbf{J} = (J_1, \dots, J_n) \in \{0, 1\}^n$ be a vector of $n$ independent Bernoulli$(1/2)$ random variables, all drawn independently from $\tilde{\mathbf{S}}$. Let $\tilde{\mathbf{S}}_{\mathbf{J}} = \{Z_{1, J_1}, Z_{2, J_2}, \dots, Z_{n, J_n}\}$. In what follows, $\tilde{\mathbf{S}}_{\mathbf{J}}$ plays the role of the training dataset $S_n$, $\tilde{\mathbf{S}} \setminus \tilde{\mathbf{S}}_{\mathbf{J}}$ plays the role of a test or "ghost" dataset $S_n'$ and $\tilde{\mathbf{S}}$ is a shuffled version of the union of the two. For an algorithm $\mathcal{A}: \mathcal{Z}^n \to \mathcal{W}$, its CMI with respect to the data distribution $\mu$ is defined as

$$\mathrm{CMI}(\mu, \mathcal{A}) \triangleq \mathrm{I}(\mathcal{A}(\tilde{\mathbf{S}}_{\mathbf{J}}); \mathbf{J} | \tilde{\mathbf{S}}).$$

The CMI captures the information that the output hypothesis of the algorithm $\mathcal{A}$ trained on $\tilde{\mathbf{S}}_{\mathbf{J}}$ provides about the membership vector $\mathbf{J}$ given the super-sample $\tilde{\mathbf{S}}$. Equivalently, the CMI measures the extent to which the training and test datasets are distinguishable given the shuffled version of the union of the two, as well as the trained model. In its simplest form, it is shown in [12] that the generalization error of an algorithm for a bounded loss in the range $[0, 1]$ can be upper-bounded as

$$\mathrm{gen}(\mu, \mathcal{A}) \leq \sqrt{\frac{2}{n} \mathrm{CMI}(\mu, \mathcal{A})}.$$

Furthermore, for a Convex-Lipschitz-Bounded (CLB) whose formal definition will follow, the generalization error of $\mathcal{A}$ was shown in [47] to be upper-bounded as

$$\mathrm{gen}(\mu, \mathcal{A}) \leq LR \sqrt{\frac{8}{n} \mathrm{CMI}(\mu, \mathcal{A})}. \tag{1}$$

**Definition 1** (SCO Problem). *A stochastic convex optimization (SCO) problem is a triple $(\mathcal{W}, \mathcal{Z}, \ell)$, where $\mathcal{W} \in \mathbb{R}^D$ is a convex set and $\ell(z, \cdot): \mathcal{W} \to \mathbb{R}$ is a convex function for every $z \in \mathcal{Z}$.*

**Definition 2** (Convex-Lipschitz-Bounded (CLB)). *An SCO problem is called CLB if i) for every $w \in \mathcal{W}$, $\|w\| \leq R$, and ii) the loss function is convex and L-Lipschitz, i.e., $\forall z \in \mathcal{Z}, \forall w_1, w_2 \in \mathcal{W}: |\ell(z, w_1) - \ell(z, w_2)| \leq L\|w_2 - w_1\|$. We denote this subclass of SCO problems by $\mathcal{C}_{L, R}$.*

## 3 New CMI-based bounds via stochastic projection and lossy compression

While the CMI-based bounds are known to be generally tighter than the corresponding MI ones and even tight in some settings [12, 14], they can become vacuous in some cases. This includes the Stochastic Convex Optimization (SCO) examples constructed in the recent works [43, 46], which we will discuss in

more detail in Section 4. For these (counter-)examples, it was shown in [43, 46] that CMI-type bounds do not vanish, so they fail to accurately describe the generalization error. In this section, we show that such limitations are *not* inherent to the CMI framework. In fact, by combining *stochastic projection* with *lossy compression* (analogously to [49], which addressed the MI case), we derive new CMI-based bounds that *do not* suffer from such limitations. For instance, when applied to the SCO examples of [43], we show in Section 4 that our new bounds resolve the limitations of other known CMI-based bounds as identified therein. These bounds are also shown in the appendices to apply to the analysis of the generalization error for subspace training algorithms trained with SGD or SGLD.

Our new bounds involve two main ingredients, *stochastic projection* and *lossy compression*.

**Stochastic projection.** Let $\Theta \in \mathbb{R}^{D \times d}$ be a random matrix with entries distributed according to some joint distribution $P_\Theta$, chosen independently of $\tilde{\mathbf{S}}$, In our approach, similar to [49], instead of considering the hypothesis $W \in \mathcal{W} \subseteq \mathbb{R}^D$ which lies in a $D$-dimensional space, we consider its *projection* $\Theta^\top W \in \mathbb{R}^d$ onto a smaller $d$-dimensional space, with $d \ll D$.

**Lossy Compression.** Let $\epsilon \in \mathbb{R}$ be given. An $\epsilon$-lossy algorithm is a (possibly) stochastic map $\hat{\mathcal{A}} \colon \mathcal{Z}^n \times \mathbb{R}^{D \times d} \to \hat{\mathcal{W}}$ that maps a pair $(S_n, \Theta)$ to a compressed hypothesis or model $\hat{W} \in \hat{\mathcal{W}} \subseteq \mathbb{R}^d$ generated according to some conditional kernel $P_{\hat{W}|S_n,\Theta}$ that satisfies

$$\mathbb{E}_{P_{S_n,W} P_\Theta P_{\hat{W}|S_n,\Theta}} \left[ \text{gen}(S_n, W) - \text{gen}(S_n, \Theta\hat{W}) \right] \le \epsilon.$$

This constraint guarantees that, when projected back onto the original hypothesis space of dimension $D$, the compressed model $\hat{W}$ has an average generalization error which is within at most $\epsilon$ from that of the original model $W$. In a sense, one works with a compressed model $\hat{W}$ which lies in a much smaller dimension space, but with the guarantee that this causes almost no increase in the generalization error. In effect, the *auxiliary* projected-back model $\Theta\hat{W}$ substitutes the original model $W$.

The concept of a lossy algorithm, also referred to as a "surrogate" or "compressed" algorithm, was introduced in [37, 51, 52] and shown therein to be key to obtaining tighter, non-vacuous, generalization bounds. In this paper, we consider a particular lossy algorithm that involves a suitable stochastic projection followed by quantization. Specifically, we constrain the general conditional $P_{\hat{W}|S_n,\Theta}$ to take the specific form $P_{\hat{W}|\Theta^\top W}$, where $W = \mathcal{A}(S_n)$. Formally, one imposes the Markov chain $(S_n, \Theta, W) - \Theta^\top W - \hat{W}$ or equivalently $P_{\hat{W}|S_n,\Theta,W} = P_{\hat{W}|\Theta^\top W}$. In other words, we let $\hat{\mathcal{A}}(S_n, \Theta) = \tilde{\mathcal{A}}(\Theta^\top \mathcal{A}(S_n))$, where $\tilde{\mathcal{A}} \colon \mathbb{R}^d \to \hat{\mathcal{W}}$ is defined via the Markov kernel $P_{\hat{W}|\Theta^\top \mathcal{A}(S_n)}$.

Our generalization bounds that will follow are expressed in terms of *disintegrated* CMI, defined as follows. Let a super-sample $\tilde{\mathbf{S}}$ and a stochastic projection matrix $\Theta$ be given. The *disintegrated* CMI of an algorithm $\hat{\mathcal{A}} \colon \mathcal{Z}^n \to \hat{\mathcal{W}}$ is defined as

$$\text{CMI}^\Theta(\tilde{\mathbf{S}}, \hat{\mathcal{A}}) \triangleq \mathsf{I}^{\tilde{\mathbf{S}},\Theta}(\hat{\mathcal{A}}(\tilde{\mathbf{S}}_{\mathbf{J}}, \Theta); \mathbf{J}),$$

where $\hat{\mathcal{A}}(\tilde{\mathbf{S}}_{\mathbf{J}}, \Theta) = \tilde{\mathcal{A}}(\Theta^\top \mathcal{A}(\tilde{\mathbf{S}}_{\mathbf{J}})) = \hat{W}$ and $\mathsf{I}^{\tilde{\mathbf{S}},\Theta}(\hat{\mathcal{A}}(\tilde{\mathbf{S}}_{\mathbf{J}}, \Theta); \mathbf{J})$ is the CMI given an instance of $\tilde{\mathbf{S}}$ and $\Theta$, computed using the joint distribution $P_{\mathbf{J}} \otimes P_{W|\tilde{\mathbf{S}}_{\mathbf{J}}} \otimes P_{\hat{W}|\Theta^\top W}$, with $P_{\mathbf{J}} = \text{Bern}(1/2)^{\otimes n}$.

The next theorem states our main generalization bound and is proved in Appendix E.

**Theorem 1.** *Let a learning algorithm* $\mathcal{A} \colon \mathcal{Z}^n \to \mathcal{W}$ *where* $\mathcal{W} \subseteq \mathbb{R}^D$ *be given. Then, for every* $\epsilon \in \mathbb{R}$, *every* $d \in \mathbb{N}$, *and every* projected model quantization *set* $\hat{\mathcal{W}} \subseteq \mathbb{R}^d$, *we have*

$$\text{gen}(\mu, \mathcal{A}) \le \inf_{P_{\hat{W}|\Theta^\top W}} \inf_{P_\Theta} \mathbb{E}_{P_{\tilde{\mathbf{S}}} P_\Theta} \left[ \sqrt{\frac{2\Delta\ell_{\hat{w}}(\tilde{\mathbf{S}}, \Theta)}{n} \text{CMI}^\Theta(\tilde{\mathbf{S}}, \hat{\mathcal{A}})} \right] + \epsilon, \tag{2}$$

*where* $\hat{W} \in \hat{\mathcal{W}}$, $\Theta \in \mathbb{R}^{D \times d}$, *the infima are over all arbitrary choices of Markov kernel* $P_{\hat{W}|\Theta^\top W}$ *and distribution* $P_\Theta$ *that satisfy the following distortion criterion:*

$$\mathbb{E}_{P_{S_n,W} P_\Theta P_{\hat{W}|\Theta^\top W}} \left[ \text{gen}(S_n, W) - \text{gen}(S_n, \Theta\hat{W}) \right] \le \epsilon, \tag{3}$$

*and the term* $\Delta\ell_{\hat{w}}(\tilde{\mathbf{S}}, \Theta)$ *is given by*

$$\Delta\ell_{\hat{w}}(\tilde{\mathbf{S}}, \Theta) \coloneqq \mathbb{E}_{P_{W|\tilde{\mathbf{S}}} P_{\hat{W}|\Theta^\top W}} \left[ \frac{1}{n} \sum_{i \in [n]} (\ell(Z_{i,0}, \Theta\hat{W}) - \ell(Z_{i,1}, \Theta\hat{W}))^2 \right]. \tag{4}$$

Observe that $P_{W|\tilde{\mathbf{S}}} = \mathbb{E}_{P_\mathbf{J}}[P_{W|\tilde{\mathbf{S}}_\mathbf{J}}]$. Also, if $\ell(\cdot, \cdot) \in [0, C]$ for some non-negative constant $C \in \mathbb{R}_+$, then it is easy to see that the term $\Delta\ell_{\hat{w}}(\tilde{\mathbf{S}}, \Theta)$ is bounded from the above as $\Delta\ell_{\hat{w}}(\tilde{\mathbf{S}}, \Theta) \leq C^2$.

The result of Theorem 1 essentially means that the generalization error of the original model is upper bounded by a term that depends on the CMI of the auxiliary model $\hat{W}$ plus an additional distortion term that quantifies the generalization gap between the auxiliary and original models. The rationale is that, although the (worst-case) CMI term still depends on the dimension $d$ after stochastic projection, this dimension corresponds to a subspace of the original hypothesis space and can be chosen arbitrarily small in order to guarantee that the bound vanishes with $n$. Also, the term in left-hand-side (LHS) of equation 3 represents the average distortion (measured by the difference of induced generalization errors) between the original model and the one obtained after projecting back the auxiliary compressed model onto the original hypothesis space. The analysis of this term may seem non-easy; but as visible from the proof, it is not. This is because, defined as a difference term, its analysis does not necessitate accounting for statistical dependencies between $S$ and $W$. Instead, one only needs to account for the effect of the following sources of randomness: **i)** the stochastic projection matrix, **ii)** the quantization noise, and **iii)** discrepancies between the empirical measure of $S$ and the true unknown distribution $\mu$. As shown in the proofs, the analysis of the distortion term involves the use of classic concentration inequalities. Furthermore, the construction of $\hat{W}$ allows us to consider the worst-case bound for the CMI-terms of the RHS of equation 2 without losing the order-wise optimality in certain cases.

We close this section by noting that it is well known that CMI-type bounds can be improved by application of suitable techniques such as *random-subset* or *individual sample* techniques or in order to get fast rates $\mathcal{O}(1/n)$ for small empirical risk regimes, see, e.g., [20, 53, 54]. These same techniques can be applied straightforwardly to our bound of Theorem 1 to get improved ones. For the sake of brevity, we do not elaborate on this here; and we refer the reader to the supplements where a single-datum version of Theorem 1 is provided.

## 4 Application to resolving recently raised limitations of classic CMI bounds

Prior works [43, 46] have recently reported carefully constructed counter-example learning problems and have shown that classic MI-based and CMI-based bounds fail to provide meaningful results when applied to them. In this section, we show that the careful addition of our stochastic projection along with our lossy compression resolves those issues, in the sense that the resulting new bound (our Theorem 1), which is still of CMI-type, now yields meaningful results when applied to those counter-examples. In essence, the improvement is brought up by: (i) noticing that the aforementioned negative results for standard CMI-based generalization error bounds rely heavily on that the dimension of the hypothesis space grows fast with $n$ (over-parameterized regime), e.g., as $\Omega(n^4 \log n)$ in the considered counter-examples of [43], which calls for suitable projection onto a smaller dimension space in which this does not hold, and (ii) properly accounting for the distortion induced in the generalization error after projection back to the original high dimensional space.

First, we recall briefly the counterexamples mentioned in [43] and [46]; and, for each of them, we show how our bound of Theorem 1 applies successfully to it. Recall the definitions of a stochastic convex optimization (SCO) problem and a Convex-Lipschitz-Bounded (CLB) SCO problem as given, respectively, in Definition 1 and Definition 2.

**Definition 3** ($\varepsilon$-learner for SCO). *Fix $\epsilon > 0$. For a given SCO problem $(\mathcal{W}, \mathcal{Z}, \ell)$, $\mathcal{A} = \{\mathcal{A}_n\}_{n \geq 1}$ is called an $\varepsilon$-learner algorithm with sample complexity $N \colon \mathbb{R} \times \mathbb{R} \to \mathbb{N}$ if the following holds: for every $\delta \in (0, 1]$ and $n \geq N(\varepsilon, \delta)$ we have that for every $\mu \in \mathcal{M}_1(\mathcal{Z})$, where $\mathcal{M}_1(\mathcal{Z})$ denotes the set of probability measures on $\mathcal{Z}$, with probability at least $1 - \delta$ over $S_n \sim \mu^{\otimes n}$ and internal randomness of $\mathcal{A}$,*

$$\mathcal{R}(\mathcal{A}_n(S_n)) - \min_{w \in \mathcal{W}} \mathcal{R}(w) \leq \varepsilon. \tag{5}$$

### 4.1 Counter-example of Attias et al. [2024] for CLB class

Denote by $\mathcal{B}_D(\nu)$ the $D$-dimensional ball of radius $\nu \in \mathbb{R}_+$.

**Definition 4** (Problem instance $\mathcal{P}_{cvx}^{(D)}$). *Let $L, R \in \mathbb{R}_+$, $\mathcal{Z} \subseteq \mathcal{B}_D(1)$, and $\mathcal{W} = \mathcal{B}_D(R)$. Define the loss function $\ell \colon \mathcal{Z} \times \mathcal{W} \to \mathbb{R}$ as*

$$\ell_c(z, w) = -L\langle w, z \rangle.$$

*We denote this SCO problem instance as $\mathcal{P}_{cvx}^{(D)}$. It is easy to see that this optimization problem belongs to the subclass $\mathcal{C}_{L,R}$ of SCO problems as defined in Definition 2.*

For this (counter-) example learning problem, [43] have shown that for every $\varepsilon$-learner there exists a data distribution for which the CMI bound of equation 1 for the optimal sample complexity, which is $\Theta\left(\left(\frac{LR}{\varepsilon}\right)^2\right)$ as shown in [50], scales just as $\Theta(LR)$. For instance, that CMI-bound on the generalization error does *not* decay with the size $n$ of the training dataset!

**Theorem 2** (CMI-accuracy tradeoff, [43, Theorems 4.1 and 5.2]). *Let $\varepsilon_0 \in (0,1)$ be a universal constant. Consider the above defined $\mathcal{P}_{cvx}^{(D)}$ problem instance with parameters $(L, R)$. Consider any $\epsilon \leq \epsilon_0$ and for any algorithm $\mathcal{A} = \{\mathcal{A}_n\}_{n \in \mathbb{N}}$ that $\varepsilon$-learns $\mathcal{P}_{cvx}^{(D)}$ with sample complexity $N(\cdot,\cdot)$. Then, the following holds: **i.** For every $\delta \leq \varepsilon$, $n \geq N(\varepsilon, \delta)$, and $D = \Omega\left(n^4 \log(n)\right)$,[3] there exists a set $\mathcal{Z} \subseteq \mathcal{B}_D(1)$ and a data distribution $\mu \in \mathcal{M}_1(\mathcal{Z})$, denoted as $\mu_{p^*}$, such that $\mathsf{CMI}(\mu, \mathcal{A}_n) = \Omega\left(\left(\frac{LR}{\varepsilon}\right)^2\right)$. **ii.** In particular, considering the optimal sample complexity $N(\varepsilon, \delta) = \Theta\left(\frac{L^2 R^2}{\varepsilon^2}\right)$, the CMI generalization bound of equation 1 equals $LR\sqrt{8\mathsf{CMI}(\mu, \mathcal{A}_n)/N(\varepsilon, \delta)} = \Theta(LR)$.*

For this example, it was further shown [43, Corollary 5.6] that application of the *individual sample* technique of [55, 56] (which is traditionally used to avoid the unbounded-ness issue as instance of so called *randomized-subset* techniques wherein the linearity of the expectation operator is used to obtain an average bound for the loss on randomly chosen subsets of the training set rather than the loss averaged over the full training set) actually yields the very same bound order-wise; and, thus, it does not resolve the issue for this counter-example.

Furthermore, as shown in [43, Equation 1], the expectation of the LHS of equation 5 can be bounded as

$$\mathbb{E}\left[\mathcal{R}(\mathcal{A}_n(S_n))\right] - \min_{w \in \mathcal{W}} \mathcal{R}(w) \leq LR\sqrt{\frac{8\mathsf{CMI}(\mu, \mathcal{A}_n)}{n}} + \mathbb{E}\left[\widehat{\mathcal{R}}_n(\mathcal{A}_n(S_n)) - \min_{w \in \mathcal{W}} \widehat{\mathcal{R}}_n(w)\right]. \quad (6)$$

Thus, while the LHS of this inequality is bounded from above by $\varepsilon$ by assumption, its right-hand side (RHS) is $\Theta(LR)$ by Theorem 2. This means that the CMI bound of equation 1 fails to describe well the excess error of the LHS. In [43], this was even somewhat extrapolated to negatively answer the question about "*whether the excess error decomposition using CMI can accurately capture the worst-case excess error of optimal algorithms for SCOs*".

The above applies for any $\varepsilon$-learner of the problem instance $\mathcal{P}_{cvx}^{(D)}$ when $\mathcal{Z} = \{\pm 1/\sqrt{D}\}^D$ and $\mu_{p^*}(z) = \prod_{k=1}^{D}\left(\frac{1 + \sqrt{D}z_k p_k^*}{2}\right)$,[4] for all $z = (z_1, \ldots, z_D)$, where $p^* = (p_1^*, \ldots, p_D^*) \in [-1,1]^D$.

The next theorem shows that when applied to the aforementioned counter-example, our new CMI-bound of Theorem 1 does *not* suffer from those shortcomings. Also, this holds true for: **(i)** arbitrary values of the dimension $D \in \mathbb{N}$ including $n$-dependent ones, **(ii)** arbitrary learning algorithms (including the $\varepsilon$-learners of $\mathcal{P}_{cvx}^{(D)}$), **(iii)** arbitrary choices of $\mathcal{Z} \subseteq \mathcal{B}_D(1)$ and **(iv)** arbitrary data distributions $\mu$.

**Theorem 3.** *For every learning algorithm $\mathcal{A} \colon \mathcal{Z}^n \to \mathcal{W}$ of the instance $\mathcal{P}_{cvx}^{(D)}$ defined as in Definition 4, the generalization bound of Theorem 1 yields*

$$\mathrm{gen}(\mu, \mathcal{A}) \leq \frac{8LR}{\sqrt{n}}.$$

*In particular, setting $N(\varepsilon, \delta) = \Theta\left(\frac{L^2 R^2}{\varepsilon^2}\right)$ for $\varepsilon$-learner algorithms we get*

$$\mathrm{gen}(\mu, \mathcal{A}) = \mathcal{O}\left(\varepsilon\right).$$

The proof of Theorem 3 is deferred to Appendix F.2.

Some remarks are in order. First, while when applied to the studied counter-example the CMI bound of equation 1 yields a bound of the order $\Theta(LR)$, i.e., one that does *not* decay with $n$, our new CMI-based bound of Theorem 1 yields one that decays with $n$ as $\mathcal{O}(LR/\sqrt{n})$. Second, when specialized to the

---

[3]The arXiv version of [43] requires a smaller increase of $D$ with $n$; namely, $D = \Omega\left(n^2 \log(n)\right)$. Here, we consider values of $D$ that are mentioned in the published PMLR version of the document, i.e., $D = \Omega\left(n^4 \log(n)\right)$; but the approach and results that will follow also hold for $D = \Omega\left(n^2 \log(n)\right)$.

[4]In the construction of [43], by changing $n$, the data distribution changes, but, for better readability, we drop such dependence in the notation.

case of $\varepsilon$-learner algorithms and considering the sample complexity $\Theta\left(\left(\frac{LR}{\varepsilon}\right)^2\right)$, we get a bound on the generalization error of the order $\mathcal{O}(\varepsilon)$. Using this bound, we can write

$$\mathbb{E}_{P_{S_n,W}}\left[\mathcal{R}(\mathcal{A}_n(S_n))\right] - \min_{w\in\mathcal{W}}\mathcal{R}(w) \leq \mathcal{O}(\varepsilon) + \mathbb{E}_{P_{S_n,W}}\left[\widehat{\mathcal{R}}_n(\mathcal{A}_n(S_n)) - \min_{w\in\mathcal{W}}\widehat{\mathcal{R}}_n(w)\right]. \qquad (7)$$

Contrasting with equation 6 and noticing that if the second term of the summation of the RHS of equation 7 (optimization error) is small then both sides of equation 7 are $\mathcal{O}(\epsilon)$, it is clear that now the excess error decomposition using our new CMI-based bound can accurately capture the worst-case excess error. Third, as it can be seen from the proof, stochastic projection onto a one-dimensional space, i.e., $d = 1$, is sufficient to get the result of Theorem 3. In essence, this is the main reason why, in sharp contrast with projection- and lossy-compression-free CMI-bounds, ours of Theorem 1 does *not* become vacuous. That is, one can reduce the effective dimension of the model for the studied example even if the original dimension $D$ is allowed to grow with $n$ as $\Omega(n^4\log(n))$ as judiciously chosen in[43] for the purpose of making classic CMI-based bounds fail. Furthermore, it is worth noting that, for this problem, the projection is performed using the famous Johnson-Lindenstrauss [57] dimension reduction algorithm. Since this dimension reduction technique is "lossy", controlling the induced distortion is critical. To do so, we introduce an additional lossy compression step by adding independent noise in the lower-dimensional space. This approach is reminiscent of lossy source coding and allows to obtain possibly tighter bounds on the quantized, projected model. Finally, we mention that for bigger class problem instances or for the memorization problem of Section 5, projection onto one-dimensional spaces may not be enough to get the desired order $\mathcal{O}(LR/\sqrt{n})$. In Appendix B, it will be shown that for generalized linear stochastic optimization problems, one may need $d = \Theta(\sqrt{n})$. Similarly, in Section 5 and Appendix C, projections with $d = n^{2r-1}$, $r < 1$ and $d = \Theta(\log n)$ are used.

### 4.2 Counter-example of Attias et al. [2024] for CSL class

The question of whether classic CMI-bounds and individual-sample versions thereof may still fail if one considers more structured subclasses of SCO problems was raised (and answered positively!) in Attias et al. [43]. For convenience, we recall the following two definitions.

**Definition 5** (Convex set-Strongly Convex-Lipschitz (CSL)). *An SCO problem is called CSL if i) the loss function is $L$-Lipschitz, and ii) the loss function is $\lambda$-strongly convex, i.e., $\forall z \in \mathcal{Z}$, $\forall w_1, w_2 \in \mathcal{W}$: $\ell(z, w_2) \geq \ell(z, w_1) + \langle\partial\ell(z, w_1), w_2 - w_1\rangle + \frac{\lambda}{2}\|w_2 - w_1\|^2$, where $\partial\ell(z, w_1)$ is the subgradient of $\ell(z, \cdot)$ at $w_1$. We denote this subclass by $\mathcal{C}_{L,\lambda}$.*

**Definition 6** (Problem instance $\mathcal{P}_{scvx}^{(D)}$). *Let $\lambda, R \in \mathbb{R}_+$, $\mathcal{Z} \subseteq \mathcal{B}_D(1)$, and $\mathcal{W} = \mathcal{B}_D(R)$. Define the loss function $\ell: \mathcal{Z} \times \mathcal{W} \to \mathbb{R}$ as $\ell_{sc}(z, w) = -L_c\langle w, z\rangle + \frac{\lambda}{2}\|w\|^2$. We denote this SCO problem as $\mathcal{P}_{scvx}^{(D)}$, which belongs to $\mathcal{C}_{L,\lambda}$, with $L = L_c + \lambda R$.*

Setting $\lambda = L_c = R = 1$, $D = \Omega(n^4\log(n))$, $\delta = \mathcal{O}(1/n^2)$, $\mathcal{Z} = \{\pm 1/\sqrt{D}\}^D$ and for a particular data distribution that is carefully chosen therein (not reproduced here for brevity), [43, Theorem 4.2] states that for any learning algorithm that $\varepsilon$-learns the problem instance $\mathcal{P}_{scvx}^{(D)}$,

$$\mathsf{CMI}(\mu, \mathcal{A}_n) = \Omega\left(\frac{1}{\varepsilon}\right).$$

Moreover, the application of the individual-sample technique does not result in better decay of the bound order-wise [43, Corollary 5.7].

Noticing that (i) the loss $\ell_{sc}(z, w) = -L_c\langle w, z\rangle + \frac{\lambda}{2}\|w\|^2$ considered in Definition 6 differs from that $\ell_{sc}(z, w) = -L\langle w, z\rangle$ of Definition 4 essentially through the added squared magnitude of the model and (ii) that addition does not alter the generalization error of a given learning algorithm, then it is easy to see that Theorem 3 also applies for the problem $\mathcal{P}_{scvx}^{(D)}$ at hand; and, in this case, it gives a bound of the order $\mathcal{O}(1/\sqrt{n})$. This is stated in the next proposition, which is proved in Appendix F.3.

**Proposition 1.** *For every learning algorithm $\mathcal{A}: \mathcal{Z}^n \to \mathcal{W}$ of the instance $\mathcal{P}_{scvx}^{(D)}$ defined as in Definition 6 the generalization bound of Theorem 1 yields*

$$\mathrm{gen}(\mu, \mathcal{A}) \leq \frac{8L_c R}{\sqrt{n}}.$$

*In particular, choosing $L_c = R = \lambda = 1$ and setting $N(\varepsilon, \delta) = \frac{c}{\varepsilon}$ for some non-negative constant $c \in \mathbb{R}_+$ for the ERM algorithm (which is an $\varepsilon$-learner – see, e.g., [50, Theorem 6]), one gets $\mathrm{gen}(\mu, \mathcal{A}) = \mathcal{O}(\sqrt{\varepsilon})$.*

### 4.3 Counter-example of Livni [2023]

The counter-example of [46] is the same as the problem instance of Definition 4, with the one difference that the loss function is taken to be the squared distance instead of the inner product, *i.e.,* $\ell(z, w) = -L \|w - x\|^2$, for some non-negative constant $L \in \mathbb{R}_+$. Livni [46] has shown that the MI bound of [11] (which is a single-datum bound) fails and becomes vacuous when evaluated for this particular learning problem. However, since $\ell(z, w) = -L \|x\|^2 - L \|w\|^2 + 2L \langle w, x \rangle$ and noticing that the squared norm terms do not alter the generalization error relative to when computed for a loss function given by only the inner-product term, it follows that Theorem 3 still applies and gives a bound of the order $\mathcal{O}(1/\sqrt{n})$ for this problem instance. In addition, for the optimal sample complexity, the bound is $\mathcal{O}(\varepsilon)$. In essence, this means that unlike the MI bound of [11], our new CMI-based bound of Theorem 1 does not become vacuous when applied to the problem at hand.

In Appendix B, we apply the bound of Theorem 1 to a wider family of generalized linear stochastic optimization problems. In particular, we show that no counter-example could be found for which the bound of Theorem 1 does not vanish, even if one considers the bigger class of generalized linear stochastic optimization problems in place of the SCO class problems of [43].

## 5 Memorization

Loosely speaking, a learning algorithm is said to "memorize" if by only observing its output model, an adversary can correctly guess elements of the training data among a given super-sample. For the CLB and CSL subclasses of problems studied in Section 4, Attias et al. [43] showed that there are problem instances for which, for any $\varepsilon$-learner algorithm, there exists a data distribution under which the learning algorithm "memorizes" most of the training data. This is obtained by designing an adversary capable of identifying a significant fraction of the training samples.

In this section, we show that given a learning algorithm $\mathcal{A}$ that memorizes the training samples, one can find (via stochastic projection and lossy compression) an alternate learning algorithm $\tilde{\mathcal{A}}$ with comparable generalization error and that does *not* memorize the training data.[5]

**Definition 7** (Recall Game [43, Definition 4.3]). *Given* $\mathcal{A} = \{\mathcal{A}_n\}_{n \geq 1}$, *let* $\mathcal{Q} \colon \mathbb{R}^D \times \mathcal{Z} \times \mathcal{M}_1(\mathcal{Z}) \to \{0, 1\}$ *be an adversary for the following game. For* $i \in [n]$, *given a fresh data point* $Z_i' \sim \mu$ *independent of* $(Z_i, W)$, *let* $Z_{i,1} = Z_i$ *and* $Z_{i,0} = Z_i'$. *Then, the adversary is given* $Z_{i,K_i}$, *where* $K_i \sim Bern(1/2)$ *is independent of other random variables. The adversary declares* $\hat{K}_i \triangleq \mathcal{Q}(W, Z_{i,K_i}, \mu)$ *as its guess of* $K_i$.

The game consists of $n$ rounds. At each round $i \in [n]$, a pair $(Z_{i,0}, Z_{i,1})$ is considered and the adversary makes two independent guesses: one for the sample $Z_{i,0}$, the other for $Z_{i,1}$.

**Definition 8** (Soundness and recall [43, Definition 4.4]). *Consider the setup of Definition 7. Assume that the adversary plays the game in* $n$ *rounds. For every round* $i \in [n]$, *the adversary plays two times, independently of each other, using respectively* $(W, Z_{i,0}, \mu)$ *and* $(W, Z_{i,1}, \mu)$ *as input. Then, for a given* $\xi \in [0, 1]$, *the adversary is said to be* $\xi$-sound *if* $\mathbb{P} (\exists i \in [n] \colon \mathcal{Q}(W, Z_{i,0}, \mu) = 1) \leq \xi$. *Also, the adversary certifies the recall of* $m$ *samples with probability* $q \in [0, 1]$ *if* $\mathbb{P} \left( \sum_{i \in [n]} \mathcal{Q}(W, Z_{i,1}, \mu) \geq m \right) \geq q$. *If both conditions are met, we say that the adversary* $(m, q, \xi)$-traces *the data.*

Clearly, the concept of $(m, q, \xi)$-*tracing* the data by an adversary is most interesting for values of $(m, q, \xi)$ that are such that: $\xi$ is small (i.e., the adversary makes accurate predictions), $m$ is large and $q$ is non-negligible (i.e., the adversary can recall a significant part of the training data). As Lemma 1, which is stated in Appendix C.1, asserts, certain values of $(m, q, \xi)$ can be attained even by a "dummy" adversary that makes guesses without even looking at the given data sample.

For the problem instance $\mathcal{P}_{cvx}^{(D)}$, Attias et al. [43] have shown that, for every $\epsilon$-learner algorithm, there exist a distribution and an adversary that is capable of identifying a significant portion of the training data.

**Theorem 4** ([43, Theorem 4.5]). *Consider the* $\mathcal{P}_{cvx}^{(D)}$ *problem instance of Definition 4 with* $L = R = 1$. *Fix arbitrary* $\xi \in (0, 1]$ *and let* $\mathcal{Z} = \{\pm 1/\sqrt{D}\}^D$. *Let* $\varepsilon_0 \in (0, 1)$ *be a universal constant. Let* $\varepsilon > 0$ *such that* $\varepsilon < \varepsilon_0$, $\delta < \varepsilon$. *Then, given any* $\varepsilon$-learner algorithm $\mathcal{A}$ *with sample complexity* $N(\varepsilon, \delta) = \Theta(\log(1/\delta)/\varepsilon^2)$, *there exist a data distribution* $\mu_{p^*}$ *and an adversary such that for* $n = N(\varepsilon, \delta)$ *and* $D = \Omega(n^4 \log(n/\xi))$, *the adversary* $\left( \Omega(1/\varepsilon^2), 1/3, \xi \right)$-traces *the data.*

---

[5]The memorization problem has also been studied in [58] via some examples in which the data distribution $\mu$ is not fixed and comes from a meta-distribution, i.e. $\mu \sim P_\mu$. Instead of using the recall game, [58] measured the amount of memorization by $I(S; W | \mu)$.

A key implication of Theorem 4 is that, for some fixed $q > 0$, the result holds even when $\xi \in (0, 1]$ is arbitrarily small and $m = \Omega(n)$ (by choosing $\varepsilon = \mathcal{O}(1/\sqrt{n})$). In other words, for the considered class of problems $\mathcal{P}_{cvx}$ with data drawn from $\mu_{p^*}$, the constructed adversary can provably trace an arbitrarily large part of the training dataset.

We show that the stochastic projection and lossy compression techniques used in the CMI framework can partially mitigate this memorization issue, in a sense that will be made precise in Theorem 8. To this end, we first establish a general result on memorization.

**Theorem 5.** *Consider any learning algorithm $\mathcal{A} = \{\mathcal{A}_n\}_{n \geq 1}$ such that $\mathsf{CMI}(\mu, \mathcal{A}_n) = o(n)$. Then, for any adversary for this learning algorithm that $(m, q, \xi)$-traces the data, the following holds: **i)** $m = o(n)$ or $\xi \geq q$, **ii)** if, for some $\alpha \in (0, 1)$ and $n_0 \in \mathbb{N}^*$, $m \geq \alpha n$ for every $n \geq n_0$, then for any $\epsilon \in (0, \alpha)$ it holds that: $\mathbb{P}\left( \sum_{i \in [n]} \mathcal{Q}(W, Z_{i,0}, \mu) \geq m' \right) \geq (\alpha - \epsilon)q$, where $m' = \left( \frac{\epsilon}{1/q + \epsilon - \alpha} \right) n - o(n) = \Omega(n)$.*

Theorem 5, whose proof is provided in Appendix G.1, applies to *any* learning problems. In particular, it is not limited to $\mathcal{P}_{cvx}^{(D)}$ or the CLB subclass. The argument relies on Fano's inequality for approximate recovery [59, Theorem 2]. We construct a suitable estimator of the index set $\mathbf{J}$ based on the adversary's guesses, and we show that if this estimator can correctly recover a fraction $c > \frac{1}{2}$ of the membership indices $\mathbf{J}$, then $\mathsf{CMI}(\mu, \mathcal{A}_n) = \Theta(n)$.

Theorem 5 **i)** means that if the CMI of a learning algorithm is of order $o(n)$, then any adversary that recalls a non-negligible fraction of the training dataset with some probability $q$ (*i.e.,* , $m = \Theta(n)$) is $q$-sound at best. This means that, in this regime, no adversary can do better than a dummy one that makes random guesses independently of the data (See Lemma 1 in Appendix C.1 for what is attainable by a dummy adversary). Theorem 5 **ii)** means that if an adversary recalls $\Omega(n)$ training samples with some probability, then it must also incorrectly guess the membership of $\Omega(n)$ test samples with some non-negligible probability.

Next, we use the result of Theorem 5 for $\mathcal{P}_{cvx}^{(D)}$ to show that while the output model $W$ of any $\varepsilon$-learner algorithm must memorize a significant fraction of the data (for some distribution) as asserted in Theorem 4 the auxiliary model $\Theta \hat{W}$ (which is obtained through suitable stochastic projection and lossy compression), achieves comparable generalization error *without* memorizing the data!

**Theorem 6.** *Consider the $P_{cvx}^{(D)}$ problem instance of Definition 4 with $L = R = 1$. For every $r > 0$, every $\mathcal{Z} \subseteq \mathcal{B}_D(1)$ and every learning algorithm $\mathcal{A}: \mathcal{Z}^n \to \mathbb{R}^D$, there exists another (compressed) algorithm $\mathcal{A}^*: \mathcal{Z}^n \to \mathbb{R}^D$, defined as $\mathcal{A}^*(S_n) \triangleq \Theta \tilde{\mathcal{A}}(\Theta^\top \mathcal{A}(S_n)) = \Theta \hat{W}$, where the projection matrix $\Theta \in \mathbb{R}^{D \times d}$, $d = 500r \log(n)$, is distributed according to some distribution $P_\Theta$ independent of $(S_n, W)$, such that for any data distribution $\mu$, the following conditions are met simultaneously:*

**i)** *the generalization error of the auxiliary model $\Theta \hat{W}$ satisfies*

$$\left| \mathbb{E}_{P_{S_n, W} P_\Theta P_{\hat{W} | \Theta^\top W}} \left[ \mathrm{gen}(S_n, W) - \mathrm{gen}(S_n, \Theta \hat{W}) \right] \right| = \mathcal{O}\left( n^{-r} \right), \tag{8}$$

**ii)** *if there exists an adversary that by having access to both $\Theta$ and $\hat{W}$ (and hence $\Theta \hat{W}$) $(m, q, \xi)$-traces the data, then it must be that: **a)** $m = o(n)$ or $\xi \geq q$, and **b)** if, for some $\alpha \in (0, 1)$ and $n_0 \in \mathbb{N}^*$, $m \geq \alpha n$ for every $n \geq n_0$, then for any $\epsilon \in (0, \alpha)$ it holds that: $\mathbb{P}\left( \sum_{i \in [n]} \mathcal{Q}(\Theta \hat{W}, Z_{i,0}, \mu) \geq m' \right) \geq (\alpha - \epsilon)q$, where $m' = \left( \frac{\epsilon}{1/q + \epsilon - \alpha} \right) n - o(n) = \Omega(n)$.*

Theorem 6, proved in Appendix G.2, holds for $\Theta$ being stochastic and shared with the adversary. In essence, it asserts that for any algorithm $\mathcal{A}(S) = W$, one can construct a suitable projected-quantized model $\hat{\mathcal{A}}(S, \Theta) = \hat{W}$ from which no adversary would be able to trace the data, for any data distribution $\mu$. It is appealing to contrast this result with that of [43, Theorem 4.5] on the necessity of memorization. Consider the SCO instance problem with $\mathcal{O}(1)$ convex-Lipschitz loss defined over the ball of radius one in $\mathbb{R}^D$ considered in [43, Theorem 4.5] and let an $\varepsilon$-learner algorithm $\mathcal{A}$ with output model $W$ and sample complexity $N(\varepsilon, \delta) = \Theta(\log(1/\delta)/\varepsilon^2)$ with $D = \Omega(n^4 \log(n/\xi))$ be given. The result of [43, Theorem 4.5] states that there exists a data distribution for which the algorithm $\mathcal{A}$ must memorize a big fraction of the training data. Applied to this particular instance problem, Theorem 6 asserts that if a random $\Theta$ is chosen and shared with the adversary then the auxiliary model $\Theta \hat{W}$ has the following guarantees: (i) for any data distribution, no adversary can trace the data, and (ii) on average over $\Theta$ the associated generalization error is arbitrarily close to that of the original model $W$. At first glance, this may seem to contradict the necessity of memorization stated in [43, Theorem 4.5]. It is important to note, however,

that the auxiliary algorithm does not satisfy the conditions required in [43, Theorem 4.5]; and, so, the latter does not apply to $\Theta\hat{W}$. In particular, while [43, Theorem 4.5] requires the model to be bounded, in our construction for every $w$ we have $\mathbb{E}_{\hat{W},\Theta}\left[\Theta\hat{W}\right] \approx w$ but $\mathbb{E}_{\hat{W},\Theta}\left[\left\|\Theta\hat{W}\right\|^2\right]$ increases roughly as $\frac{D}{d}$ (see Lemma 2 in Appendix C.4.1). As discussed after Lemma 2, this causes $\mathbb{E}_{\hat{W},\Theta}\left[\left\|\Theta\hat{W}\right\|^2\right]$ to grow as $\Omega(n^3)$ when $D = \Omega(n^4 \log(n/\xi))$, i.e., it becomes arbitrarily large as $n$ increases. Intuitively, this is what prevents an adversary from guessing correctly whether a sample has (or not) been used for training, and which makes some key proof steps of Attias et al. fail when applied to the auxiliary model $\Theta\hat{W}$. These steps are discussed in detail in Appendix C.4.2.

A somewhat weaker version of Theorem 6, which is stated in Theorem 8 in Appendix C.2, holds for the projection matrix $\Theta$ being *deterministic*. In a sense, it provides a stronger guarantee on the generalization error of the auxiliary model, in that the closeness to the performance of the original model holds now for the given $\Theta$ and not only in average over $\Theta$ as in Theorem 6. However, this comes at the expense of the auxiliary algorithm being dependent on the data distribution. A consequence of this is that the result does not preclude the existence of other distributions for which there would exist adversaries capable of tracing the data. Moreover, in Theorem 9 in Appendix C.3, we show that a similar result holds if one considers the closeness in terms of the population risk, instead of the generalization error.

Summarizing, neither of the results of Theorem 6 and Theorem 8 contradict those of [43]. In essence, they assert that for any learning algorithm $\mathcal{A}$ one can find an alternate auxiliary algorithm via stochastic projection combined with lossy compression for which no adversary would be able to trace the data; and, yet, the found auxiliary algorithm has generalization error that is arbitrarily close to that of the original model. Appendix C.3 extends this closeness to the population risk.

# 6 Implications and Concluding Remarks

**Sample-compression schemes**

Formally, a learning algorithm is a sample compression scheme of size $k \in \mathbb{N}$ if there exists a pair of mappings $(\phi, \psi)$ such that for all samples $S = (Z_1, \ldots, Z_n)$ of size $n \geq k$, the map $\phi$ compresses the sample into a length-$k$ sequence which the map $\psi$ uses to reconstruct the output of the algorithm, i.e., $\mathcal{A}(S) = \psi(\phi(S))$. Steinke and Zakynthinou [12] establish that if an algorithm $\mathcal{A}_n$ is a sample-compression scheme $(\phi, \psi)$ of size $k$, then it must be that the associated CMI is bounded from above as $\mathrm{CMI}(\mathcal{A}_n) \leq k \log(2n)$. The finding of [43] that, for certain SCO problem instances, every $\varepsilon$-learner algorithm must have CMI that blows up with $n$ (faster than $n$) was used therein to refute the existence of such sample-compression schemes for the studied SCO problems. The results of this paper may constitute a path to obtaining such schemes when the definition is extended to involve approximate reconstruction (in terms of induced generalization error) instead of the strict $\mathcal{A}_n(\cdot) = \psi(\phi(\cdot))$ of Littlesone and Warmuth [60].

**Fingerprinting codes and privacy attacks**

In [61], the authors study the problem of designing privacy attacks on mean estimators that expose a fraction of the training data. They show that a well-designed adversary can guess membership of the training samples from the output of every algorithm that estimates mean with high precision. Our results suggest that stochastic projection and lossy compression might be useful to construct differentially private codes that prevent such fingerprinting type attacks. For instance, while noise would naturally be one constituent of the recipe in this context, its injection in a suitable smaller subspace of the summary statistics might be the key enabler of privacy guarantees in such contexts.

**Concluding remarks**

In this work, we revisit recent limitations identified in conditional mutual information-based generalization bounds. By incorporating stochastic projections and lossy compression mechanisms into the CMI framework, we derive bounds that remain informative in stochastic convex optimization, thereby offering a new perspective on the results in [43, 46]. Our approach also provides a constructive resolution to the memorization phenomenon described in [43], by showing that for any algorithm and data distribution, one can construct an alternative model that does not trace training data while achieving comparable generalization.

Like prior work on information-theoretic bounds, our analysis applies to stochastic convex optimization. A natural, open question is whether and how these results can be extended to more general learning settings. Another key direction is to translate our theoretical findings into actionable design principles for learning algorithms with controlled generalization and compressibility.

## Acknowledgments

The authors thank the anonymous reviewers for their many insightful comments and suggestions. Their feedback and the ensuing discussions led to the alternative variants of Theorem 8 (*i.e.*, Theorem 6 and Theorem 9), and greatly shaped some of the paper's discussions. Kimia Nadjahi would also like to thank Mahdi Haghifam for the helpful discussions.

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

# Appendices

The appendices are organized as follows:

- In Appendix A, we present some extensions of Theorem 1, that are used in the subsequent sections.
- The results of Section 4 have been extended to a wider family of generalized linear stochastic optimization problems in Appendix B.
- Further results on memorization are presented in Appendix C. In particular
  - In Appendix C.1, we discuss what values of $(m, q, \xi)$ can be achieved by a "dummy adversary".
  - In Appendix C.2, we consider the case where the projection matrix $\Theta$ is fixed and shared with the adversary.
  - In Appendix C.3, we discuss how to provide guarantees on the closeness in terms of the population risk between the projected-quantized model to the original model.
  - In Appendix C.4, we provide technical lemmas used in the main text on reconciliation of our results with those of [43].
- The generalization error of subspace training algorithms is investigated in Appendix D. In particular, in Appendix D.1, we develop generalization bounds for the case where iterative optimization algorithms such as SGD and SGLD are used for the optimization of the subspace training algorithms.
- The proof of Theorem 1 is presented in Appendix E.
- In Appendix F, we present the proofs of the results presented in Section 4 and Appendix B regarding the applications of Theorem 1 to resolving recently raised limitations of classic CMI bounds. In particular,
  - a general Johnson-Lindenstrauss projection scheme $\mathrm{JL}(d, c_w, \nu)$ is introduced in Appendix F.1, which is used in the following subsections, with different choices of $(d, c_w, \nu)$,
  - Theorem 3 is proved in Appendix F.2,
  - Proposition 1 is proved in Appendix F.3,
  - Theorem 7 is proved in Appendix F.4,
  - and Lemma 4 is proved in Appendix F.5.
- Appendix G contains the proofs of the results in Section 5 and Appendix C, about the memorization. More precisely,
  - Theorem 5 is proved in Appendix G.1,
  - Theorem 6 is proved in Appendix G.2,
  - Lemma 1 is proved in Appendix G.3,
  - Theorem 8 is proved in Appendix G.4,
  - Theorem 9 is proved in Appendix G.5,
  - Lemma 2 is proved in Appendix G.6,
  - and Lemma 5 is proved in Appendix G.7.
- Lastly, Appendix H contains the proofs of the results of Appendix D on the generalization error of subspace training algorithms when trained using SGD or SGLD. More precisely,
  - Lemma 3 is proved in Appendix H.1,
  - Theorem 10 is proved in Appendix H.2,
  - Theorem 13 is proved in Appendix H.3,
  - and Lemma 6 is proved Appendix H.4.

## A  Extensions of Theorem 1

As mentioned in Section 3, Theorem 1 can be improved in several ways, similar to those proposed in [20, 53, 54]. Here, we state only the single-datum version of Theorem 1, which is used in Appendix D, followed by a remark about extending Theorem 1 and its corollary to more general lossy compression algorithms. Denote

$$\mathbf{J}_{-i} = \mathbf{J}_{[n]\setminus\{i\}}, \quad \tilde{\mathbf{S}}_{-i} \triangleq \tilde{\mathbf{S}}_{[n]\setminus\{i\},[2]} = \tilde{\mathbf{S}} \setminus \{Z_{i,0}, Z_{i,1}\}.$$

**Corollary 1.** *Consider the setup of Theorem 1. Then,*

$$\text{gen}(\mu, \mathcal{A}) \leq \inf_{P_{\hat{W}|\Theta^\top W}} \inf_{P_\Theta} \frac{1}{n} \sum_{i \in [n]} \mathbb{E}_{P_{\tilde{\mathbf{S}}} P_\Theta} \left[ \sqrt{2\Delta\ell_{\hat{w},i}(\tilde{\mathbf{S}}, \Theta) \mathsf{CMI}_i^\Theta(\tilde{\mathbf{S}}, \hat{\mathcal{A}})} \right] + \epsilon, \tag{9}$$

*and*

$$\text{gen}(\mu, \mathcal{A}) \leq \inf_{P_{\hat{W}|\Theta^\top W}} \inf_{P_\Theta} \frac{1}{n} \sum_{i \in [n]} \mathbb{E}_{P_{\tilde{\mathbf{S}}} P_\Theta P_{\mathbf{J}_{-i}}} \left[ \sqrt{2\Delta\ell_{\hat{w},i}(\tilde{\mathbf{S}}, \Theta) \mathsf{CMI}_{i,\mathbf{J}_{-i}}^\Theta(\tilde{\mathbf{S}}, \hat{\mathcal{A}})} \right] + \epsilon, \tag{10}$$

*where the infima are over $P_{\hat{W}|\Theta^\top W}$ and $P_\Theta$ that satisfy the distortion criterion*

$$\mathbb{E}_{P_{S_n,W} P_\Theta P_{\hat{W}|\Theta^\top W}} \left[ \text{gen}(S_n, W) - \text{gen}(S_n, \Theta\hat{W}) \right] \leq \epsilon, \tag{11}$$

*and where*

$$\mathsf{CMI}_i^\Theta(\tilde{\mathbf{S}}, \hat{\mathcal{A}}) \triangleq \mathsf{I}^{\tilde{\mathbf{S}},\Theta}(\hat{\mathcal{A}}(\tilde{\mathbf{S}}_\mathbf{J}, \Theta); J_i),$$

$$\mathsf{CMI}_{i,\mathbf{J}_{-i}}^\Theta(\tilde{\mathbf{S}}, \hat{\mathcal{A}}) \triangleq \mathsf{I}^{\tilde{\mathbf{S}},\mathbf{J}_{-i},\Theta}(\hat{\mathcal{A}}(\tilde{\mathbf{S}}_\mathbf{J}, \Theta); J_i),$$

$$\Delta\ell_{\hat{w},i}(\tilde{\mathbf{S}}, \Theta) \triangleq \mathbb{E}_{P_{W|\tilde{\mathbf{S}}} P_{\hat{W}|\Theta^\top W}} \left[ (\ell(Z_{i,0}, \Theta\hat{W}) - \ell(Z_{i,1}, \Theta\hat{W}))^2 \right].$$

To derive inequality 9, first note that by equation 11, it is sufficient to show that

$$\text{gen}(\mu, \hat{\mathcal{A}}) \leq \inf_{P_{\hat{W}|\Theta^\top W}} \inf_{P_\Theta} \frac{1}{n} \sum_{i \in [n]} \mathbb{E}_{P_{\tilde{\mathbf{S}}} P_\Theta} \left[ \sqrt{2\Delta\ell_{\hat{w},i}(\tilde{\mathbf{S}}, \Theta) \mathsf{CMI}_i^\Theta(\tilde{\mathbf{S}}, \hat{\mathcal{A}})} \right].$$

Next, using the linearity of the expectation, we can write

$$\mathbb{E}[\text{gen}(S_n, \hat{W})] = \frac{1}{n} \sum_{i \in [n]} \mathbb{E}[\text{gen}(\{Z_i\}, \hat{W})]$$

$$= \frac{1}{n} \sum_{i \in [n]} \mathbb{E}_{\tilde{\mathbf{S}}_{-i}} \left[ \mathbb{E}_{Z_i, \hat{W}}[\text{gen}(\{Z_i\}, \hat{W})] \right]. \tag{12}$$

Then applying Theorem 1 for each of the terms $\mathbb{E}_{Z_i, \hat{W}}[\text{gen}(\{Z_i\}, \hat{W})]$ yields equation 9.

The inequality 10 can be achieved similarly, by considering

$$\mathbb{E}[\text{gen}(S_n, \hat{W})] = \frac{1}{n} \sum_{i \in [n]} \mathbb{E}_{\tilde{\mathbf{S}}_{-i},\mathbf{J}_{-i}} \left[ \mathbb{E}_{Z_i, \hat{W}}[\text{gen}(\{Z_i\}, \hat{W})] \right],$$

instead of equation 12.

The results of Theorem 1 and, consequently, Corollary 1, are valid for a broader class of learning algorithms, $\mathcal{A}$, and lossy compression algorithms, $\hat{\mathcal{A}}$, as discussed in the remark below and shown in the proof of Theorem 1 in Appendix E.

**Remark 1.** *As shown in Appendix E, the bounds of Theorem 1 and consequently Corollary 1 hold if the learning algorithm $\mathcal{A}$ is aware of the projection matrix $\Theta$, i.e., if $\mathcal{A}: \mathcal{Z}^n \times \mathbb{R}^{D \times d} \to \mathcal{W}$ takes both the dataset $S$ and the projection matrix $\Theta$ as input in order to learn the model $W$. Moreover, the results of Theorem 1 and Corollary 1 are valid if the quantization step can also depend on $S$, $\Theta$ and $\mathcal{A}(S, \Theta)$. In this general case, $\hat{W} = \hat{\mathcal{A}}(S, \Theta) = \tilde{\mathcal{A}}(\Theta, S, \mathcal{A}(S, \Theta)) = \hat{W}$. This setting trivially includes the case in which $\mathcal{A}: \mathcal{Z}^n \to \mathcal{W}$ and the quantization depends only on $\Theta^\top \mathcal{A}(S, \Theta)$. For the ease of the exposition, we found it better not to state the result in its most general form.*

# B   Generalized linear stochastic optimization problems

In this section, we show that our bound of Theorem 1 can be applied successfully to get useful bounds on the generalization error of a family of generalized linear stochastic optimization problems that is wider than the ones considered previously in related prior art.

**Definition 9** (Generalized linear stochastic optimization). *Let $L, B, R \in \mathbb{R}_+$ and $\mathcal{W} = \mathcal{B}_D(R)$. Define the loss function $\ell_{gl}\colon \mathcal{Z} \times \mathcal{W} \to \mathbb{R}$ as*

$$\ell_{gl}(z, w) = g\left(\langle w, \phi(z)\rangle, z\right) + r(w),$$

*where $g\colon \mathbb{R} \times \mathcal{Z} \to \mathbb{R}$ is $L$-Lipschitz with respect to the first argument, $\phi\colon \mathcal{Z} \to \mathcal{B}_D(B)$ and $r\colon \mathcal{W} \to \mathbb{R}$ is some arbitrary function. Denote this problem as $\mathcal{P}_{glso}^{(D)}$.*

This class of problems is larger than the one considered in [50]. For instance, while the results of [50] require the $L$-Lipschitz function $g(\cdot, \cdot)$ and the function $r(\cdot)$ to be both convex to hold, our next theorem applies to arbitrary $L$-Lipschitz functions $g(\cdot, \cdot)$ and arbitrary functions $r(\cdot)$.

**Theorem 7.** *For every learning algorithm $\mathcal{A}\colon \mathcal{Z}^n \to \mathcal{W}$ of the instance problem $\mathcal{P}_{glso}^{(D)}$ defined in Definition 9, the generalization bound of Theorem 1 yields*

$$\mathrm{gen}(\mu, \mathcal{A}) = \mathcal{O}\left(\frac{LRB}{\sqrt[4]{n}}\right).$$

The proof, stated in Appendix F.4, is based on Theorem 1. In order to find a proper stochastic projection and quantization, we use the Johnson-Lindenstrauss (JL) dimensional reduction transformation in a space of dimension $d$. Then, we apply lossy compression to the projected model. Thanks to the combined projection-quantization, the disintegrated CMI can be bounded easily in the $d$-dimensional space. However, there are two main caveats to using the JL Lemma directly. First, one needs to bound the term $\Delta\ell_{\hat{w}}(\tilde{\mathbf{S}}, \Theta)$ (see equation 4). This is particularly difficult since the JL Lemma does not guarantee distance preservation in the original space of dimension $D$ after projecting back the quantized model. Second, bounding the distortion term is less easy than in Theorem 3, since using the Lipschitz property requires bounding the absolute value of the difference between inner products of the original and projected-quantized models. In essence, this is the reason why, by opposition to JL transformation for which it suffices to take $d = \log(n)$, here one needs a higher-dimensional projection space comparatively, with $d = \sqrt{n}$.

Theorem 7 shows that no counter-example could be found for which the bound of Theorem 1 does not vanish, even if one considers the bigger class of generalized linear stochastic optimization problems of Definition 9 in place of the SCO class problems of [43]. The convergence rate $\mathcal{O}(1/\sqrt[4]{n})$ of Theorem 7 is, however, not optimal. A better rate, $\mathcal{O}(1/\sqrt{n})$, seems to be achievable using Rademacher analysis and Talagrand's contraction lemma [62]. Using a more refined analysis, the same rate might be possible to achieve using our Theorem 1. More precisely, in the part of the current proof of Theorem 7 that analyses the distortion term, we do *not* account for the discrepancy between the empirical measure of $S$ and the true distribution $\mu$; and, instead, we consider a worst-case scenario. A finer analysis that takes such discrepancy into account should lead to a sharper expected concentration bound for the distortion term, and, so, a better rate.

## C   Further results on memorization

In this section, we provide further results on memorization. In Appendix C.1, we show that even a "dummy" adversary can trace the data for some values of $(m, q, \xi)$. In Appendix C.2, we study the case where the projection matrix $\Theta$ is deterministic. In Appendix C.3, we provide another variant of Theorem 8, in which we can guarantee the closeness of the projected-quantized model to the original model in terms of population risk (instead of the generalization error considered in Theorem 8). Finally, in Appendix C.4, we present some technical lemmas used in the discussions of Section 5 on the relation of our results with those established in [43].

### C.1   Dummy adversary

In this section, we show that certain values of $(m, q, \xi)$ are attainable by a "dummy" adversary who makes guesses without even looking at the given data sample.

**Lemma 1.** *Given a learning algorithm $\mathcal{A}_n\colon \mathcal{Z}^n \to \mathcal{W}$, there exists an adversary that $(m, q, \xi)$-traces the data for some $m \in [0, n]$ and $q, \xi \in [0, 1]$ if one of the following conditions holds: i) $\xi \geq q$, or ii) there exists an $\alpha \in [0, 1 - \xi] \cap [0, 1)$ such that $\sqrt[n]{1 - \frac{\xi}{1-\alpha}} + \sqrt{\frac{1}{2n}\log\left(\frac{1}{1-q/(1-\alpha)}\right)} + \frac{m}{n} \leq 1$.*

This lemma, proved in Appendix G.3, implies in particular that even a dummy adversary can $(m, q, \xi)$-trace the data in several cases: when $\xi$ is small, when $q$ is large, or when $\xi$ is small and $q$ is large, provided that $m = o(n)$.

## C.2 Deterministic projection

In this section, we show that in Theorem 6, one can allow $\Theta$ to be deterministic. However, this comes at the cost of being specific to a given data distribution.

**Theorem 8.** *Consider the $P_{cvx}^{(D)}$ problem instance of Definition 4 with $L = R = 1$. For every $r < 1$, every $\mathcal{Z} \subseteq \mathcal{B}_D(1)$, every data distribution $\mu$, and every learning algorithm $\mathcal{A}$, there exist a projection matrix $\Theta \in \mathbb{R}^{D \times d}$ with $d = \lceil n^{2r-1} \rceil$, a Markov Kernel $P_{\hat{W}|\Theta^\top W}$ and a compression algorithm $\mathcal{A}_\Theta^* \colon \mathcal{Z}^n \to \mathbb{R}^d$, defined as $\mathcal{A}_\Theta^*(S_n) \triangleq \tilde{\mathcal{A}}(\Theta^\top \mathcal{A}(S_n)) = \hat{W}$, such that the following conditions are met simultaneously:*

  *i) the generalization error of the auxiliary model $\Theta\hat{W}$ satisfies*

$$\left| \mathbb{E}_{P_{S_n,W} P_{\hat{W}|\Theta^\top W}} \left[ \text{gen}(S_n, W) - \text{gen}(S_n, \Theta\hat{W}) \right] \right| = \mathcal{O}\left(n^{-r}\right), \tag{13}$$

  *where the expectation is taken over $(S_n, W, \hat{W}) \sim P_{S_n,W} P_{\hat{W}|\Theta^\top W}$.*

  *ii) if there exists an adversary that by having access to both $\Theta$ and $\hat{W}$ (and hence $\Theta\hat{W}$) $(m, q, \xi)$-traces the data, then it must be that: **a)** $m = o(n)$ or $\xi \geq q$, and **b)** if, for some $\alpha \in (0, 1)$ and $n_0 \in \mathbb{N}^*$, $m \geq \alpha n$ for every $n \geq n_0$, then for any $\epsilon \in (0, \alpha)$ it holds that: $\mathbb{P}\left( \sum_{i \in [n]} \mathcal{Q}(\Theta\hat{W}, Z_{i,0}, \mu) \geq m' \right) \geq (\alpha - \epsilon)q$, where $m' = \left( \frac{\epsilon}{1/q + \epsilon - \alpha} \right) n - o(n) = \Omega(n)$.*

As shown in the proof in Appendix G.4, the constraint on the difference generalization error can be replaced with one with a faster decay with $n$, namely

$$\mathbb{E}_{P_{S_n,W} P_{\hat{W}|\Theta^\top W}} \left[ \text{gen}(S_n, W) - \text{gen}(S_n, \Theta\hat{W}) \right] = \mathcal{O}\left(n^{-r}\right)$$

for some $r \in [R]$ and $d = 500r \log(n)$. Also, if $n = N(\varepsilon, \delta)$, then $m, m' = \Omega\left(1/\varepsilon^2\right)$, which means that any adversary who $(m, q, \xi)$-traces the training data is deemed to misclassify any arbitrary big part of the test samples.

For the proof of Theorem 8, we first apply the projection-quantization approach of Theorem 3. Then, for a proper $\Theta$ that satisfies the distortion criterion of equation 13 and for which the CMI is $o(n)$ we apply Theorem 8. Note two important differences with Theorem 3. First, because one now deals with *absolute value* of the average difference of generalization errors one also needs to lower bound the average distortion. Also, for $r > 1/2$ a faster convergence rate of $\mathcal{O}(n^{-r})$ is required. This renders the analysis trickier and requires projection on a space of dimension $n^{2r-1}$.

## C.3 Guarantees on the population risk

In this section, we demonstrate that the closeness guarantee of the projected-quantized model and the original model can also be provided in terms of population risk.

**Theorem 9.** *Consider the $P_{cvx}^{(D)}$ problem instance of Definition 4 with $L = R = 1$. For every $r < 1/2$, every $\mathcal{Z} \subseteq \mathcal{B}_D(1)$, every data distribution $\mu$, and every learning algorithm $\mathcal{A}$, there exist a projection matrix $\Theta \in \mathbb{R}^{D \times d}$ with $d = \lceil n^{2r} \rceil$, a Markov Kernel $P_{\hat{W}|\Theta^\top W}$ and a compression algorithm $\mathcal{A}_\Theta^* \colon \mathcal{Z}^n \to \mathbb{R}^d$, defined as $\mathcal{A}_\Theta^*(S_n) \triangleq \tilde{\mathcal{A}}(\Theta^\top \mathcal{A}(S_n)) = \hat{W}$, such that the following conditions are met simultaneously:*

  *i) the generalization error of the auxiliary model $\Theta\hat{W}$ satisfies*

$$\left| \mathbb{E}_{P_{S_n,W} P_{\hat{W}|\Theta^\top W}} [\mathcal{L}(W) - \mathcal{L}(\Theta\hat{W})] \right| = \mathcal{O}\left(n^{-r}\right),$$

  *where the expectation is taken over $(S_n, W, \hat{W}) \sim P_{S_n,W} P_{\hat{W}|\Theta^\top W}$.*

  *ii) if there exists an adversary that by having access to both $\Theta$ and $\hat{W}$ (and hence $\Theta\hat{W}$) $(m, q, \xi)$-traces the data, then it must be that: **a)** Either $m = o(n)$ or $\xi \geq q$, and **b)** if $m = \Omega(n)$ then there exists $m' = \Omega(n)$ and $q' \in (0, 1]$ such that for sufficiently large $n$, it holds that $\mathbb{P}\left( \sum_{i \in [n]} \mathcal{Q}(\Theta\hat{W}, Z_{i,0}, \mu) \geq m' \right) \geq q'$.*

This result is proved in Appendix G.5. Furthermore, similarly to Theorem 8, the constraint on the difference of population risks can be replaced with one with a faster decay with $n$, namely

$$\mathbb{E}_{P_{S_n}, W P_{\hat{W} | \Theta^\top W}} \left[ \mathcal{L}(W) - \mathcal{L}(\Theta \hat{W}) \right] = \mathcal{O}\left( n^{-r} \right), \tag{14}$$

for some $r \in [R]$ and $d = 500r \log(n)$.

## C.4    Reconciliation with results of Attias et al. 2024

In this section, we provide the technical lemma showing that the norm two of the projected-quantized model, used in our results, is unbounded. Furthermore, we discuss in detail the steps of the proofs in [43] where this bounded assumption is needed.

### C.4.1    Uboundedness of the norm two of the projected-quantized model

In this section, for the projected-quantized algorithm $\Theta \hat{W}$, used in Theorem 8 and Theorem 6, we show that $\mathbb{E}_{\hat{W}, \Theta} \left[ \left\| \Theta \hat{W} \right\|^2 \right]$ blows-up with $n$ when $D/d$ grows with $n$. This lemma is proved in Appendix G.6.

**Lemma 2.** *Consider the $JL(d, c_w, \nu)$ transformation described in Appendix F.1, with some $d \in \mathbb{N}^+$, $c_w \in \left[ 1, \sqrt{5/4} \right)$, and $\nu \in (0,1]$. Then, for every $w \in \mathcal{W}$,*

$$\mathbb{E}_{\Theta, V_\nu} \left[ \left\| \Theta \hat{W} \right\|^2 \right] \geq \left( \frac{D + d + 1}{d} \right) \|w\|^2 - \sqrt{\frac{(D + d + 3)(D + d + 5)(d + 2)}{d^3}} \|w\|^2 e^{-0.1 d (c_w^2 - 1)^2}$$
$$- \frac{D \nu^2}{d}.$$

Consider $\|w\| = 1$ and let $D = n^4 \log(n/\xi)$ as considered in [43, Theorem 4.5]. We note that the notation $d$ used in [43] corresponds to the notation $D$ in this paper.

Then, considering the constructions used for Theorem 8 and Theorem 6, we have $c_w = 1.1$ and $\nu = 0.4$. Moreover, $d$ is chosen either as

$$d = 500r \log(n),$$

or

$$d = n^{2r - 1}, \quad r < 1.$$

Using Lemma 2 with these choices give

$$\mathbb{E}_{\Theta, V_\nu} \left[ \left\| \Theta \hat{W} \right\|^2 \right] = \Omega\left( n^4 \right).$$

and

$$\mathbb{E}_{\Theta, V_\nu} \left[ \left\| \Theta \hat{W} \right\|^2 \right] = \Omega\left( n^{4 - 2r} \log(n) \right) = \Omega\left( n^3 \log(n) \right),$$

respectively. Hence, in both cases $\mathbb{E}_{\Theta, V_\nu} \left[ \left\| \Theta \hat{W} \right\|^2 \right]$ grows at least as fast as $\Omega(n^3)$.

### C.4.2    Details of needed boundedness assumption in Attias et al.

As discussed before, [43, Theorem 4.1] and [43, Theorem 4.5] require the model to be bounded. However, as shown in the previous section, this assumption does not hold for the projected-quantized algorithm $\Theta \hat{W}$ when $D/d$ and $d$ grow with $n$. In this section, we discuss precisely where the bounded model assumption is necessary in the proofs of the impossibility results of [43].

- Proof of [43, Theorem 4.1] and recall analysis in the proof of [43, Theorem 4.5], in part, relies on an established upper bound $\Omega(1/\varepsilon^2)$ on the term $\mathbb{E}[|\mathcal{I}|]$, where the set $\mathcal{I}$ is the subset of columns of supersample such that one of the samples has a large correlation with the output of the algorithm and the other one has small correlation with the output of the algorithm.

- To establish this upper bound, in the last inequality of Page 19 of [43], it is assumed that the norm of the model is bounded. Now, when working with the model $\Theta \hat{W}$, the right-hand side of this inequality needs to be replaced by the $D/d$-dependent quantity $8\varepsilon^4 n^2 \frac{D}{d} + 2\varepsilon^2 = \Omega(n^5)$ when $D = \Omega(n^4)$ and $d = o(n)$. This has to be contrasted with the actual bound $8\varepsilon^4 n^2 + 2\varepsilon^2$ when the bounded model's norm assumption holds. Thus, one important issue is that, this quantity now being non-negligible, the LHS of (9) can no longer be lower-bounded by the RHS of the inequality (9).

  - Another step, used for establishing the upper bound on $\mathbb{E}[|\mathcal{I}|]$, is the step that upper bounds $\mathbb{P}(\mathcal{E}^c) = \mathcal{O}(1/n^2)$, for the event $\mathcal{E}$ defined on top of Page 19 of [43]. In this case again, in the set of equations before equation (12), it is assumed that the norm of the model is bounded to derive $\|A\hat{\theta}^2\| \leq 144^2 \varepsilon^4$. However, since norm two of $\Theta \hat{W}$ is $\Omega(n^3)$, then these steps are ot valid and hence the analysis does not give $\mathbb{P}(\mathcal{E}^c) = \mathcal{O}(1/n^2)$ anymore.

- Another proof step of [43, Theorem 4.1], used also in the soundness analysis in the proof of [43, Theorem 4.5], relies on upper bounds for the error event $\mathcal{G}^c$, defined on [43, Page 18] as the probability that the correlation between the model output and the held-out samples is significant. These upper bounds, [43, Equations 11] in the proof of [43, Theorem 4.1] and also on [43, 29] in the soundness analysis in the proof of [43, Theorem 4.5], are based on an application of [43, Lemma B.8] and by assuming that the norm two of the model is bounded by 1. These steps again fail if the norm two of the model grows as $\Omega(D/d) = \Omega(n^3)$.

# D   Random subspace training algorithms

The generalization bounds of Theorem 1 and Corollary 1 apply to any arbitrary learning algorithm. In this section, we show how this bound can be applied to random subspace training algorithms. Then, we consider the case where they are trained using an iterative optimization algorithm.

Let $\mathrm{St}(d, D) = \{\Theta \in \mathbb{R}^{D \times d} : \Theta^\top \Theta = \mathrm{I}_d\}$ be the Stiefel manifold, equipped with the uniform distribution $P_\Theta$. Moreover, for a given $\Theta \in \mathbb{R}^{D \times d}$, let $\mathcal{W}_{\Theta,d} \triangleq \{w \in \mathbb{R}^D : \exists w' \in \mathbb{R}^d \text{ s.t. } w = \Theta w'\}$. Random subspace training algorithms first randomly generate an instance of $\Theta$ according to $P_\Theta$, which is kept frozen during training. A random subspace training algorithm $\mathcal{A}^{(d)} \colon \mathcal{Z}^n \times \mathbb{R}^{D \times d} \to \mathcal{W}_{\Theta,d}$ is a learning algorithm that takes the dataset $S$ and the projection matrix $\Theta$ as input, and chooses a model $W \in \mathcal{W}_{\Theta,d}$, by choosing a $W' \in \mathbb{R}^d$.

In other words, $\mathcal{A}^{(d)}(S, \Theta) = \Theta W'$, or alternatively, since $\Theta^\top \Theta = \boldsymbol{I}_d$, $W' = \Theta^\top \mathcal{A}^{(d)}(S, \Theta)$. Hence, using Corollary 1 and by noting Remark 1, we can obtain the following result.

**Corollary 2.** *Consider a random subspace training algorithm and a loss function $\ell \colon \mathcal{Z} \times \mathbb{R}^D \to [0, C]$. Then, for any $\epsilon \in \mathbb{R}$ and the quantization set $\hat{\mathcal{W}} \subseteq \mathbb{R}^d$, we have*

$$\mathrm{gen}(\mu, \mathcal{A}^{(d)}) \leq \inf_{P_{\hat{W}|W',\Theta,S}} \mathbb{E}_{P_\Theta P_{\tilde{\mathbf{S}}}} \left[ \frac{C}{n} \sum_{i \in [n]} \sqrt{2\mathsf{CMI}_i^\Theta(\tilde{\mathbf{S}}, \hat{W})} \right] + \epsilon,$$

*and*

$$\mathrm{gen}(\mu, \mathcal{A}^{(d)}) \leq \inf_{P_{\hat{W}|W',\Theta,S}} \mathbb{E}_{P_\Theta P_{\tilde{\mathbf{S}}} P_{\mathbf{J}_{-i}}} \left[ \frac{C}{n} \sum_{i \in [n]} \sqrt{2\mathsf{CMI}_{i,\mathbf{J}_{-i}}^\Theta(\tilde{\mathbf{S}}, \hat{W})} \right] + \epsilon, \tag{15}$$

*where $\hat{W} \in \hat{\mathcal{W}}$ and the infimum are over all Markov kernels $P_{\hat{W}|W',S,\Theta}$ that satisfies the following distortion criterion:*

$$\mathbb{E}_{P_S P_\Theta P_{W'|S,\Theta} P_{\hat{W}|W',S,\Theta}} \left[ \mathrm{gen}(S, \Theta W') - \mathrm{gen}(S, \Theta \hat{W}) \right] \leq \epsilon. \tag{16}$$

This bound is used in the following subsection, when SGD or SGLD are used for random subspace training. Note that the above bound includes the case of $\hat{W} = W'$ and $\epsilon = 0$, which results in the lossless bounds of $\mathrm{gen}(\mu, \mathcal{A}^{(d)}) \leq \mathbb{E}_{P_{\tilde{\mathbf{S}}} P_\Theta} \left[ \frac{C}{n} \sum_{i \in [n]} \sqrt{2\mathsf{CMI}_i^\Theta(\tilde{\mathbf{S}}, W')} \right]$ and $\mathrm{gen}(\mu, \mathcal{A}^{(d)}) \leq \mathbb{E}_{P_{\tilde{\mathbf{S}}} P_\Theta P_{\mathbf{J}_{-i}}} \left[ \frac{C}{n} \sum_{i \in [n]} \sqrt{2\mathsf{CMI}_{i,\mathbf{J}_{-i}}^\Theta(\tilde{\mathbf{S}}, W')} \right]$.

The results presented in the next section are extensions and improvements in some aspects, upon previous work on bounding the generalization error of SGLD without projection [38, 63–67].

## D.1 Generalization bounds for SGD and SGLD Algorithms

In this section, we consider subspace training algorithms that are trained using an iterative optimization algorithm such as *mini-batch* Stochastic Gradient Descent (SGD) or Stochastic Gradient Langevin dynamics (SGLD).

Let $b \in \mathbb{N}$ bet the mini-batch size, and let

$$V_t \triangleq \{i_{t,1}, \ldots, i_{t,b}\},$$

be the sample indices chosen at time $t \in [T]$, *i.e.,* given $\tilde{\mathbf{S}} \in \mathcal{Z}^{n \times 2}$ and $\mathbf{J} = (J_1, \ldots, J_n)$, the chosen indices at time $t$ are $\tilde{\mathbf{S}}_{V_t, \mathbf{J}} \triangleq \tilde{\mathbf{S}}_{V_t, \mathbf{J}_{V_t}} \triangleq \left\{ Z_{i_{t,1}, J_{i_{t,1}}}, \ldots, Z_{i_{t,b}, J_{i_{t,b}}} \right\}$. Furthermore, denote

$$\widehat{\mathcal{R}}(V_t, W) \triangleq \frac{1}{b} \sum_{i \in V_t} \ell\left(Z_{i, J_i}, W\right).$$

We use also the notation $\mathbf{V} \triangleq (V_1, \ldots, V_T)$ and recall that $\mathbf{J}_{-i} \triangleq \mathbf{J}_{[n] \setminus \{i\}}$.

The considered noisy iterative optimization algorithm consists of the following steps:

- *(Initialization)* Sample $\Theta \in \mathbb{R}^{D \times d}$ and set the initial model's parameters to $W_0 = \Theta W_0'$, where $W_0' \in \mathbb{R}^d$.

- *(Iterate)* For $t \in [T]$, apply the update rule

$$W_t' = \text{Proj} \left\{ W_{t-1}' - \eta_t \nabla_{w'} \widehat{\mathcal{R}}(V_t, \Theta W_{t-1}') + \sigma_t \varepsilon_t \right\}, \tag{17}$$

  with $\eta_t > 0$ (the learning rate), $\sigma_t \geq 0$ (the variance of the Gaussian noise), and $\varepsilon_t \sim \mathcal{N}(\mathbf{0}_d, \mathbf{I}_d)$ (the isotropic Gaussian noise). Here, the projection is an optional operator often used to keep the norm of the model parameters bounded.

- *(Output)* Return the final hypothesis $W_T = \Theta W_T'$.

Note that here, we train on a subspace of dimension $d < D$ defined by $\Theta$ (randomly picked at initialization and fixed during training). Note also that when $\sigma_t = 0$ for all $t \in [T]$, this algorithm reduces to the mini-batch SGD (with projection).

### D.1.1 Mutual information of a mixture of two Gaussians and the component

To state our results, we start by defining two useful functions. Suppose that

$$X = (1 - J)Y_1 + JY_2,$$

where $(J, Y_1, Y_2)$ are independent real-valued random variables defined as follows: $J \sim \text{Bern}(p)$, $Y_1 \sim \mathcal{N}(0, 1)$, and $Y_2 \sim \mathcal{N}(a, 1)$, for some $a \in \mathbb{R}$. Then, it is easy to show that $\mathsf{I}(X; J) = f(a, p)$, where the function $f \colon \mathbb{R} \times [0, 1] \to [0, \log 2]$ is defined as[6]

$$f(a, p) \triangleq h(g_{a,p}(x)) - \log(\sqrt{2\pi e}) = -\mathbb{E}_{g_{a,p}(x)}\left[\log(g_{a,p}(x)] - \log(\sqrt{2\pi e}). \tag{18}$$

Here, $g_{a,p} \colon \mathbb{R} \times [0, 1] \to \mathbb{R}_+$ is defined as a mixture of two scalar Gaussian distributions with probabilities $p$ and $1 - p$:

$$g_{a,p}(x) \triangleq \frac{1}{\sqrt{2\pi}} \left( p e^{-\frac{x^2}{2}} + (1 - p) e^{-\frac{(x-a)^2}{2}} \right). \tag{19}$$

The following lemma, proved in the supplements, establishes some properties of the function $f(a, p)$.

**Lemma 3.** *i) For every $p \in [0, 1]$, $f(0, p) = 0$. ii) For every $p \in [0, 1]$, $f(a, p) = f(-a, p)$ and $f(a, p)$ is an strictly increasing function of $a$ in the range $[0, \infty)$. iii) $\lim_{a \to \infty} f(a, p) = \log(2) h_b(p)$. iv) For every $a \in \mathbb{R}$, $f(a, p) = f(a, 1 - p)$ and for $a \neq 0$, $f(a, p)$ is strictly increasing with respect to $p$ in the range $[0, 1/2]$.*

---

[6] All logarithms are considered to have the base of $e$.

### D.1.2 Lossless generalization bound

We start by stating our bound in its simplest form.

**Theorem 10.** *Suppose that $\ell \in [0, C]$. Then, the generalization error of a random subspace training algorithm, optimized using iterations defined in 17, is upper-bounded as*

$$\text{gen}(\mu, \mathcal{A}^{(d)}) \leq \frac{C\sqrt{2}}{n} \sum_{i \in [n]} \mathbb{E}_{\tilde{\mathbf{S}}, \Theta, \mathbf{V}, \mathbf{J}_{-i}} \left[ \sqrt{\sum_{t\,:\;i \in V_t} \mathbb{E}_{p_{t,i}, \Delta_{t,i}} \left[ f\left( \frac{\eta_t}{b\sigma_t} \Delta_{t,i}, p_{t,i} \right) \right]} \right],$$

*where*

$$\Delta_{t,i} \triangleq \left\| \nabla_{w'} \ell \left( \Theta W'_{t-1}, Z_{i,0} \right) - \nabla_{w'} \ell \left( \Theta W'_{t-1}, Z_{i,1} \right) \right\|,$$

$$p_{t,i} \triangleq \mathbb{P} \left( J_i = 0 \middle| \tilde{\mathbf{S}}, \Theta, \mathbf{V}, \mathbf{J}_{-i}, W'_{t-1}, \{W'_r, W'_{r-1} : r < t, i \in V_r \} \right). \tag{20}$$

This result is proved in Appendix H.2.

In the bound of equation 20, the term $f\left( \frac{\eta_t}{\sigma_t} \Delta_{t,i}, p_{t,i} \right)$ is an increasing function with respect to $\frac{\eta_t}{\sigma_t}$, $\Delta_{t,i}$, and a decreasing function with respect to $|p_{t,i} - 1/2|$. As $t$ increases, the learning algorithm "memorizes" more of the dataset; therefore, $|p_{t,i} - 1/2|$ increases and thus these terms decrease. Furthermore, the learning rate decreases, causing this term to decrease more.

Note that by Lemma 3, $f(\cdot, p)$ is maximized for $p = \frac{1}{2}$. Hence, a simpler upper bound from Theorem 10 can be achieved by replacing $p_{t,i}$ by $\frac{1}{2}$.

### D.1.3 Lossy generalization bound

The bound of Theorem 10 has a clear shortcoming; whenever $\frac{\eta_t}{\sigma_t}$ is very small, the bound becomes loose. In particular, for SGD where $\sigma_t = 0$, the bound becomes vacuous. In this section, to overcome this issue, we consider a lossy version of the above bound. While the lossy bound can be stated without any further assumptions, for a more concrete bound, we make the following assumptions.

**Assumption 11** (Lipschitzness). *The loss function is $\mathfrak{L}$-Lipschitz, i.e., for any $w'_1, w'_2 \in \mathbb{R}^d$, any $z \in \mathcal{Z}$, and any $\Theta \in \text{St}(d, D)$, we have $|\ell\left(z, \Theta w'_1\right) - \ell\left(z, \Theta w'_2\right)| \leq \mathfrak{L}\|w'_1 - w'_2\|$.*

Note that since $\Theta^\top \Theta = \mathrm{I}_d$, then $\|w'_1 - w'_2\| = \|\Theta w'_1 - \Theta w'_2\|$.

**Assumption 12** (Contractivity). *There exists some $\alpha \in \mathbb{R}^+$, such that for any $w'_1, w'_2 \in \mathcal{W}'$, $z \in \mathcal{Z}$, and $\Theta \in \text{St}(d, D)$, we have*

$$\left\| \left( w'_1 - \eta \nabla_{w'} \ell(z, \Theta w'_1) \right) - \left( w'_2 - \eta \nabla_{w'} \ell(z, \Theta w'_2) \right) \right\| \leq \alpha \left\| w'_1 - w'_2 \right\|.$$

*Whenever $\alpha < 1$, we say the projected SGLD is $\alpha$-contractive.*

Similar assumptions have been used in previous works, such as [68]. In fact, the contractivity property of SGD has been theoretically proved under certain conditions, such as when the loss function is smooth and strongly convex [68–70].

In addition to being sensitive to cases where $\frac{\eta_t}{\sigma_t}$ is very small, the bound of Theorem 10 does not account for the "forgetting" effect of the iterative optimization algorithms: the information obtained by $W'_T$ about $J_i$ in the initial iterations will eventually fade out, as $T$ increases. To account for this effect, similar to [66, 67], we assume that $\mathcal{W}' = \mathcal{B}_D(R)$,[7] for some $R \in \mathbb{R}_+$.

**Theorem 13.** *Suppose that $\ell \in [0, C]$, $\mathcal{W}' = \mathcal{B}_D(R)$, for some $R \in R_+$, and Assumptions 11 and 12 hold with constants $\mathfrak{L} \in \mathbb{R}_+$ and $\alpha \leq 1$, respectively. Then, for any set of $\{\nu_t\}_{t \in [T]}$, such that $\nu_t \in \mathbb{R}_+$, the generalization error of a random subspace training algorithm, optimized using iterations defined in 17, is upper bounded as*

$$\text{gen}(\mu, \mathcal{A}^{(d)}) \leq \frac{C\sqrt{2}}{n} \sum_{i \in [n]} \mathbb{E}_{\tilde{\mathbf{S}}, \Theta, \mathbf{V}, \mathbf{J}_{-i}} \left[ \sqrt{\sum_{t\,:\;i \in V_t} A_{t,i} \mathbb{E}_{\hat{p}_{t,i}, \hat{\Delta}_{t,i}} \left[ f\left( \frac{\eta_t}{b\hat{\sigma}_t} \hat{\Delta}_{t,i}, \hat{p}_{t,i} \right) \right]} \right]$$

$$+ \frac{2\sqrt{2}\mathfrak{L}\,\Gamma\left(\frac{d+1}{2}\right)}{\Gamma\left(\frac{d}{2}\right)} \sum_{t \in [T]} \nu_t \alpha^{T-t}, \tag{21}$$

---

[7]In this setup, for $w' \in \mathcal{W}'$, $\text{Proj}\{w'\} = w'$ and otherwise $\text{Proj}\{w'\} = \frac{R}{\|w'\|} w'$.

*where*

$$\hat{\Delta}_{t,i} \triangleq \left\| \nabla_{w'} \ell \left( \Theta \hat{W}_{t-1}, Z_{i,0} \right) - \nabla_{w'} \ell \left( \Theta \hat{W}_{t-1}, Z_{i,1} \right) \right\|,$$

$$\hat{p}_{t,i} \triangleq \mathbb{P}\left( J_i = 0 \big| \tilde{\mathbf{S}}, \Theta, \mathbf{V}, \mathbf{J}_{-i}, \hat{W}_{t-1} \right),$$

$$\hat{\sigma}_t \triangleq \sqrt{\sigma_t^2 + \nu_t^2},$$

$$q_t \triangleq 1 - 2\Phi\left( \frac{R + \eta_t \mathfrak{L}}{\hat{\sigma}_t} \right),$$

$$A_{t,i} \triangleq \prod_{r \in [t+1:T]: \ i \notin V_r} q_r,$$

*where $\hat{W}_t$ are random variables that satisfy*

$$\left\| \hat{W}_t - W_t' \right\| \leq \sum_{r \in [t]} \alpha^{t-r} \nu_r \left\| \varepsilon_r' \right\|,$$

*for $\varepsilon_t' \sim \mathcal{N}(\mathbf{0}_d, \mathbf{I}_d)$, which is an auxiliary additional noise, independent of all other random variables, and where $\Phi(x) \triangleq \int_x^\infty \frac{1}{\sqrt{2\pi}} \exp(-y^2/2) \mathrm{d}y$ is the Gaussian complementary cumulative distribution function (CCDF).*

This theorem is proved in Appendix H.3. Here, we discuss some remarks.

First, the "gained" information from the initial iterations fades as $T \to \infty$, when $q_t < 1$ (note that always $q_t \leq 1$).

Second, we note that, unlike in Theorem 10, where $p_{t,i}$ depends on all past iterations in which sample $i$ is used, in this theorem, $\hat{p}_{t,i}$ depends only on the immediate past iteration. It can be shown that a similar result can be achieved for Theorem 13 *i.e.,* allowing $\hat{p}_{t,i}$ to depend on all past iterations, at the expense of replacing all $\{q_t\}_t$ by 1.

Third, it can be observed that if $\forall t \in [T]: \nu_t = 0$, we recover Theorem 10, except for the definition of $p_{t,i}$, that can be adjusted at the expense of replacing all $\{q_t\}_t$ by 1, as explained above. Furthermore, by increasing $\nu_t$, the second term in equation 21, i.e. the "distortion" term, increases; but the first "rate" term decreases since $f\left( \frac{\eta_t}{\sqrt{\sigma_t^2 + \nu_t^2}} \hat{\Delta}_{t,i}, \hat{p}_{t,i} \right)$ decreases. Therefore, in general, the lossy bound can outperform the lossless bound. In particular, for SGD, *i.e.,* when $\sigma_t = 0$, the lossless bound and previous works (for the case of no projection) [38, 63–67] become vacuous, while the lossy bound does not.

Lastly, to achieve this bound, we considered a sequence of parallel "perturbed" iterations. In each of these auxiliary iterations, we introduced an additional independent noise $\nu_t \varepsilon_t'$, where $\varepsilon_t' \sim \mathcal{N}(\mathbf{0}_d, \mathbf{I}_d)$. It can be seen that for the contractive SGD/SGLD, the effect of added perturbation in the initial iterations vanishes as $T \to \infty$. Therefore, once again, it can be seen that the effect of the increase in mutual information from the initial iterations eventually fades.

## E   Proof of Theorem 1

We prove the theorem in its most general form stated in Remark 1. This means that we assume that the learning algorithm $\mathcal{A}$ is also aware of the projection matrix $\Theta$, *i.e.,* $\mathcal{A}: \mathcal{Z}^n \times \mathbb{R}^{D \times d} \to \mathcal{W}$ takes both the dataset $S_n$ and the projection matrix $\Theta$ as input to learn $W$. Moreover, we allow the quantization step to depend on $S$, $\Theta$, and $\mathcal{A}(S_n, \Theta)$. In this general case, $\hat{W} = \hat{\mathcal{A}}(S, \Theta) = \tilde{\mathcal{A}}(\Theta, S_n, \mathcal{A}(S, \Theta))$. We denote this general compressed algorithm by $P_{\hat{W}|S_n, W, \Theta}$. Note that $P_{\hat{W}|\Theta^\top W}$ is a special case of this more general setup.

Fix some $\epsilon \in \mathbb{R}$ and the quantization set $\hat{\mathcal{W}}$. Consider any Markov kernel $P_{\hat{W}|S_n, W, \Theta}$ and $P_\Theta$ that satisfy the following distortion criterion:

$$\mathbb{E}_{P_{S_n} P_\Theta P_{W, \hat{W}|S_n, \Theta}} \left[ \mathrm{gen}(S_n, W) - \mathrm{gen}(S_n, \Theta\hat{W}) \right] \leq \epsilon.$$

Using this condition, it is sufficient to show that

$$\mathrm{gen}(\mu, \hat{\mathcal{A}}) = \mathbb{E}_{P_{S_n} P_\Theta P_{W, \hat{W}|S_n, \Theta}} \left[ \mathrm{gen}(S_n, \Theta\hat{W}) \right] \leq \mathbb{E}_{P_{\tilde{\mathbf{S}}} P_\Theta} \left[ \sqrt{\frac{2\Delta\ell_{\hat{w}}(\tilde{\mathbf{S}}, \Theta)}{n} \mathsf{CMI}^\Theta(\tilde{\mathbf{S}}, \hat{\mathcal{A}})} \right],$$

where

$$\Delta\ell_{\hat{w}}(\tilde{\mathbf{S}}, \Theta) := \mathbb{E}_{P_{\hat{W}|\tilde{\mathbf{S}}, \Theta}} \left[ \frac{1}{n} \sum_{i \in [n]} (\ell(Z_{i,0}, \Theta\hat{W}) - \ell(Z_{i,1}, \Theta\hat{W}))^2 \right].$$

Denote the marginal distribution of $(S_n, \Theta, \hat{W})$ under $P_{S_n} P_{\Theta} P_{W, \hat{W}|S_n, \Theta}$ by $P_{S_n, \Theta, \hat{W}}$ and conditional distribution of $\hat{W}$ given $(S_n, \Theta)$ by $P_{\hat{W}|S_n, \Theta}$. Hence, $P_{S_n, \Theta, \hat{W}} = P_{S_n} P_{\Theta} P_{\hat{W}|S_n, \Theta}$ and

$$\begin{aligned}
\mathrm{gen}(\mu, \hat{\mathcal{A}}) &= \mathbb{E}_{P_{S_n} P_{\Theta} P_{\hat{W}|S_n, \Theta}} \left[ \mathrm{gen}(S_n, \Theta\hat{W}) \right] \\
&= \mathbb{E}_{P_{\tilde{\mathbf{S}}} P_{\Theta} P_{\mathbf{J}} P_{\hat{W}|\tilde{\mathbf{S}}_{\mathbf{J}}, \Theta}} \left[ \widehat{\mathcal{R}}(\tilde{\mathbf{S}}_{\mathbf{J}^c}, \Theta\hat{W}) - \widehat{\mathcal{R}}(\tilde{\mathbf{S}}_{\mathbf{J}}, \Theta\hat{W}) \right].
\end{aligned}$$

It is hence sufficient to show that for any $\tilde{\mathbf{S}}$ and $\Theta$,

$$\mathbb{E}_{P_{\mathbf{J}} P_{\hat{W}|\tilde{\mathbf{S}}_{\mathbf{J}}, \Theta}} \left[ \widehat{\mathcal{R}}(\tilde{\mathbf{S}}_{\mathbf{J}^c}, \Theta\hat{W}) - \widehat{\mathcal{R}}(\tilde{\mathbf{S}}_{\mathbf{J}}, \Theta\hat{W}) \right] \leq \sqrt{\frac{2\Delta\ell_{\hat{w}}(\tilde{\mathbf{S}}, \Theta)}{n} \mathsf{CMI}^{\Theta}(\tilde{\mathbf{S}}, \hat{\mathcal{A}})}.$$

Denote $P_{\hat{W}|\tilde{\mathbf{S}}, \Theta} \triangleq \mathbb{E}_{P_{\mathbf{J}}} \left[ P_{\hat{W}|\tilde{\mathbf{S}}_{\mathbf{J}}, \Theta} \right]$ and $P_{\mathbf{J}, \hat{W}|\tilde{\mathbf{S}}, \Theta} \triangleq P_{\mathbf{J}} P_{\hat{W}|\tilde{\mathbf{S}}_{\mathbf{J}}, \Theta} \triangleq P_{\mathbf{J}|\tilde{\mathbf{S}}, \Theta, \hat{W}} P_{\hat{W}|\tilde{\mathbf{S}}, \Theta}$ be the conditional distributions of $(\mathbf{J}, \hat{W})$ given $(\tilde{\mathbf{S}}, \Theta)$. Note that the marginal distribution of $\mathbf{J}$ under $P_{\mathbf{J}, \hat{W}|\tilde{\mathbf{S}}, \Theta}$ is $P_{\mathbf{J}}$, i.e.,

$$\mathbb{E}_{P_{\hat{W}|\tilde{\mathbf{S}}, \Theta}} \left[ P_{\mathbf{J}|\tilde{\mathbf{S}}, \Theta, \hat{W}} \right] = P_{\mathbf{J}}.$$

Now, fix some $\lambda \neq 0$ that will be determined later. We have

$$\begin{aligned}
\mathbb{E}_{P_{\mathbf{J}|\tilde{\mathbf{S}}, \hat{W}, \Theta}} &\left[ \widehat{\mathcal{R}}(\tilde{\mathbf{S}}_{\mathbf{J}^c}, \Theta\hat{W}) - \widehat{\mathcal{R}}(\tilde{\mathbf{S}}_{\mathbf{J}}, \Theta\hat{W}) \right] \\
&\overset{(a)}{\leq} \frac{1}{\lambda} D_{KL} \left( P_{\mathbf{J}|\tilde{\mathbf{S}}, \hat{W}, \Theta} \big\| P_{\mathbf{J}} \right) + \frac{1}{\lambda} \log \left( \mathbb{E}_{P_{\mathbf{J}}} \left[ e^{\lambda(\widehat{\mathcal{R}}(\tilde{\mathbf{S}}_{\mathbf{J}^c}, \Theta\hat{W}) - \widehat{\mathcal{R}}(\tilde{\mathbf{S}}_{\mathbf{J}}, \Theta\hat{W}))} \right] \right) \\
&= \frac{1}{\lambda} D_{KL} \left( P_{\mathbf{J}|\tilde{\mathbf{S}}, \hat{W}, \Theta} \big\| P_{\mathbf{J}} \right) + \frac{1}{\lambda} \log \left( \mathbb{E}_{P_{\mathbf{J}}} \left[ e^{\frac{\lambda}{n} \sum_{i \in [n]} (-1)^{J_i} (\ell(Z_{i,0}, \Theta\hat{W}) - \ell(Z_{i,1}, \Theta\hat{W}))} \right] \right) \\
&\overset{(b)}{\leq} \frac{1}{\lambda} D_{KL} \left( P_{\mathbf{J}|\tilde{\mathbf{S}}, \hat{W}, \Theta} \big\| P_{\mathbf{J}} \right) + \frac{1}{\lambda} \sum_{i \in [n]} \frac{\lambda^2 (\ell(Z_{i,0}, \Theta\hat{W}) - \ell(Z_{i,1}, \Theta\hat{W}))^2}{2n^2}.
\end{aligned}$$

where $(a)$ follows from Donsker-Varadhan's inequality and $(b)$ by the inequality $\frac{1}{2}(e^{-x} + e^x) \leq e^{x^2/2}$.

Hence,

$$\begin{aligned}
\mathbb{E}_{P_{\mathbf{J}} P_{\hat{W}|\tilde{\mathbf{S}}_{\mathbf{J}}, \Theta}} &\left[ \widehat{\mathcal{R}}(\tilde{\mathbf{S}}_{\mathbf{J}^c}, \Theta\hat{W}) - \widehat{\mathcal{R}}(\tilde{\mathbf{S}}_{\mathbf{J}}, \Theta\hat{W}) \right] \\
&= \mathbb{E}_{P_{\hat{W}|\tilde{\mathbf{S}}, \Theta} P_{\mathbf{J}|\tilde{\mathbf{S}}, \hat{W}, \Theta}} \left[ \widehat{\mathcal{R}}(\tilde{\mathbf{S}}_{\mathbf{J}^c}, \Theta\hat{W}) - \widehat{\mathcal{R}}(\tilde{\mathbf{S}}_{\mathbf{J}}, \Theta\hat{W}) \right] \\
&\leq \frac{1}{\lambda} D_{KL} \left( P_{\hat{W}|\tilde{\mathbf{S}}, \Theta} P_{\mathbf{J}|\tilde{\mathbf{S}}, \hat{W}, \Theta} \big\| P_{\mathbf{J}} P_{\hat{W}|\tilde{\mathbf{S}}, \Theta} \right) \\
&\quad + \frac{\lambda}{2n} \mathbb{E}_{P_{\hat{W}|\tilde{\mathbf{S}}, \Theta}} \left[ \frac{1}{n} \sum_{i \in [n]} (\ell(Z_{i,0}, \Theta\hat{W}) - \ell(Z_{i,1}, \Theta\hat{W}))^2 \right] \\
&= \frac{1}{\lambda} D_{KL} \left( P_{\hat{W}|\tilde{\mathbf{S}}, \Theta} P_{\mathbf{J}|\tilde{\mathbf{S}}, \hat{W}, \Theta} \big\| P_{\mathbf{J}} P_{\hat{W}|\tilde{\mathbf{S}}, \Theta} \right) + \frac{\lambda \Delta\ell_{\hat{w}}(\tilde{\mathbf{S}}, \Theta)}{2n} \\
&= \frac{1}{\lambda} D_{KL} \left( P_{\mathbf{J}} P_{\hat{W}|\tilde{\mathbf{S}}_{\mathbf{J}}, \Theta} \big\| P_{\mathbf{J}} P_{\hat{W}|\tilde{\mathbf{S}}, \Theta} \right) + \frac{\lambda \Delta\ell_{\hat{w}}(\tilde{\mathbf{S}}, \Theta)}{2n} \\
&= \frac{1}{\lambda} \mathsf{CMI}^{\Theta}(\tilde{\mathbf{S}}, \hat{\mathcal{A}}) + \frac{\lambda \Delta\ell_{\hat{w}}(\tilde{\mathbf{S}}, \Theta)}{2n} \\
&\leq \sqrt{\frac{2\Delta\ell_{\hat{w}}(\tilde{\mathbf{S}}, \Theta) \mathsf{CMI}^{\Theta}(\tilde{\mathbf{S}}, \hat{\mathcal{A}})}{n}},
\end{aligned}$$

where the last step is followed by letting

$$\lambda \triangleq \sqrt{\frac{2n \mathsf{CMI}^{\Theta}(\tilde{\mathbf{S}}, \hat{\mathcal{A}})}{\Delta\ell_{\hat{w}}(\tilde{\mathbf{S}}, \Theta)}}.$$

This completes the proof.

# F Proofs of Section 4 and Appendix B: Application to raised limitations of CMI bounds

For the proofs of Section 4 and Appendix B, we always consider the normalized setup, *i.e.*, $R = 1$, $L = 1$ (for Theorem 3), $L_c = 1$ (for Proposition 1), and $B = 1$ (for Theorem 7). The proof applies for arbitrary values of $(R, L, L_c, B)$, by simply scaling the constants.

All proofs are based on Theorem 1, with a particular class of choices of $P_\Theta$ and $P_{\hat{W}|\Theta^\top W}$, called the choices from the scheme JL$(d, c_w, \nu)$ for some $d \in \mathbb{N}$, $c_w \in \left[1, \sqrt{5/4}\right)$, and $\nu \in (0, 1]$, described in Appendix F.1. For a given JL$(d, c_w, \nu)$, we then use Theorem 1 for some suitable $\epsilon \in \mathbb{R}$:

$$\mathrm{gen}(\mu, \mathcal{A}) \leq \mathbb{E}_{P_{\tilde{\mathbf{S}}} P_\Theta} \left[ \sqrt{\frac{2\Delta\ell_{\hat{w}}(\tilde{\mathbf{S}}, \Theta)}{n} \mathsf{CMI}^\Theta(\tilde{\mathbf{S}}, \hat{\mathcal{A}})} \right] + \epsilon. \tag{22}$$

Recall that the term $\Delta\ell_{\hat{w}}(\tilde{\mathbf{S}}, \Theta)$ is defined as

$$\Delta\ell_{\hat{w}}(\tilde{\mathbf{S}}, \Theta) := \mathbb{E}_{P_{W|\tilde{\mathbf{S}}} P_{\hat{W}|\Theta^\top W}} \left[ \frac{1}{n} \sum_{i \in [n]} (\ell(Z_{i,0}, \Theta\hat{W}) - \ell(Z_{i,1}, \Theta\hat{W}))^2 \right],$$

and the choices of $P_\Theta$ and $P_{\hat{W}|\Theta^\top W}$ should satisfy the distortion criterion

$$\mathbb{E}_{P_{S_n, W} P_\Theta P_{\hat{W}|\Theta^\top W}} \left[ \mathrm{gen}(S_n, W) - \mathrm{gen}(S_n, \Theta\hat{W}) \right] \leq \epsilon. \tag{23}$$

For brevity, we often use the notation

$$\Delta(W, \Theta\hat{W}; S_n) := \mathrm{gen}(S_n, W) - \mathrm{gen}(S_n, \Theta\hat{W}).$$

Furthermore, denote the $D$-dimensional ball of radius $\nu \in \mathbb{R}_+$ and center $w \in \mathbb{R}^D$ by $\mathcal{B}_D(w, \nu)$. If $w = \mathbf{0}_D$, for simplicity we write $\mathcal{B}_D(\mathbf{0}_D, \nu) \equiv \mathcal{B}_D(\nu)$, where $\mathbf{0}_D$ designates the all-zero vector in $\mathbb{R}^D$.

## F.1 Johnson-Lindenstrauss projection scheme

Fix some constant $c_w \in \left[1, \sqrt{\frac{5}{4}}\right)$ and $\nu \in (0, 1]$. Let $d \in \mathbb{N}^*$ and $\Theta$ be a matrix of size $D \times d$ whose elements are i.i.d. samples from $\mathcal{N}(0, 1/d)$. For a given $\Theta$ and $W = \mathcal{A}(S_n)$, in the scheme JL$(d, c_w, \nu)$, let

$$U := \begin{cases} \Theta^\top W, & \text{if } \|\Theta^\top W\| \leq c_w, \\ \mathbf{0}_d, & \text{otherwise.} \end{cases} \tag{24}$$

Let $V_\nu$ be a random variable that takes value uniformly over $\mathcal{B}_d(\nu)$. Let $\hat{W} \in \hat{\mathcal{W}} = \mathcal{B}_d(c_w + \nu)$ be defined as

$$\hat{W} = U + V_\nu. \tag{25}$$

This means that $\hat{W}$ is a random variable that takes value uniformly over $\mathcal{B}_d(U, \nu)$:

$$\hat{W} \sim \mathrm{Unif}(\mathcal{B}_d(U, \nu)).$$

In other words, we define $\hat{W}$ as a quantization of $W' = \Theta^\top W$ obtained as follows: if $\|\Theta^\top W\| \leq c_w$, then $\hat{W}$ is uniformly sampled from $\mathcal{B}_d(\Theta^\top W, \nu)$; otherwise, $\hat{W}$ is uniformly sampled from $\mathcal{B}_d(\nu)$. Such quantization has been previously used in [22] to establish a generalization bound for the distributed SVM learning algorithm.

**Disintegrated CMI bound:** The disintegrated CMI bound $\mathsf{CMI}^\Theta(\tilde{\mathbf{S}}, \hat{\mathcal{A}})$ in the scheme JL$(d, c_w, \nu)$ can be upper bounded as

$$\begin{aligned} \mathsf{CMI}^\Theta(\tilde{\mathbf{S}}, \hat{\mathcal{A}}) &= h^{\tilde{\mathbf{S}}, \Theta}(\hat{W}) - h^{\tilde{\mathbf{S}}, \Theta}(\hat{W}|\mathbf{J}) \\ &\overset{(a)}{\leq} h^{\tilde{\mathbf{S}}, \Theta}(\hat{W}) - h^{\tilde{\mathbf{S}}, \Theta}(\hat{W}|\mathbf{J}, W) \\ &\overset{(b)}{=} h^{\tilde{\mathbf{S}}, \Theta}(\hat{W}) - h(\hat{W}|\Theta^\top W) \\ &\overset{(c)}{\leq} \log(\mathrm{Volume}(\mathcal{B}_d(c_w + \nu))) - \log(\mathrm{Volume}(\mathcal{B}_d(\nu))) \\ &= d \log\left(\frac{c_w + \nu}{\nu}\right), \end{aligned} \tag{26}$$

where

- $h^{\tilde{\mathbf{S}},\Theta}(\hat{W})$ is the differential entropy of $\hat{W} \sim P_{\hat{W}|\tilde{\mathbf{S}},\Theta}$, $h^{\tilde{\mathbf{S}},\Theta}(\hat{W}|\mathbf{J}) = \mathbb{E}_{\mathbf{J}}\left[h^{\tilde{\mathbf{S}},\Theta,\mathbf{J}}(\hat{W})\right]$, and $h^{\tilde{\mathbf{S}},\Theta,\mathbf{J}}(\hat{W})$ is the differential entropy of $\hat{W} \sim P_{\hat{W}|\tilde{\mathbf{S}},\Theta,\mathbf{J}}$,

- $(a)$ follows from the fact that conditioning does not increase the entropy,

- $(b)$ yields due to Markov chain $\hat{W} - \Theta^{\top}W - (\tilde{\mathbf{S}}, \Theta, \mathbf{J}, W)$,

- and $(c)$ holds since i) $\|\hat{W}\| \leq c_w + \nu$ by construction and hence $h^{\tilde{\mathbf{S}},\Theta}(\hat{W})$ is upper bounded by the differential entropy of a random variable taking value uniformly over $\mathcal{B}_d(c_w + \nu)$, and ii) since given $\Theta^{\top}W$, $\hat{W}$ is chosen uniformly over a $d$-dimensional ball either around $\mathbf{0}_d$ or $\Theta^{\top}W$, depending on $\|\Theta^{\top}W\|$.

## F.2 Proof of Theorem 3

As explained in Appendix F, we consider the case $L = R = 1$, and use Theorem 1 using the JL$(d, c_w, \nu)$ transformation described in Appendix F.1, with some $d \in \mathbb{N}^+$, $c_w \in \left[1, \sqrt{5/4}\right)$, and $\nu \in (0, 1]$. To do so, we start by bounding $\mathsf{CMI}^{\Theta}(\tilde{\mathbf{S}}, \hat{\mathcal{A}})$, the distortion equation 23, and $\mathbb{E}_{\tilde{\mathbf{S}},\Theta}[\Delta \ell_{\hat{w}}(\tilde{\mathbf{S}}, \Theta)]$.

**Bound on the disintegrated CMI:** It is shown in equation 26 that

$$\mathsf{CMI}^{\Theta}(\tilde{\mathbf{S}}, \hat{\mathcal{A}}) = \leq d \log\left(\frac{c_w + \nu}{\nu}\right). \tag{27}$$

**Bound on the distortion:** Next, we bound the distortion term. By definition, and using the linearity of expectation, we obtain

$$\Delta(W, \Theta\hat{W}; S_n) = \mathrm{gen}(S_n, W) - \mathrm{gen}(S_n, \Theta\hat{W})$$

$$= \mathbb{E}_{Z \sim \mu}[-\langle W, Z\rangle] + \frac{1}{n}\sum_{i=1}^{n}\langle W, Z_i\rangle + \mathbb{E}_{Z \sim \mu}[\langle\Theta\hat{W}, Z\rangle] - \frac{1}{n}\sum_{i=1}^{n}\langle\Theta\hat{W}, Z_i\rangle$$

$$= -\langle W, \mathbb{E}_{Z \sim \mu}[Z] - \frac{1}{n}\sum_{i=1}^{n}Z_i\rangle + \langle\Theta\hat{W}, \mathbb{E}_{Z \sim \mu}[Z] - \frac{1}{n}\sum_{i=1}^{n}Z_i\rangle$$

$$= -\langle W, \bar{Z}\rangle + \langle\hat{W}, \Theta^{\top}\bar{Z}\rangle, \tag{28}$$

where $\bar{Z} \triangleq \mathbb{E}_{Z \sim \mu}[Z] - \frac{1}{n}\sum_{i=1}^{n}Z_i$.

Additionally, since for any $(x, y) \in \mathbb{R}^D \times \mathbb{R}^D$, $\mathbb{E}_{\Theta}[\langle\Theta^{\top}x, \Theta^{\top}y\rangle] = \langle x, y\rangle$, then

$$\mathbb{E}_{\hat{W},\Theta,W,S_n}[\Delta(W, \Theta\hat{W}; S_n)] = \mathbb{E}_{\hat{W},\Theta,W,S_n}[-\langle W, \bar{Z}\rangle + \langle\hat{W}, \Theta^{\top}\bar{Z}\rangle]$$

$$= \mathbb{E}_{\hat{W},\Theta,W,S_n}[-\langle\Theta^{\top}W, \Theta^{\top}\bar{Z}\rangle + \langle\hat{W}, \Theta^{\top}\bar{Z}\rangle]$$

$$= \mathbb{E}_{\hat{W},\Theta,W,S_n}[\langle\hat{W} - \Theta^{\top}W, \Theta^{\top}\bar{Z}\rangle].$$

Let $\mathcal{E}$ be the event that $\|\Theta^{\top}W\| > c_w$ and denote by $\mathcal{E}^c$ the complementary event of $\mathcal{E}$. By the law of total expectation,

$$\mathbb{E}_{\hat{W},\Theta,W,S_n}[\Delta(W, \Theta\hat{W}; S_n)] = \mathbb{E}[\langle\hat{W} - \Theta^{\top}W, \Theta^{\top}\bar{Z}\rangle \mid \mathcal{E}]\mathbb{P}(\mathcal{E}) + \mathbb{E}[\langle\hat{W} - \Theta^{\top}W, \Theta^{\top}\bar{Z}\rangle \mid \mathcal{E}^c]\mathbb{P}(\mathcal{E}^c). \tag{29}$$

By definition of $\hat{W}$, $\mathbb{E}[\hat{W}] = 0$ under $\mathcal{E}$, $\mathbb{E}[\hat{W}] = \Theta^{\top}W$ otherwise. Therefore, equation 29 can be simplified as

$$\mathbb{E}_{\hat{W},\Theta,W,S_n}[\Delta(W, \Theta\hat{W}; S_n)] = \mathbb{E}[-\langle\Theta^{\top}W, \Theta^{\top}\bar{Z}\rangle \mid \mathcal{E}]\mathbb{P}(\mathcal{E})$$

$$= \mathbb{E}[-\langle\Theta^{\top}W, \Theta^{\top}\bar{Z}\rangle\mathbb{1}\{\mathcal{E}\}]$$

$$\leq \mathbb{E}[\|\Theta^{\top}W\|\|\Theta^{\top}\bar{Z}\|\mathbb{1}\{\mathcal{E}\}] \tag{30}$$

$$\leq \mathbb{E}[\|\Theta^{\top}\bar{Z}\|^2]^{1/2}\,\mathbb{E}[\|\Theta^{\top}W\|^4]^{1/4}\,\mathbb{E}[\mathbb{1}\{\mathcal{E}\}]^{1/4}, \tag{31}$$

where equation 30 follows from Cauchy-Schwarz inequality, and equation 31 results from Hölder's inequality.

Now, we bound each of the terms $\mathbb{E}[\|\Theta^{\top}\bar{Z}\|^2]$, $\mathbb{E}[\|\Theta^{\top}W\|^4]$, and $\mathbb{E}[\mathbb{1}\{\mathcal{E}\}]$.

- Since the elements of $\Theta \in \mathbb{R}^{D \times d}$ are i.i.d. from $\mathcal{N}(0, 1/d)$, then for any fixed vector $x \in \mathbb{R}^D$, each entry of $\frac{\sqrt{d}\Theta^\top x}{\|x\|}$ is an independent random variable distributed according to $\mathcal{N}(0,1)$. Hence, $V_x = \left\| \frac{\sqrt{d}\Theta^\top x}{\|x\|} \right\|^2$ is a chi-squared random variable with $d$-degrees of freedom, and we have

$$\mathbb{E}[V_x] = d.$$

This concludes that for any $\bar{z}$,

$$\mathbb{E}\left[\|\Theta^\top \bar{z}\|^2\right] = \|\bar{z}\|^2.$$

- Moreover, since $V_x$ is a chi-squared distribution with $d$-degrees of freedom, we have that

$$\mathbb{E}[V_x^2] = \mathbb{E}[V_x]^2 + \mathbb{E}[(V_x - \mathbb{E}[V_x])^2] = d^2 + 2d.$$

Hence for every $w \in \mathcal{W}$,

$$\begin{aligned}
\mathbb{E}_\Theta\left[\|\Theta^\top w\|^4\right] &= \frac{\|w\|^4}{d^2} \mathbb{E}_\Theta\left[\left\|\frac{\sqrt{d}\Theta^\top w}{\|w\|}\right\|^4\right] \\
&= \frac{\|w\|^4}{d^2} \mathbb{E}_\Theta\left[V_w^2\right] \\
&= \left(1 + \frac{2}{d}\right)\|w\|^4.
\end{aligned}$$

- By [71, Lemma 9], for any $w \in \mathcal{B}_D(1)$, if $c_w \in [1, \frac{\sqrt{5}}{2})$,

$$\mathbb{P}(\mathcal{E}) \le e^{-0.21 d(c_w^2 - 1)^2}. \tag{32}$$

More precisely by [71, Lemma 9] we have for any $t \in [0, 1/4]$ and any $w \in \mathcal{B}_d(1)$,

$$\mathbb{P}\left(\|\Theta^\top w\|^2 - \|w\|^2 > t\|w\|^2\right) \le e^{-0.21 dt^2}.$$

We note that this inequality is a "single-sided" tail bound version of [71, Lemma 9] (while therein stated as a "double-sided" tail bound). This explains why RHS of the inequality in [71, Lemma 9] is $2e^{-0.21 dt^2}$, while here we have $e^{-0.21 dt^2}$.

Next, note that $(t+1)\|w\|^2 \le (t+1)$, hence

$$\mathbb{P}\left(\|\Theta^\top w\|^2 - \|w\|^2 > t\|w\|^2\right) = \mathbb{P}\left(\|\Theta^\top w\|^2 > (t+1)\|w\|^2\right) \ge \mathbb{P}\left(\|\Theta^\top w\|^2 > t+1\right).$$

Thus, by letting $t = c_w^2 - 1$ for $c_w \in [1, \sqrt{5/4})$, we have

$$\mathbb{P}\left(\|\Theta^\top w\| \ge c_w\right) \le e^{-0.21 dt^2} = e^{-0.21 d(c_w^2 - 1)^2}.$$

Combining the above upper bounds on $\mathbb{E}[\|\Theta^\top \bar{Z}\|^2]$, $\mathbb{E}[\|\Theta^\top W\|^4]$, and $\mathbb{E}[\mathbb{1}\{\mathcal{E}\}]$, we obtain,

$$\begin{aligned}
\mathbb{E}_{\hat{W}, \Theta, W, S_n}[\Delta(W, \Theta\hat{W}; S_n)] &\le \mathbb{E}[\|\bar{Z}\|^2]^{1/2} \, \mathbb{E}[\|\Theta^\top W\|^4]^{1/4} \, e^{-\frac{0.21}{4} d(c_w^2 - 1)^2} \\
&\le \mathbb{E}[\|\bar{Z}\|^2]^{1/2} \, \mathbb{E}_W[\|W\|^2 + \frac{2}{d}\|W\|^4]^{1/4} \, e^{-\frac{0.21}{4} d(c_w^2 - 1)^2} \\
&\le \mathbb{E}[\|\bar{Z}\|^2]^{1/2} \left(1 + \frac{2}{d}\right)^{1/4} e^{-\frac{0.21}{4} d(c_w^2 - 1)^2}, \tag{33}
\end{aligned}$$

where equation 33 follows from assuming that $\mathcal{W} \subseteq \mathcal{B}_D(1)$.

It remains then to upper bound $\mathbb{E}[\|\bar{Z}\|^2]$. By definition of $\bar{Z}$ and the linearity of expectation, we have

$$
\begin{aligned}
\mathbb{E}[\|\bar{Z}\|^2] &= \mathbb{E}\big[\|\mathbb{E}[Z] - \frac{1}{n}\sum_{i=1}^{n} Z_i\|^2\big] \\
&= \mathbb{E}[\|\frac{1}{n}\sum_{i=1}^{n}(\mathbb{E}[Z] - Z_i)\|^2] \\
&= \frac{1}{n^2}\mathbb{E}\left[\left(\sum_{i=1}^{n}(\mathbb{E}[Z] - Z_i)\right)^\top \left(\sum_{j=1}^{n}(\mathbb{E}[Z] - Z_j)\right)\right] \\
&= \frac{1}{n^2}\mathbb{E}\left[\sum_{i=1}^{n}\sum_{j=1}^{n}(\mathbb{E}[Z] - Z_i)^\top(\mathbb{E}[Z] - Z_j)\right] \\
&= \frac{1}{n^2}\mathbb{E}\left[\sum_{i=1}^{n}(\mathbb{E}[Z] - Z_i)^\top(\mathbb{E}[Z] - Z_i) + \sum_{i\neq j}(\mathbb{E}[Z] - Z_i)^\top(\mathbb{E}[Z] - Z_j)\right] \\
&= \frac{1}{n^2}\mathbb{E}\left[\sum_{i=1}^{n}\|\mathbb{E}[Z] - Z_i\|^2 + \sum_{i\neq j}\mathrm{Cov}(Z_i, Z_j)\right] \\
&= \frac{1}{n^2}\mathbb{E}\left[\sum_{i=1}^{n}\|\mathbb{E}[Z] - Z_i\|^2\right] &(34) \\
&\leq \frac{4}{n}, &(35)
\end{aligned}
$$

where equation 34 results from $\mathrm{Cov}(Z_i, Z_j) = 0$ for $i \neq j$ since $Z_i, Z_j$ are independent, and equation 35 follows from $\mathcal{Z} \subseteq \mathcal{B}_D(1)$ (thus, for any $i$, $\|\mathbb{E}[Z] - Z_i\| \leq 2$).

Combining equation 33 and equation 34, we conclude that the distortion is bounded by

$$
\mathbb{E}_{\hat{W},\Theta,W,S_n}[\Delta(W, \Theta\hat{W}; S_n)] \leq \frac{2}{\sqrt{n}}\left(1 + \frac{2}{d}\right)^{1/4} e^{-\frac{0.21}{4}d(c_w^2 - 1)^2}. \tag{36}
$$

**Bound on $\mathbb{E}_{\tilde{\mathbf{S}},\Theta}[\Delta\ell_{\hat{w}}(\tilde{\mathbf{S}}, \Theta)]$:** We have

$$
\begin{aligned}
\mathbb{E}_{P_{\mathbf{S}}P_\Theta}[\Delta\ell_{\hat{w}}(\tilde{\mathbf{S}}, \Theta)] &:= \mathbb{E}_{P_{\tilde{\mathbf{S}}}P_\Theta P_{W|\tilde{\mathbf{S}}}P_{\hat{W}|\Theta^\top W}}\left[\frac{1}{n}\sum_{i\in[n]}(\ell(Z_{i,0}, \Theta\hat{W}) - \ell(Z_{i,1}, \Theta\hat{W}))^2\right] \\
&= \mathbb{E}_{P_{\tilde{\mathbf{S}}}P_\Theta P_{W|\tilde{\mathbf{S}}}P_{\hat{W}|\Theta^\top W}}\left[\frac{1}{n}\sum_{i\in[n]}\left\langle \hat{W}, \Theta^\top(Z_{i,0} - Z_{i,1})\right\rangle^2\right] \\
&\overset{(a)}{\leq} \mathbb{E}_{P_{\tilde{\mathbf{S}}}P_\Theta P_{W|\tilde{\mathbf{S}}}P_{\hat{W}|\Theta^\top W}}\left[\frac{1}{n}\sum_{i\in[n]}\|\Theta^\top(Z_{i,0} - Z_{i,1})\|^2\|\hat{W}\|^2\right] \\
&\overset{(b)}{\leq} (c_w + \nu)^2 \mathbb{E}_{P_{\tilde{\mathbf{S}}}P_\Theta}\left[\frac{1}{n}\sum_{i\in[n]}\|\Theta^\top(Z_{i,0} - Z_{i,1})\|^2\right] \\
&\overset{(c)}{\leq} 4(c_w + \nu)^2, &(37)
\end{aligned}
$$

where $(a)$ follows by Cauchy–Schwarz inequality, $(b)$ is derived since $\|\hat{w}\| \leq (c_w + \nu)$, and $(c)$ since for any fixed $z$, each entry of $\frac{\Theta^\top z}{\|z\|}$ is an independent random variable distributed according to $\mathcal{N}(0, \frac{1}{d})$ and hence

$$
\mathbb{E}_\Theta\left[\left\|\Theta^\top z\right\|^2\right] = \|z\|^2 \leq 4,
$$

since $\|Z_{i,0} - Z_{i,1}\| \leq 2$.

**Generalization Bound:**  Now, let

$$\epsilon := \frac{2}{\sqrt{n}} \left(1 + \frac{2}{d}\right)^{1/4} e^{-\frac{0.21}{4} d(c_w^2 - 1)^2}.$$

Inequality 36 shows that the above choices of $P_\Theta$ and $P_{\hat{W}|\Theta^\top W}$ (according to the scheme $\mathrm{JL}(d, c_w, \nu)$) satisfy the distortion criterion equation 23. Hence, equation 22 gives

$$\mathrm{gen}(\mu, \mathcal{A}) \leq \mathbb{E}_{P_{\tilde{\mathbf{S}}} P_\Theta} \left[ \sqrt{\frac{2\Delta\ell_{\hat{w}}(\tilde{\mathbf{S}}, \Theta)}{n} \mathsf{CMI}^\Theta(\tilde{\mathbf{S}}, \hat{\mathcal{A}})} \right] + \frac{2}{\sqrt{n}} \left(1 + \frac{2}{d}\right)^{1/4} e^{-\frac{0.21}{4} d(c_w^2 - 1)^2}$$

$$\overset{(a)}{\leq} \mathbb{E}_{P_{\tilde{\mathbf{S}}} P_\Theta} \left[ \sqrt{\frac{2\Delta\ell_{\hat{w}}(\tilde{\mathbf{S}}, \Theta)}{n} d \log\left(\frac{c_w + \nu}{\nu}\right)} \right] + \frac{2}{\sqrt{n}} \left(1 + \frac{2}{d}\right)^{1/4} e^{-\frac{0.21}{4} d(c_w^2 - 1)^2}$$

$$\overset{(b)}{\leq} \sqrt{\frac{2d\, \mathbb{E}_{P_{\tilde{\mathbf{S}}} P_\Theta} \left[\Delta\ell_{\hat{w}}(\tilde{\mathbf{S}}, \Theta)\right]}{n} \log\left(\frac{c_w + \nu}{\nu}\right)} + \frac{2}{\sqrt{n}} \left(1 + \frac{2}{d}\right)^{1/4} e^{-\frac{0.21}{4} d(c_w^2 - 1)^2}$$

$$\overset{(c)}{\leq} \sqrt{\frac{8d(c_w + \nu)^2}{n} \log\left(\frac{c_w + \nu}{\nu}\right)} + \frac{2}{\sqrt{n}} \left(1 + \frac{2}{d}\right)^{1/4} e^{-\frac{0.21}{4} d(c_w^2 - 1)^2},$$

where $(a)$ is achieved using equation 27, $(b)$ by Jensen inequality and due to the concavity of the function $\sqrt{x}$, and $(c)$ is derived using equation 37.

The proof is completed by letting

$$d = 1, \quad c_w = 1, \quad \nu = 0.4.$$

## F.3  Proof of Proposition 1

As explained in Appendix F, it is sufficient to consider the case $L_c = R = 1$. We have

$$\mathrm{gen}(\mu, \mathcal{A}) = \mathbb{E}_{P_{S_n, W}} [\mathcal{R}(W) - \widehat{\mathcal{R}}_n(W)]$$

$$= \frac{1}{n} \sum_{i \in [n]} \mathbb{E}_{P_{S_n, W}} \left[ \mathbb{E}_{Z \sim \mu}[\ell_{sc}(Z, W)] - \ell_{sc}(Z_i, W) \right]$$

$$\overset{(a)}{=} \frac{1}{n} \sum_{i \in [n]} \mathbb{E}_{P_{S_n, W}} \left[ \mathbb{E}_{Z \sim \mu}[-\langle W, Z\rangle] + \langle W, Z_i\rangle \right]$$

$$\overset{(b)}{=} \frac{1}{n} \sum_{i \in [n]} \mathbb{E}_{P_{S_n, W}} \left[ \mathbb{E}_{Z \sim \mu}[\ell_c(Z, W)\rangle] - \ell_c(Z_i, W) \right]$$

$$\overset{(c)}{\leq} \frac{8}{\sqrt{n}},$$

where $(a)$ by definition of $\ell_{sc}(z, w) = -\langle w, z\rangle + \frac{\lambda}{2}\|w\|^2$ by Definition 6, $(b)$ holds since by Definition 4, we have $\ell_c(z, w) = -L\langle w, z\rangle$, and $(c)$ follows by Theorem 3.

## F.4  Proof of Theorem 7

As explained in Appendix F, we consider the case $L = R - B = 1$. First, note that similar to the proof of Proposition 1, the generalization error does not change, if we consider the loss function $\ell_{glm}(z, w) \triangleq g\left(\langle w, \phi(z)\rangle, z\right) - g\left(0, z\right)$ instead of $\ell_{gl}(z, w) = g\left(\langle w, \phi(z)\rangle, z\right) + r(w)$. More precisely,

$$\mathrm{gen}(\mu, \mathcal{A}) = \mathbb{E}_{P_{S_n, W}} [\mathcal{R}(W) - \widehat{\mathcal{R}}_n(W)]$$

$$= \frac{1}{n} \sum_{i \in [n]} \mathbb{E}_{P_{S_n, W}} \left[ \mathbb{E}_{Z \sim \mu}[\ell_{gl}(Z, W)] - \ell_{gl}(Z_i, W) \right]$$

$$= \frac{1}{n} \sum_{i \in [n]} \mathbb{E}_{P_{S_n, W}} \left[ \mathbb{E}_{Z \sim \mu}[g\left(\langle W, \phi(Z)\rangle, Z\right) + r(W)] - g\left(\langle W, \phi(Z_i)\rangle, Z_i\right) - r(W) \right]$$

$$= \frac{1}{n} \sum_{i \in [n]} \mathbb{E}_{P_{S_n, W}} \left[ \mathbb{E}_{Z \sim \mu}[g\left(\langle W, \phi(Z)\rangle, Z\right)] - g\left(\langle W, \phi(Z_i)\rangle, Z_i\right) \right]$$

$$\overset{(a)}{=} \frac{1}{n} \sum_{i \in [n]} \mathbb{E}_{P_{S_n, W}} \left[ \mathbb{E}_{Z \sim \mu} [g\left( \langle W, \phi(Z) \rangle, Z \right) - g\left( 0, Z \right)] - g\left( \langle W, \phi(Z_i) \rangle, Z_i \right) + g\left( 0, Z_i \right) \right]$$

$$= \frac{1}{n} \sum_{i \in [n]} \mathbb{E}_{P_{S_n, W}} \left[ \mathbb{E}_{Z \sim \mu} [\ell_{glm}(Z, W)] - \ell_{glm}(Z_i, W) \right],$$

where $(a)$ follows since $\mathbb{E}_{Z \sim \mu}[g(0, Z)] = \mathbb{E}_{Z_i \sim \mu}[g(0, Z_i)]$.

Hence, for the rest of the proof, we consider the generalization with respect to the following loss function:

$$\ell_{glm}(z, w) \triangleq g\left( \langle w, \phi(z) \rangle, z \right) - g\left( 0, z \right).$$

Note that due the Lipschitzness of the function $g(\cdot, \cdot)$ with respect to its first argument, for every $z \in \mathcal{Z}$ and $w \in \mathcal{W}$, we have

$$|\ell_{glm}(z, w)| = |g\left( \langle w, \phi(z) \rangle, z \right) - g\left( 0, z \right)| \leq |\langle w, \phi(z) \rangle|. \tag{38}$$

Furthermore since $\|w\|, \|\phi(z)\| \leq 1$, using Cauchy-Schwarz inequality yields

$$|\ell_{glm}(z, w)| \leq 1.$$

Now, we proceed to establish a generalization bound with respect to the loss function $\ell_{glm}(z, w)$. We use Theorem 1 with the $\mathsf{JL}(d, c_w, \nu)$ transformation described in Appendix F.1, for some $d \in \mathbb{N}^+$, $c_w \in \left[ 1, \sqrt{5/4} \right)$, and $\nu \in (0, 1]$. To do so, We start by bounding $\mathsf{CMI}^{\Theta}(\tilde{\mathbf{S}}, \hat{\mathcal{A}})$, the distortion equation 23, and $\mathbb{E}_{\tilde{\mathbf{S}}, \Theta}[\Delta \ell_{\hat{w}}(\tilde{\mathbf{S}}, \Theta)]$.

**Bound on the disintegrated CMI:** It is shown in equation 26 that

$$\mathsf{CMI}^{\Theta}(\tilde{\mathbf{S}}, \hat{\mathcal{A}}) = \leq d \log \left( \frac{c_w + \nu}{\nu} \right).$$

**Bound on the distortion:** Next, we bound the distortion term.

$$\Delta(W, \Theta \hat{W}; S_n) = \mathrm{gen}(S_n, W) - \mathrm{gen}(S_n, \Theta \hat{W})$$

$$= \mathbb{E}_{Z \sim \mu}[\ell_{glm}(Z, W)] - \frac{1}{n} \sum_{i=1}^{n} \ell_{glm}(Z_i, W) - \mathbb{E}_{Z \sim \mu}[\ell_{glm}(Z, \Theta \hat{W})] + \frac{1}{n} \sum_{i=1}^{n} \ell_{glm}(Z_i, \Theta \hat{W})$$

$$= \mathbb{E}_{Z \sim \mu} \left[ g\left( \langle W, \phi(Z) \rangle, Z \right) \right] - \frac{1}{n} \sum_{i=1}^{n} g\left( \langle W, \phi(Z_i) \rangle, Z_i \right)$$

$$\quad - \mathbb{E}_{Z \sim \mu} \left[ g\left( \left\langle \Theta \hat{W}, \phi(Z) \right\rangle, Z \right) \right] + \frac{1}{n} \sum_{i=1}^{n} g\left( \left\langle \Theta \hat{W}, \phi(Z_i) \right\rangle, Z_i \right)$$

$$\leq \mathbb{E}_{Z \sim \mu} \left[ \left| g\left( \langle W, \phi(Z) \rangle, Z \right) - g\left( \left\langle \Theta \hat{W}, \phi(Z) \right\rangle, Z \right) \right| \right]$$

$$\quad + \frac{1}{n} \sum_{i=1}^{n} \left| g\left( \langle W, \phi(Z_i) \rangle, Z_i \right) - g\left( \left\langle \Theta \hat{W}, \phi(Z_i) \right\rangle, Z_i \right) \right|$$

$$\overset{(a)}{\leq} \mathbb{E}_{Z \sim \mu} \left[ \left| \left\langle W - \Theta \hat{W}, \phi(Z) \right\rangle \right| \right] + \frac{1}{n} \sum_{i \in [n]} \left| \left\langle W - \Theta \hat{W}, \phi(Z_i) \right\rangle \right|, \tag{39}$$

where $(a)$ holds due to Lipschitzness of the function $g$ with respect to its first argument.

Hence,

$$\mathbb{E}_{\hat{W}, \Theta, W, S_n} \left[ \Delta(W, \Theta \hat{W}; S_n) \right] \leq 2 \sup_{z, w} \mathbb{E}_{\hat{W}, \Theta \sim P_\Theta P_{\hat{W} | \Theta^\top w}} \left[ \left| \left\langle w - \Theta \hat{W}, \phi(z) \right\rangle \right| \right]$$

$$= 2 \sup_{z, w} \mathbb{E}_{\hat{W}, \Theta \sim P_\Theta P_{\hat{W} | \Theta^\top w}} \left[ \left| \langle w, \phi(z) \rangle - \left\langle \hat{W}, \Theta^\top \phi(z) \right\rangle \right| \right]$$

$$\leq 2 \sup_{z, w} \left( \mathbb{E}_\Theta \left[ \left| \langle w, \phi(z) \rangle - \left\langle U, \Theta^\top \phi(z) \right\rangle \right| \right] + \mathbb{E}_{V_\nu, \Theta} \left[ \left| \left\langle V_\nu, \Theta^\top \phi(z) \right\rangle \right| \right] \right), \tag{40}$$

where the last step follows since by equation 25, $\hat{W} = U + V_\nu$.

In the rest, we fix $z$ and $w$ and upper bound each of the terms in the right-hand side of equation 40:

$$C_1 \triangleq \mathbb{E}_{\Theta \sim P_\Theta} \left[ \left| \langle w, \phi(z) \rangle - \left\langle U, \Theta^\top \phi(z) \right\rangle \right| \right],$$

$$C_2 \triangleq \mathbb{E}_{V_\nu, \Theta \sim \text{Uniform}(\mathcal{B}_d(\nu)) P_\Theta} \left[ \left| \left\langle V_\nu, \Theta^\top \phi(z) \right\rangle \right| \right].$$

Let $\mathcal{E}$ be the event that $\|\Theta^\top W\| > c_w$ and denote by $\mathcal{E}^c$ the complementary event of $\mathcal{E}$.

- We start by bounding $C_1$.

$$C_1 = \mathbb{E}_\Theta \left[ \left| \langle w, \phi(z) \rangle - \left\langle U, \Theta^\top \phi(z) \right\rangle \right| \mathbb{1}\{\mathcal{E}\} \right] + \mathbb{E}_\Theta \left[ \left| \langle w, \phi(z) \rangle - \left\langle U, \Theta^\top \phi(z) \right\rangle \right| \mathbb{1}\{\mathcal{E}^c\} \right]$$

$$\overset{(a)}{=} \mathbb{E}_\Theta \left[ \left| \langle w, \phi(z) \rangle - \left\langle \mathbf{0}_d, \Theta^\top \phi(z) \right\rangle \right| \mathbb{1}\{\mathcal{E}\} \right] + \mathbb{E}_\Theta \left[ \left| \langle w, \phi(z) \rangle - \left\langle \Theta^\top w, \Theta^\top \phi(z) \right\rangle \right| \mathbb{1}\{\mathcal{E}^c\} \right]$$

$$\leq \mathbb{E}_\Theta \left[ |\langle w, \phi(z) \rangle| \mathbb{1}\{\mathcal{E}\} \right] + \mathbb{E}_\Theta \left[ \left| \langle w, \phi(z) \rangle - \left\langle \Theta^\top w, \Theta^\top \phi(z) \right\rangle \right| \right]$$

$$\overset{(b)}{\leq} \mathbb{E}_\Theta \left[ \mathbb{1}\{\mathcal{E}\} \right] + \mathbb{E}_\Theta \left[ \left| \langle w, \phi(z) \rangle - \left\langle \Theta^\top w, \Theta^\top \phi(z) \right\rangle \right| \right]$$

$$\overset{(c)}{\leq} e^{-0.21d(1-c_w^2)^2} + \mathbb{E}_\Theta \left[ \left| \langle w, \phi(z) \rangle - \left\langle \Theta^\top w, \Theta^\top \phi(z) \right\rangle \right| \right], \tag{41}$$

where $(a)$ holds since by equation 24, under $\mathcal{E}$, $U = \mathbf{0}_d$, and under $\mathcal{E}^c$, $U = \Theta^\top W$, $(b)$ is derived since $\|w\|, \|\phi(z)\| \leq 1$ and hence, Cauchy-Schwarz inequality yields $|\langle w, \phi(z) \rangle| \leq 1$, and $(c)$ derived by equation 32.

Thus, to bound $C_1$, it remained to bound $\mathbb{E}_\Theta \left[ \left| \langle w, \phi(z) \rangle - \left\langle \Theta^\top w, \Theta^\top \phi(z) \right\rangle \right| \right]$. We use a trick borrowed from [71, Proof of Theorem 9]. Note that $\|w\|, \|\phi(z)\| \leq 1$. Hence, to upper bound $\mathbb{E}_\Theta \left[ \left| \langle w, \phi(z) \rangle - \left\langle \Theta^\top w, \Theta^\top \phi(z) \right\rangle \right| \right]$, it is sufficient to consider the case where $\|w\| = \|\phi(z)\| = 1$. Let

$$v \triangleq w - \langle w, \phi(z) \rangle \phi(z),$$

$$\hat{v} \triangleq \frac{v}{\|v\|}.$$

It is easy to verify that $\langle v, \phi(z) \rangle = 0$. Hence, since $\phi(z) \perp v$, we have

$$\|v\| = \sqrt{\|w\|^2 - \langle w, \phi(z) \rangle^2 \|\phi(z)\|^2} = \sqrt{1 - \langle w, \phi(z) \rangle^2} \leq 1.$$

Now, for every $r \in [d]$, denote the $r$'th row of $\Theta^\top \in \mathbb{R}^{d \times D}$ by $T_r$ and let

$$X_r \triangleq \langle T_r, \phi(z) \rangle,$$

$$Y_r \triangleq \langle T_r, \hat{v} \rangle.$$

Since $\phi(z) \perp v$ and since the Gaussian distributions are rotationally invariant, we have that $X_1, \ldots, X_d, Y_1, \ldots, Y_d$ are i.i.d. Gaussian random variables distributed according to $\mathcal{N}(0, 1/d)$. Hence, using the identity $w = v + \langle w, \phi(z) \rangle \phi(z)$, we can write

$$\left| \langle w, \phi(z) \rangle - \langle \Theta^\top w, \Theta^\top \phi(z) \rangle \right| = \left| \langle w, \phi(z) \rangle - \left\langle \Theta^\top \left( v + \langle w, \phi(z) \rangle \phi(z) \right), \Theta^\top \phi(z) \right\rangle \right|$$

$$= \left| \langle w, \phi(z) \rangle \left( 1 - \left\| \Theta^\top \phi(z) \right\|^2 \right) - \|v\| \langle \Theta^\top \hat{v}, \Theta^\top \phi(z) \rangle \right|$$

$$\overset{(a)}{\leq} \left| \left\| \Theta^\top \phi(z) \right\|^2 - 1 \right| + \left| \langle \Theta^\top \hat{v}, \Theta^\top \phi(z) \right|$$

$$= \left| \left\| \Theta^\top \phi(z) \right\|^2 - 1 \right| + \left| \sum_{r \in [d]} X_r Y_r \right|, \tag{42}$$

where $(a)$ is derived using the inequalities $\|\langle w, \phi(z) \rangle \leq 1$ and $\|v\| \leq 1$.

We bound the expectation over $\Theta$ of each of these terms, denoted respectively as

$$C_{1,1} \triangleq \mathbb{E}_\Theta \left[ \left| \left\| \Theta^\top \phi(z) \right\|^2 - 1 \right| \right],$$

$$C_{1,2} \triangleq \mathbb{E}_\Theta \left[ \left| \sum_{r \in [d]} X_r Y_r \right| \right].$$

- Note that the distribution of

$$d \left\| \Theta^\top \phi(z) \right\|^2,$$

is a chi-squared distribution $\chi^2(d)$ with $d$-degrees of freedom. Moreover, asymptotically as $d \to \infty$, $\chi^2(d)$ converges to $\mathcal{N}(d, 2d)$. Equivalently, asymptotically, $\chi^2(d) - d \to \mathcal{N}(0, 2d)$. Combining this asymptotic behavior with the fact that for a Gaussian random variable $\mathfrak{Z} \sim \mathcal{N}(0, \sigma^2)$, with $\sigma \in \mathbb{R}_+$, we have that $\mathbb{E}[|\mathfrak{Z}|] = \sigma \sqrt{\frac{2}{\pi}}$, yield

$$C_{1,1} \leq \mathcal{O} \left( \frac{1}{\sqrt{d}} \right). \tag{43}$$

- To bound the term $C_{1,2}$, notice that $\sum_{r \in [d]} X_r Y_r$ converges to a random variable with Gaussian distribution $\mathcal{N}(0, 1/d)$, as $d \to \infty$. Hence, once again using the fact that for a Gaussian random variable $\mathfrak{Z} \sim \mathcal{N}(0, \sigma^2)$, $\mathbb{E}[|\mathfrak{Z}|] = \sigma \sqrt{\frac{2}{\pi}}$, yield

$$C_{1,2} \triangleq \mathbb{E}_\Theta \left[ \left| \sum_{r \in [d]} X_r Y_r \right| \right] = \mathcal{O} \left( \frac{\nu}{\sqrt{d}} \right).$$

Combining equation 41, equation 42, and equation 43 gives

$$C_1 \triangleq \mathbb{E}_{\Theta \sim P_\Theta} \left[ \left| \langle w, \phi(z) \rangle - \left\langle U, \Theta^\top \phi(z) \right\rangle \right| \right] \leq e^{-0.21d(1-c_w^2)^2} + \mathcal{O} \left( \frac{\|\phi(z)\| \|w\|}{\sqrt{d}} \right) \tag{44}$$

$$\leq e^{-0.21d(1-c_w^2)^2} + \mathcal{O} \left( \frac{1}{\sqrt{d}} \right). \tag{45}$$

- Now to bound $C_2$, let $V_\nu = (V_{\nu,1}, \ldots, V_{\nu,d})$.

$$C_2 = \mathbb{E}_{\Theta \sim P_\Theta} \mathbb{E}_{V_\nu \sim \text{Uniform}(\mathcal{B}_d(\nu))} \left[ \left| \left\langle V_\nu, \Theta^\top \phi(z) \right\rangle \right| \right]$$

$$\overset{(a)}{=} \mathbb{E}_{\Theta \sim P_\Theta} \mathbb{E}_{V_\nu \sim \text{Uniform}(\mathcal{B}_d(\nu))} \left[ |V_{\nu,1}| \left\| \Theta^\top \phi(z) \right\| \right]$$

$$= \mathbb{E}_{\Theta \sim P_\Theta} \left[ \left\| \Theta^\top \phi(z) \right\| \right] \mathbb{E}_{V_\nu \sim \text{Uniform}(\mathcal{B}_d(\nu))} \left[ |V_{\nu,1}| \right]$$

$$\overset{(b)}{\leq} \mathbb{E}_{V_\nu \sim \text{Uniform}(\mathcal{B}_d(\nu))} \left[ |V_{\nu,1}| \right]$$

$$\overset{(c)}{=} \frac{\nu \Gamma \left( \frac{d+1}{2} + \frac{1}{2} \right)}{\sqrt{\pi} \Gamma \left( \frac{d+1}{2} + 1 \right)}$$

$$\overset{(d)}{\leq} \frac{\nu \sqrt{2}}{\sqrt{\pi(d+1))}}, \tag{46}$$

where $(a)$ holds by the symmetry of the distribution of $V_\nu$, $(b)$ holds since $\mathbb{E}_{\Theta \sim P_\Theta} \left[ \|\Theta^\top \phi(z)\| \right] \leq \mathbb{E}_{\Theta \sim P_\Theta} \left[ \|\Theta^\top \phi(z)\|^2 \right]^{1/2} = \|\phi(z)\| \leq 1$, $(c)$ holds by Lemma 4, proved in Appendix F.5, and $(d)$ holds since by using Gautschi's inequality we have $\frac{\Gamma(x+1/2)}{\Gamma(x+1)} \leq \frac{1}{\sqrt{x}}$.

**Lemma 4.** *Let* $V_\nu = (V_{\nu,1}, \ldots, V_{\nu,d}) \sim \text{Uniform}(\mathcal{B}_d(\nu))$. *Then,* $\mathbb{E}_{V_\nu \sim \text{Uniform}(\mathcal{B}_d(\nu))} \left[ |V_{\nu,1}| \right] = \frac{\nu \Gamma \left( \frac{d+2}{2} \right)}{\sqrt{\pi} \Gamma \left( \frac{d+3}{2} \right)}$.

Combining equation 39. equation 45, and equation 46 gives

$$\mathbb{E}_{\hat{W}, \Theta, W, S_n} \left[ \Delta(W, \Theta \hat{W}; S_n) \right] \leq e^{-0.21d(1-c_w^2)^2} + \mathcal{O} \left( \frac{1}{\sqrt{d}} \right). \tag{47}$$

**Bound on** $\mathbb{E}_{\tilde{\mathbf{S}}, \Theta}[\Delta \ell_{\hat{w}}(\tilde{\mathbf{S}}, \Theta)]$**:** We have

$$|\ell_{glm}(z, \Theta \hat{w})| \overset{(a)}{\leq} |\langle \Theta \hat{w}, \phi(z) \rangle|$$

$$= \left| \left\langle \hat{w}, \Theta^\top \phi(z) \right\rangle \right|$$

$$\leq \|\hat{w}\| \|\Theta^\top \phi(z)\|$$

$$\overset{(b)}{\leq} (c_w + \nu) \|\Theta^\top \phi(z)\|, \tag{48}$$

where $(a)$ holds by equation 38 and $(b)$ since by construction $\|\hat{w}\| \leq c_w + \nu$.

Hence,

$$
\begin{aligned}
\mathbb{E}_{\tilde{\mathbf{S}},\Theta}[\Delta\ell_{\hat{w}}(\tilde{\mathbf{S}},\Theta)] :=& \mathbb{E}_{P_{\tilde{\mathbf{S}}} P_{\Theta} P_{W|\tilde{\mathbf{S}}} P_{\hat{W}|\Theta^{\top}W}} \left[ \frac{1}{n} \sum_{i\in[n]} (\ell_{glm}(Z_{i,0}, \Theta\hat{W}) - \ell_{glm}(Z_{i,1}, \Theta\hat{W}))^2 \right] \\
&\overset{(a)}{\leq} (c_w + \nu)^2 \mathbb{E}_{P_{\tilde{\mathbf{S}}} P_{\Theta} P_{W|\tilde{\mathbf{S}}} P_{\hat{W}|\Theta^{\top}W}} \left[ \frac{1}{n} \sum_{i\in[n]} (\|\Theta^{\top}\phi(Z_{i,0})\| + \|\Theta^{\top}\phi(Z_{i,1})\|)^2 \right] \\
=& (c_w + \nu)^2 \mathbb{E}_{P_{\tilde{\mathbf{S}}} P_{\Theta}} \left[ \frac{1}{n} \sum_{i\in[n]} (\|\Theta^{\top}\phi(Z_{i,0})\| + \|\Theta^{\top}\phi(Z_{i,1})\|)^2 \right] \\
=& 4(c_w + \nu)^2 \sup_{z} \mathbb{E}_{P_{\Theta}} \left[ \|\Theta^{\top}\phi(z)\|^2 \right] \\
\overset{(b)}{=}& 4(c_w + \nu)^2 \sup_{z} \|\phi(z)\|^2 \\
\leq& 4(c_w + \nu)^2,
\end{aligned}
$$

where

- $(a)$ follows from equation 48,
- $(b)$ since for any fixed $z$, each entry of $\frac{\Theta^{\top}z}{\|z\|}$ is an independent random variable distributed according to $\mathcal{N}(0, \frac{1}{d})$ and hence

$$
\mathbb{E}_{\Theta} \left[ \|\Theta^{\top}z\|^2 \right] = \|z\|^2.
$$

**Generalization Bound:** Now, using Theorem 1 for the above choices of $P_{\Theta}$ and $P_{\hat{W}|\Theta^{\top}W}$ (according to the scheme JL$(d, c_w, \nu)$) gives

$$
\begin{aligned}
\text{gen}(\mu, \mathcal{A}) \leq& \mathbb{E}_{\tilde{\mathbf{S}},\Theta} \left[ \sqrt{\frac{2\Delta\ell_{\hat{w}}(\tilde{\mathbf{S}},\Theta)}{n} \mathsf{CMI}^{\Theta}(\tilde{\mathbf{S}}, \hat{\mathcal{A}})} \right] + \mathbb{E}_{\hat{W},\Theta,W,S_n} \left[ \Delta(W, \Theta\hat{W}; S_n) \right] \\
\overset{(a)}{\leq}& \mathbb{E}_{\tilde{\mathbf{S}},\Theta} \left[ \sqrt{\frac{2\Delta\ell_{\hat{w}}(\tilde{\mathbf{S}},\Theta)}{n} d \log\left( \frac{c_w + \nu}{\nu} \right)} \right] + e^{-0.21d(1-c_w^2)^2} + \mathcal{O}\left( \frac{1}{\sqrt{d}} \right) \\
\overset{(b)}{\leq}& \sqrt{\frac{2d \, \mathbb{E}_{\tilde{\mathbf{S}},\Theta}\left[ \Delta\ell_{\hat{w}}(\tilde{\mathbf{S}},\Theta) \right]}{n} \log\left( \frac{c_w + \nu}{\nu} \right)} + e^{-0.21d(1-c_w^2)^2} + \mathcal{O}\left( \frac{1}{\sqrt{d}} \right) \\
\overset{(c)}{\leq}& \sqrt{\frac{8d(c_w + \nu)^2}{n} \log\left( \frac{c_w + \nu}{\nu} \right)} + e^{-0.21d(1-c_w^2)^2} + \mathcal{O}\left( \frac{1}{\sqrt{d}} \right),
\end{aligned}
$$

where $(a)$ is achieved using equation 27 and equation 47, $(b)$ by Jensen inequality and due to the concavity of the function $\sqrt{x}$, and $(c)$ is derived using equation 37.

The proof is completed by letting

$$
d = \sqrt{n}, \quad c_w = 1.1, \quad \nu = 0.5.
$$

### F.5 Proof of Lemma 4

Note that

$$
\mathbb{E}_{V_{\nu} \sim \text{Uniform}(\mathcal{B}_d(\nu))} \left[ |V_{\nu,1}| \right] = \nu \mathbb{E}_{X \sim \text{Uniform}(\mathcal{B}_d(1))} \left[ |X_1| \right],
$$

where $X = (X_1, \ldots, X_d) \sim \text{Uniform}(\mathcal{B}_d(1))$. Hence, it is sufficient to show that $\mathbb{E}_{X \sim \text{Uniform}(\mathcal{B}_d(1))} \left[ |X_1| \right] = \frac{\Gamma\left(\frac{d+2}{2}\right)}{\sqrt{\pi}\Gamma\left(\frac{d+3}{2}\right)}$.

First, we compute the marginal distribution of $X_1$. Note that

$$
\begin{aligned}
f_{X_1}(x_1) &= \frac{1}{\text{Volume}(\mathcal{B}_d(1))} \int_{x_2=-\sqrt{1-x_1^2}}^{\sqrt{1-x_1^2}} \cdots \int_{x_d=-\sqrt{1-x_1^2-\cdots-x_{d-1}^2}}^{\sqrt{1-x_1^2-\cdots-x_{d-1}^2}} \mathrm{d}x_2 \cdots \mathrm{d}x_d \\
&= \frac{\text{Volume}\left(\mathcal{B}_{d-1}\left(\sqrt{1-x_1^2}\right)\right)}{\text{Volume}(\mathcal{B}_d(1))} \\
&= \frac{\Gamma\left(\frac{d+2}{2}\right)}{\sqrt{\pi}\Gamma\left(\frac{d+1}{2}\right)} \left(1-x_1^2\right)^{\frac{d-1}{2}}.
\end{aligned}
$$

Now, we have

$$
\begin{aligned}
\mathbb{E}_{X\sim\text{Uniform}(\mathcal{B}_d(1))}\left[|X_1|\right] &= E_{X_1\sim f_{X_1}}\left[|X_1|\right] \\
&= \frac{2\Gamma\left(\frac{d+2}{2}\right)}{\sqrt{\pi}\Gamma\left(\frac{d+1}{2}\right)} \int_{x_1=0}^{1} x_1\left(1-x_1^2\right)^{\frac{d-1}{2}} \mathrm{d}x_1 \\
&\stackrel{(a)}{=} \frac{\Gamma\left(\frac{d+2}{2}\right)}{\sqrt{\pi}\Gamma\left(\frac{d+1}{2}\right)} \int_{u=0}^{1} \left(1-u\right)^{\frac{d-1}{2}} \mathrm{d}u \\
&\stackrel{(b)}{=} \frac{\Gamma\left(\frac{d+2}{2}\right)}{\sqrt{\pi}\Gamma\left(\frac{d+1}{2}\right)} \text{Beta}\left(1,(d+1)/2\right) \\
&= \frac{\Gamma\left(\frac{d+2}{2}\right)}{\sqrt{\pi}\Gamma\left(\frac{d+1}{2}\right)} \times \frac{\Gamma(1)\Gamma\left(\frac{d+1}{2}\right)}{\Gamma\left(\frac{d+3}{2}\right)} \\
&= \frac{\Gamma\left(\frac{d+2}{2}\right)}{\sqrt{\pi}\Gamma\left(\frac{d+3}{2}\right)},
\end{aligned}
$$

where $(a)$ is achieved by letting $u = x_1^2$ and in $(b)$, $\text{Beta}(\cdot,\cdot)$ is the Beta function.

# G  Proofs of Section 5 and Appendix C: Memorization

In this section, we provide the proofs of Section 5 and Appendix C. Recall that for a given $K_i$, the adversary outputs its guess of $K_i$ as $\hat{K}_i \triangleq \mathcal{Q}(W, Z_{i,K_i}, \mu)$. Throughout the proofs and for better readability, we sometimes denote $\hat{K}_i = 1$ by $\hat{K}_i = $ 'in' and $\hat{K}_i = 0$ by $\hat{K}_i = $ 'not in', referring to the semantic meaning that the given $Z_{i,K_i}$ is part of the training dataset or not.

## G.1  Proof of Theorem 5

We prove each part separately. As stated in the beginning of Appendix G, throughout the proofs and for better readability, we sometimes denote $\hat{K}_i = 1$ by $\hat{K}_i = $ 'in' and $\hat{K}_i = 0$ by $\hat{K}_i = $ 'not in', referring to the semantic meaning that the given $Z_{i,K_i}$ is part of the training dataset or not.

### G.1.1  Part i.

We prove the result by contradiction. Suppose that there exists an adversary for the algorithm $\mathcal{A}$ that is $\xi$-sound and certifies a recall of $m$ samples with probability $q$, where $\xi < q$ and $m = \Omega(n)$. As before, we denote the output of the learning algorithm by $\mathcal{A}_n(S_n) = W$.

Recall that $\tilde{\mathbf{S}}_{\mathbf{J}} = \{Z_{1,J_1}, Z_{2,J_2}, \ldots, Z_{n,J_n}\}$ is the training dataset $S_n$ and $\tilde{\mathbf{S}} \setminus \tilde{\mathbf{S}}_{\mathbf{J}}$ is the test dataset $S_n'$.

Define $\hat{J}_i \in \{0,1\}$ as follows:

$$
\hat{J}_i = \begin{cases} 0, & \text{if } \mathcal{Q}(\hat{W}, Z_{i,0}, \mu) = \text{'in' and } \mathcal{Q}(\hat{W}, Z_{i,1}, \mu) = \text{'not in'}, \\ 1, & \text{if } \mathcal{Q}(\hat{W}, Z_{i,0}, \mu) = \text{'not in' and } \mathcal{Q}(\hat{W}, Z_{i,1}, \mu) = \text{'in'}, \\ U_i, & \text{otherwise}, \end{cases}
$$

where $U_i \sim \text{Bern}(1/2)$ is a binary uniform random variable, independent of other random variables.

Recall that given $\mathcal{A}$, a $\xi$-sound adversary means that,

$$
\mathbb{P}\left(\exists i \in [n] \colon \mathcal{Q}(W, Z_{i,J_i^c}, \mu) = \text{'in'}\right) \leq \xi,
$$

and an adversary certifying a recall of $m$ samples means that,

$$\mathbb{P}\left(\sum_{i\in[n]} \mathbb{1}\{\mathcal{Q}(W, Z_{i,J_i}, \mu) = \text{`in'}\} \geq m\right) \geq q.$$

Since we assumed $m = \Omega(n)$, there exists $c_1 \in (0, 1]$ and $n_0 \in \mathbb{N}$ such that, for all $n \geq n_0$, $m \geq c_1 n$. The second condition then yields,

$$\mathbb{P}\left(\sum_{i\in[n]} \mathbb{1}\{\mathcal{Q}(W, Z_{i,J_i}, \mu) = \text{`in'}\} \geq c_1 n\right) \geq q.$$

Define the Hamming distance $d_H \colon \{0, 1\}^n \times \{0, 1\}^n \to [n]$ between binary vectors $\mathbf{J}$ and $\hat{\mathbf{J}}$ as

$$d_H\left(\mathbf{J}, \hat{\mathbf{J}}\right) = \sum_{i\in[n]} \mathbb{1}\{J_i \neq \hat{J}_i\}.$$

Next, we use Fano's inequality with approximate recovery [59, Theorem 2]. Let $t = \frac{1}{n}\left\lfloor \frac{n}{2}\left(1 - \frac{c_1}{2}\right)\right\rfloor$ and denote

$$P_{e_t} \triangleq \mathbb{P}\left(d_H\left(\mathbf{J}, \hat{\mathbf{J}}\right) > nt\right),$$

$$N_{\hat{\mathbf{j}}} \triangleq \sum_{\mathbf{j}\in\{0,1\}^n} \mathbb{1}\left\{d_H(\mathbf{j}, \hat{\mathbf{j}}) \leq nt\right\}.$$

Note that $N_{\hat{\mathbf{j}}}$ is the same for all $\hat{\mathbf{j}} \in \{0, 1\}^n$. Indeed, $d_H(\mathbf{j}, \hat{\mathbf{j}}) = d_H(\mathbf{j} \oplus \mathbf{a}, \hat{\mathbf{j}} \oplus \mathbf{a})$, where $\oplus$ denotes the modulo two summation, for any $\mathbf{a} \in \{0, 1\}^n$, and $\sum_{\mathbf{j}\in\{0,1\}^n} \mathbb{1}\left\{d_H(\mathbf{j} \oplus \mathbf{a}, \hat{\mathbf{j}} \oplus \mathbf{a}) \leq nt\right\} = \sum_{\mathbf{j}\in\{0,1\}^n} \mathbb{1}\left\{d_H(\mathbf{j}, \hat{\mathbf{j}} \oplus \mathbf{a}) \leq nt\right\}$. Hence, $N_{\hat{\mathbf{j}}} = N_{\hat{\mathbf{j}}\oplus\mathbf{a}}$ for any $\mathbf{a}$, and the maximum over $\hat{\mathbf{j}}$ of $N_{\hat{\mathbf{j}}}$ is equal to $N_{\mathbf{1}_n}$.

With these notations, we have

$$o(n) \overset{(a)}{=} \mathsf{I}(\mathbf{J}; W|\tilde{\mathbf{S}})$$

$$\overset{(b)}{=} \mathsf{I}(\mathbf{J}; W|\tilde{\mathbf{S}}, \mathbf{K})$$

$$\overset{(c)}{=} \mathsf{I}(\mathbf{J}; W, \hat{\mathbf{J}}|\tilde{\mathbf{S}}, \mathbf{K})$$

$$\overset{(d)}{\geq} \mathsf{I}(\mathbf{J}; \hat{\mathbf{J}}|\tilde{\mathbf{S}}, \mathbf{K})$$

$$\overset{(e)}{\geq} \mathsf{I}(\mathbf{J}; \hat{\mathbf{J}})$$

$$\overset{(f)}{\geq} (1 - P_{e_t})\log\left(\frac{2^n}{N_{\mathbf{1}_n}}\right) - \log(2)$$

$$\overset{(g)}{\geq} n\left(1 - P_{e_t}\right)\left(1 - h_b(t)\right) - (1 - P_{e_t})\log(c_3) - \log(2),$$

where $(a)$ follows by the assumption of the theorem, $(b)$ results from $\mathbf{K}$ is independent of $(W, \tilde{\mathbf{S}}, \mathbf{J})$, $(c)$ results from $\mathsf{I}(\mathbf{J}; \hat{\mathbf{J}}|W, \tilde{\mathbf{S}}, \mathbf{K}) = 0$ since $\hat{\mathbf{J}}$ is a function of $(W, \tilde{\mathbf{S}}, \mathbf{K})$, $(d)$ results from $\mathsf{I}(\mathbf{J}; W, \hat{\mathbf{J}}|\tilde{\mathbf{S}}, \mathbf{K}) = \mathsf{I}(\mathbf{J}; \hat{\mathbf{J}}|\tilde{\mathbf{S}}, \mathbf{K}) + \mathsf{I}(\mathbf{J}; W|\tilde{\mathbf{S}}, \mathbf{K}, \hat{\mathbf{J}}) \geq \mathsf{I}(\mathbf{J}; \hat{\mathbf{J}}|\tilde{\mathbf{S}}, \mathbf{K})$ (by the positivity of mutual information), $(e)$ is due to the identities below,

$$\mathsf{I}(\mathbf{J}; \hat{\mathbf{J}}|\tilde{\mathbf{S}}, \mathbf{K}) = H(\mathbf{J}) - H(\mathbf{J}|\hat{\mathbf{J}}, \tilde{\mathbf{S}}, \mathbf{K}) \geq H(\mathbf{J}) - H(\mathbf{J}|\hat{\mathbf{J}}) = \mathsf{I}(\mathbf{J}; \hat{\mathbf{J}}),$$

$(f)$ results from applying Fano's inequality with approximate recovery [59, Theorem 2], and $(g)$ is derived using the claim, proved later below, that $N_{\mathbf{1}_n} \leq c_3 2^{nh_b(t)}$ for some constant $c_3 \in \mathbb{R}_+$ and for $n$ sufficiently large.

Note that $t = \frac{1}{n}\left\lfloor \frac{n}{2}\left(1 - \frac{c_1}{2}\right)\right\rfloor < 1/2$ and as $n \to \infty$, $t \to \frac{1-c_1/2}{2} < 1/2$. Hence, since $h_b(x)$ is a continuous function of $x \in [0, 1]$, $1 - h_b(t)$ converges to the constant $1 - h_b\left(\frac{1-c_1/2}{2}\right) > 0$. Hence, if we show that for sufficiently large $n$, $1 - P_{e_t} > 0$, we obtain a contradiction. Since the left-hand side is of order $o(n)$, which is greater than the right-hand side, which is $\Omega(n)$, and the proof is complete.

Hence, it remains to show for $n$ sufficiently large, **Claim i)** $N_{\mathbf{1}_n} \le c_3 2^{nh_b(t)}$ for some constant $c_3 \in \mathbb{R}_+$, and **Claim ii)** $P_{e_t} < 1$.

**Proof of Claim i)**

We have

$$
\begin{aligned}
N_{\mathbf{1}_n} &= \sum_{\mathbf{j} \in \{0,1\}^n} \mathbb{1}\{d_H(\mathbf{j}, \mathbf{1}_n) \le nt\} \\
&= \sum_{i=0}^{nt} \binom{n}{i} \\
&\overset{(a)}{\le} \sum_{i=0}^{nt} \binom{n'}{i} \\
&\overset{(b)}{\le} 2^{n'-1} \frac{\binom{n'}{nt+1}}{\binom{n'}{\frac{n'}{2}}} \\
&\overset{(c)}{\le} 2^{n'h_b((nt+1)/n')} \sqrt{\frac{1}{4\pi(nt/n' + 1/n')(1 - nt/n' - 1/n')}} \\
&\overset{(d)}{\le} c_3 2^{nh_b(t)},
\end{aligned}
\tag{49}
$$

where $n' = 2\left\lceil \frac{n}{2} \right\rceil$ and $c_3 \in \mathbb{R}_+$, $(a)$ results from $n' \ge n$, $(b)$ follows from applying [72, Proposition 5.18][8] ($n'$ is even and $nt \le n'/2 - 1$), $(c)$ is derived using the relation

$$
e^{mh_b(j/m)} \sqrt{\frac{m}{8j(m-j)}} \le \binom{m}{j} \le e^{mh_b(j/m)} \sqrt{\frac{m}{2\pi j(m-j)}},
$$

which is valid for any $m \in \mathbb{N}$ and $1 \le j \le m-1$ (see [73, Exercise 5.8.a]), and $(d)$ holds for sufficiently large $n$, using $n \le n' \le n+1$.

**Proof of Claim ii)** Define the following events: $\mathcal{E}_1 \triangleq \left\{ \exists i \in [n] \colon \mathcal{Q}(W, Z_{i,J_i^c}, \mu) = \text{`in'} \right\}$, $\mathcal{E}_2 \triangleq \left\{ \sum_{i \in [n]} \mathbb{1}\{\mathcal{Q}(W, Z_{i,J_i}, \mu) = \text{`in'}\} < c_1 n \right\}$. Then, we have

$$
\begin{aligned}
P_{e_t} &\triangleq \mathbb{P}\left( d_H\left(\mathbf{J}, \hat{\mathbf{J}}\right) > nt \right) \\
&= \mathbb{P}\left( d_H\left(\mathbf{J}, \hat{\mathbf{J}}\right) > nt, \mathcal{E}_1^c, \mathcal{E}_2^c \right) + \mathbb{P}\left(\mathcal{E}_1, \mathcal{E}_2\right) \\
&\overset{(a)}{=} \mathbb{P}\left( \sum_{i \in [n]} \mathbb{1}\{U_i \ne J_i\} \mathbb{1}\{\mathcal{Q}(W, Z_{i,J_i}, \mu) = \text{`not in'}\} > nt, \mathcal{E}_1^c, \mathcal{E}_2^c \right) + \mathbb{P}\left(\mathcal{E}_1, \mathcal{E}_2\right) \\
&\overset{(b)}{\le} \sum_{r \in [\lceil n(1-c_1)\rceil]} \mathbb{P}\bigg( \sum_{i \in [n]} \mathbb{1}\{U_i \ne J_i\} \mathbb{1}\{\mathcal{Q}(W, Z_{i,J_i}, \mu) = \text{`not in'}\} > nt, \\
&\qquad\qquad\qquad\qquad \sum_{i \in [n]} \mathbb{1}\{\mathcal{Q}(W, Z_{i,J_i}, \mu) = \text{`not in'}\} = r, \mathcal{E}_1^c, \mathcal{E}_2^c \bigg) + \mathbb{P}\left(\mathcal{E}_1, \mathcal{E}_2\right) \\
&\le \sum_{r \in [\lceil n(1-c_1)\rceil]} \mathbb{P}\bigg( \sum_{i \in [n]} \mathbb{1}\{U_i \ne J_i\} \mathbb{1}\{\mathcal{Q}(W, Z_{i,J_i}, \mu) = \text{`not in'}\} > nt, \\
&\qquad\qquad\qquad\qquad \sum_{i \in [n]} \mathbb{1}\{\mathcal{Q}(W, Z_{i,J_i}, \mu) = \text{`not in'}\} = r \bigg) + \mathbb{P}\left(\mathcal{E}_1, \mathcal{E}_2\right) \\
&= \sum_{r \in [\lceil n(1-c_1)\rceil]} \mathbb{P}\bigg( \sum_{i \in [n]} \mathbb{1}\{U_i \ne J_i\} \mathbb{1}\{\mathcal{Q}(W, Z_{i,J_i}, \mu) = \text{`not in'}\} > nt \Big| \sum_{i \in [n]} \mathbb{1}\{\mathcal{Q}(W, Z_{i,J_i}, \mu) = \text{`not in'}\} = r \bigg)
\end{aligned}
$$

---

[8]See also https://mathoverflow.net/questions/17202/sum-of-the-first-k-binomial-coefficients-for-fixed-n for a reformulation.

$$\times \mathbb{P}\left(\sum_{i\in[n]} \mathbb{1}\left\{\mathcal{Q}(W, Z_{i,J_i}, \mu) = \text{`not in'}\right\} = r\right)$$

$$+ \mathbb{P}\left(\mathcal{E}_1, \mathcal{E}_2\right)$$

$$\overset{(c)}{=} \sum_{r\in[nt, \lceil n(1-c_1)\rceil]} \mathbb{P}\left(\sum_{i\in[n]} \mathbb{1}\left\{U_i \neq J_i\right\}\mathbb{1}\left\{\mathcal{Q}(W, Z_{i,J_i}, \mu) = \text{`not in'}\right\} > nt \Big| \sum_{i\in[n]} \mathbb{1}\left\{\mathcal{Q}(W, Z_{i,J_i}, \mu) = \text{`not in'}\right\} = r\right)$$

$$\times \mathbb{P}\left(\sum_{i\in[n]} \mathbb{1}\left\{\mathcal{Q}(W, Z_{i,J_i}, \mu) = \text{`not in'}\right\} = r\right)$$

$$+ \mathbb{P}\left(\mathcal{E}_1, \mathcal{E}_2\right)$$

$$\overset{(d)}{=} \sum_{r\in[nt, \lceil n(1-c_1)\rceil]} e^{-2r\left(\frac{nt}{r} - \frac{1}{2}\right)^2} \mathbb{P}\left(\sum_{i\in[n]} \mathbb{1}\left\{\mathcal{Q}(W, Z_{i,J_i}, \mu) = \text{`not in'}\right\} = r\right) + \mathbb{P}\left(\mathcal{E}_1, \mathcal{E}_2\right)$$

$$\leq \max_{r\in[nt, \lceil n(1-c_1)\rceil]} e^{-2r\left(\frac{nt}{r} - \frac{1}{2}\right)^2} + \mathbb{P}\left(\mathcal{E}_1, \mathcal{E}_2\right)$$

$$\overset{(e)}{=} e^{-2\lceil n(1-c_1)\rceil\left(\frac{nt}{\lceil n(1-c_1)\rceil} - \frac{1}{2}\right)^2} + \mathbb{P}\left(\mathcal{E}_1, \mathcal{E}_2\right)$$

$$\overset{(f)}{\leq} e^{-2\lceil n(1-c_1)\rceil\left(\frac{nt}{\lceil n(1-c_1)\rceil} - \frac{1}{2}\right)^2} + \mathbb{P}\left(\mathcal{E}_1\right) + \mathbb{P}\left(\mathcal{E}_2\right)$$

$$\leq e^{-2\lceil n(1-c_1)\rceil\left(\frac{nt}{\lceil n(1-c_1)\rceil} - \frac{1}{2}\right)^2} + \xi + 1 - q,$$

and we justify the main steps hereafter:

- $(a)$ holds since under the event $\mathcal{E}_1^c$, we have that $\forall i \in [n], \mathbb{1}\{\mathcal{Q}(W, Z_{i,J_i^c}, \mu) = \text{`in'}\} = 0$ and also whenever i) both $\mathcal{Q}(W, Z_{i,J_i^c}, \mu) = \text{`not in'}$ and $\mathcal{Q}(W, Z_{i,J_i}, \mu) = \text{`not in'}$, $\hat{J}_i$ is chosen as $U_i$ and hence the Hamming difference of the $i$'th coordinate is $\mathbb{1}\{U_i \neq J_i\}$, and ii) when $\mathcal{Q}(W, Z_{i,J_i^c}, \mu) = \text{`not in'}$ and $\mathcal{Q}(W, Z_{i,J_i}, \mu) = \text{'in'}$, $\hat{J}_i$ is chosen as $J_i$ and hence the Hamming difference of the $i$'th coordinate is 0.
- $(b)$ holds since under the event $\mathcal{E}_2^c$, we have that $\sum_{i\in[n]} \mathbb{1}\{\mathcal{Q}(W, Z_{i,J_i}, \mu) = \text{`not in'}\} \leq n(1 - c_1)$,
- $(c)$ holds since for $r < nt$, the probability is zero,
- $(d)$ holds by Hoeffding's inequality for the independent uniform random variables $\mathbb{1}\{U_i \neq J_i\}$ and since $nt > n(1 - c_1/2)/2 \geq r/2$ for $n$ sufficiently large,
- $(e)$ holds for $n$ large enough since,

$$\log\left(\max_{r\in[nt, \lceil n(1-c_1)\rceil]} e^{-2r\left(\frac{nt}{r} - \frac{1}{2}\right)^2}\right) = - \min_{r\in[nt, \lceil n(1-c_1)\rceil]} 2r\left(\frac{nt}{r} - \frac{1}{2}\right)^2$$

$$= - \min_{\frac{r}{nt}\in[1, \frac{\lceil n(1-c_1)\rceil}{nt}]} 2nt\frac{r}{nt}\left(\frac{nt}{r} - \frac{1}{2}\right)^2$$

$$= - 2nt \min_{x\in[1, \frac{\lceil n(1-c_1)\rceil}{nt}]} x\left(\frac{1}{x} - \frac{1}{2}\right)^2$$

$$= - 2nt \min_{x\in[1, \frac{\lceil n(1-c_1)\rceil}{nt}]} \left(\frac{1}{x} - 1 + \frac{x}{4}\right)$$

$$\overset{(*)}{=} - 2nt\left(\frac{nt}{\lceil n(1-c_1)\rceil} - 1 + \frac{\lceil n(1-c_1)\rceil}{4nt}\right)$$

$$= - 2\lceil n(1-c_1)\rceil\left(\frac{nt}{\lceil n(1-c_1)\rceil} - \frac{1}{2}\right)^2,$$

where $(*)$ is derived since i) for $n$ sufficiently large, $\frac{\lceil n(1-c_1)\rceil}{nt} = \frac{\lceil n(1-c_1)\rceil}{\lfloor\frac{n}{2}(1-\frac{c_1}{2})\rfloor}$ which is less than 2 for $n$ large, and ii) since $\left(\frac{1}{x} - 1 + \frac{x}{4}\right)$ is decreasing in the range $(0, 2]$,
- $(f)$ results from $\mathbb{P}\left(\mathcal{E}_1, \mathcal{E}_2\right) \leq \mathbb{P}\left(\mathcal{E}_1\right) + \mathbb{P}\left(\mathcal{E}_2\right)$.

Since for sufficiently large $n$, $e^{-\lceil n(1-c_1)\rceil \left(\frac{nt}{\lceil n(1-c_1)\rceil} - \frac{1}{2}\right)^2}$ (which converges to $e^{-\frac{nc_1^2}{8(1-c_1)}}$) gets sufficiently small, hence, if $\xi < q$, then $P_{e_t} < 1$. This completes the proof of **Claim ii)**, and hence of **Part i)**.

### G.1.2  Part ii.

Similarly to **Part i)** (Appendix G.1.1), we will prove the result by contradiction: assume that there exists an adversary for $\mathcal{A}$ such that

$$\mathbb{P}\left(\exists i \in [n]\colon \mathcal{Q}(W, Z_{i,J_i^c}, \mu) = \text{`in'}\right) \leq \xi,$$

and

$$\mathbb{P}\left(\sum_{i\in[n]} \mathbb{1}\{\mathcal{Q}(W, Z_{i,J_i}, \mu) = \text{`in'}\} \geq \alpha n\right) \geq q.$$

This also gives,

$$\mathbb{P}\left(\sum_{i\in[n]} \mathbb{1}\{\mathcal{Q}(W, Z_{i,J_i}, \mu) = \text{`not in'}\} \geq n(1-\alpha)\right) \leq 1 - q. \tag{50}$$

In our proof, we allow the adversary to be stochastic. We denote expectations and probabilities with respect to the adversary's randomness (which is independent of all other random variables) by $\mathbb{E}_{\mathcal{Q}}[\cdot]$ and $\mathbb{P}_{\mathcal{Q}}[\cdot]$, where needed. The main part of the proof relies on the following lemma, which we state below but prove later (in Appendix G.7) for better readability.

**Lemma 5.** *The following holds.*

$$\mathbb{E}_{W,\tilde{\mathbf{S}},\mathbf{J},\mathcal{Q}}\left[\sum_{i\in[n]} \left(\mathbb{1}\{\mathcal{Q}(W, Z_{i,J_i^c}, \mu) = \text{`not in'}\} - \mathbb{1}\{\mathcal{Q}(W, Z_{i,J_i}, \mu) = \text{`not in'}\}\right)\right] = o(n).$$

By Lemma 5, we have

$$\mathbb{E}_{W,\tilde{\mathbf{S}},\mathbf{J},\mathcal{Q}}\left[\sum_{i\in[n]} \mathbb{1}\{\mathcal{Q}(W, Z_{i,J_i^c}, \mu) = \text{` not in'}\}\right]$$

$$= o(n) + \mathbb{E}_{W,\tilde{\mathbf{S}},\mathbf{J},\mathcal{Q}}\left[\sum_{i\in[n]} \mathbb{1}\{\mathcal{Q}(W, Z_{i,J_i}, \mu) = \text{`not in'}\}\right]$$

$$\overset{(a)}{\leq} o(n) + n(1-q) + n(1-\alpha)q$$

$$= o(n) + n(1 - \alpha q),$$

where $(a)$ holds using (50) and $\sum_{i\in[n]} \mathbb{1}\{\mathcal{Q}(W, Z_{i,J_i}, \mu) = \text{`not in'}\} \leq n$.

Hence, using Markov's inequality,

$$\mathbb{P}\left(\sum_{i\in[n]} \mathbb{1}\{\mathcal{Q}(W, Z_{i,J_i^c}, \mu) = \text{`not in'}\} \geq n - m'\right) \leq \frac{o(n) + n(1-\alpha q)}{n - m'},$$

or equivalently,

$$\mathbb{P}\left(\sum_{i\in[n]} \mathbb{1}\{\mathcal{Q}(W, Z_{i,J_i^c}, \mu) = \text{`in'}\} \geq m'\right) \geq 1 - \frac{o(n) + n(1-\alpha q)}{n - m'}.$$

Hence, for any

$$q' \in (0, \alpha q), \quad m' = n - \frac{o(n) + n(1-\alpha q)}{1 - q'} = \frac{n(\alpha q - q' - o(1))}{1 - q'},$$

we have

$$\mathbb{P}\left(\sum_{i\in[n]} \mathbb{1}\{\mathcal{Q}(W, Z_{i,J_i^c}, \mu) = \text{`in'}\} \geq m'\right) \geq q'.$$

Hence, by varying $q'$ over the interval $(0, \alpha q)$, the ratio $m'/n$ changes asymptotically from 0 to $\alpha q$. In other words, if $n$ is sufficiently large, then for any

$$\epsilon \in (0, \alpha), \quad m' = \left(\frac{\epsilon}{1/q + \epsilon - \alpha}\right)n - o(n) = \Omega(n),$$

we have

$$\mathbb{P}\left(\sum_{i \in [n]} \mathbb{1}\left\{\mathcal{Q}(W, Z_{i, J_i^c}, \mu) = \text{`in'}\right\} \geq m'\right) \geq (\alpha - \epsilon)q.$$

This completes the proof of **Part ii)**.

## G.2   Proof of Theorem 6

To prove Theorem 6, we show that for any learning algorithm $\mathcal{A}: \mathcal{Z} \to \mathbb{R}^D$, the projected-quantized algorithm, defined as

$$\mathcal{A}^*(S_n) \triangleq \Theta\tilde{\mathcal{A}}(\Theta^\top \mathcal{A}(S_n)) = \Theta\hat{W},$$

satisfies equation 8 and

$$\mathsf{CMI}(\mu, \mathcal{A}^*(S_n)) \leq \mathbb{E}_\Theta\left[\mathsf{CMI}^\Theta(\mu, \mathcal{A}^*(S_n))\right] = o(n), \tag{51}$$

for any distribution $\mu$. Having shown this, applying Theorem 5 completes the proof.

Fix any arbitrary distribution $\mu$. Consider the construction of $\mathrm{JL}(d, c_w, \nu)$, described in Appendix F.1. It is shown in equation 26 that

$$\mathsf{CMI}^\Theta(\tilde{\mathbf{S}}, \tilde{\mathcal{A}}) \leq d\log\left(\frac{c_w + \nu}{\nu}\right),$$

which, together with the data-processing inequality, yield

$$\mathsf{CMI}^\Theta(\mu, \mathcal{A}^*(S_n)) \leq d\log\left(\frac{c_w + \nu}{\nu}\right). \tag{52}$$

Furthermore, similar to equation 36, where it is shown that

$$\mathbb{E}_{P_{S_n, W}P_\Theta P_{\hat{W}|\Theta^\top W}}\left[\mathrm{gen}(S_n, W) - \mathrm{gen}(S_n, \Theta\hat{W})\right] \leq \frac{2}{\sqrt{n}}\left(1 + \frac{2}{d}\right)^{1/4} e^{-\frac{0.21}{4}d(c_w^2-1)^2},$$

it can be shown that

$$\left|\mathbb{E}_{P_{S_n, W}P_\Theta P_{\hat{W}|\Theta^\top W}}\left[\mathrm{gen}(S_n, W) - \mathrm{gen}(S_n, \Theta\hat{W})\right]\right| \leq \frac{2}{\sqrt{n}}\left(1 + \frac{2}{d}\right)^{1/4} e^{-\frac{0.21}{4}d(c_w^2-1)^2}. \tag{53}$$

Plugging the choices

$$d = 500r\log(n), \quad c_w = 1.1, \quad \nu = 0.4,$$

in equation 52 and equation 53 result equation 51 and equation 8, which completes the proof.

## G.3   Proof of Lemma 1

If $m = 0$, then consider an adversary that always outputs $\mathcal{Q}(W, Z, \mu) = 0$, for any $Z \in \mathcal{Z}$.

In the following, we assume that $m = nm' \neq 0$. Let $V \in \{0, 1\}$ be a binary random variable, independent of all other random variables, such that $P(V = 0) = \alpha$. For example, if there exists a set $\mathcal{B} \subseteq \mathcal{W}$ such that $\mathcal{P}(W \in \mathcal{B}) = \alpha$, then the adversary can set $V = \mathbb{1}\{W \notin \mathcal{B}\}$.

Consider an adversary that first picks a random $V$. If $V = 0$, then for any $Z \in \mathcal{Z}$, it declares $\mathcal{Q}(W, Z, \mu) = 0$. Otherwise (*i.e.*, $V = 1$), it declares $\mathcal{Q}(W, Z, \mu) = 0$ with probability $r_n$ and $\mathcal{Q}(W, Z, \mu) = 1$ with probability $1 - r_n$, independently of $(W, Z, \mu)$.

If $V = 0$, the adversary never recalls $m$ samples with any positive probability

$$\mathbb{P}\left(\sum_{i \in [n]} \mathcal{Q}(W, Z_{i,1}, \mu) \geq m\right) = \mathbb{P}\left(\sum_{i \in [n]} \mathcal{Q}(W, Z_{i,1}, \mu) \geq m, V = 1\right)$$

$$= (1 - \alpha)\mathbb{P}\left(\sum_{i \in [n]} \mathcal{Q}(W, Z_{i,1}, \mu) \geq m\big|V = 1\right).$$

Moreover,

$$\mathbb{P}\left(\exists i \in [n] \colon \mathcal{Q}(W, Z_{i,0}, \mu) = 1\right) = \mathbb{P}\left(\exists i \in [n] \colon \mathcal{Q}(W, Z_{i,0}, \mu) = 1, V = 1\right)$$
$$= (1 - \alpha)\mathbb{P}\left(\exists i \in [n] \colon \mathcal{Q}(W, Z_{i,0}, \mu) = 1 \big| V = 1\right).$$

Using the above two relations, this adversary is $\xi$-sound and recalls $m$ samples with probability $q$ if, restricting to $V = 1$, the adversary is $\frac{\xi}{(1-\alpha)}$-sound and recalls $m$ samples with probability $\frac{q}{(1-\alpha)}$. For the adversary to be $\frac{\xi}{(1-\alpha)}$-sound given $V = 1$, we should have $\mathbb{P}\left(\forall i \in [n], \mathcal{Q}(W, Z_{i,0}, \mu) = 0\right) \geq 1 - \frac{\xi}{(1-\alpha)}$. Hence, this adversary is $\xi$-sound if and only if

$$r_n^n \geq 1 - \frac{\xi}{(1-\alpha)},$$

therefore,

$$r_n \geq \sqrt[n]{1 - \frac{\xi}{(1-\alpha)}}.$$

Next, when $V = 1$, to find the probability of recalling $m = nm'$ samples with probability $\frac{q}{(1-\alpha)}$, note that the probability of $\mathcal{Q}(W, Z_{i,1}, \mu) = 1$ is equal to $(1 - r_n)$. We consider two cases:

i. If $r_n = 0$, $\mathbb{P}\left(\sum_{i \in [n]} \mathcal{Q}(W, Z_{i,1}, \mu) \geq m | V = 1\right) = 1$.

ii. If $m' < 1 - r_n$, using Hoeffding's inequality, we have

$$\mathbb{P}\left(\sum_{i \in [n]} \mathcal{Q}(W, Z_{i,1}, \mu) \geq m | V = 1\right) \geq 1 - e^{-2n(m' + r_n - 1)^2}.$$

Considering these two cases separately,

i. We should find a value of $\alpha$ such that $\frac{q}{(1-\alpha)} \leq 1$ and $0 \geq \sqrt[n]{1 - \frac{\xi}{(1-\alpha)}}$. Both conditions are satisfied for $\alpha = 1 - \xi$, if $\xi \geq q$.

ii. It is sufficient to find a value for $r_n$ such that $m' < (1 - r_n)$, $r_n \geq \sqrt[n]{1 - \frac{\xi}{(1-\alpha)}}$ and $1 - e^{-2n(m' + r_n - 1)^2} \geq \frac{q}{(1-\alpha)}$. If, $1 - m' - \sqrt{\frac{1}{2n} \log\left(\frac{1}{1 - \frac{q}{(1-\alpha)}}\right)} \geq 0$, then let

$$r_n \triangleq 1 - m' - \sqrt{\frac{1}{2n} \log\left(\frac{1}{1 - \frac{q}{(1-\alpha)}}\right)}.$$

It satisfies the first condition and the recall condition. Lastly, the soundness condition is satisfied if for sufficiently large $n$, we have

$$\sqrt[n]{1 - \frac{\xi}{(1-\alpha)}} + \sqrt{\frac{1}{2n} \log\left(\frac{1}{1 - \frac{q}{(1-\alpha)}}\right)} + \frac{m}{n} \leq 1.$$

### G.4 Proof of Theorem 8

We prove the theorem and the comment after it, separately.

In the first case and to prove Theorem 8, we show that for every $r < 1$ there exists a projection matrix $\Theta \in \mathbb{R}^{D \times d}$ with $d = \lceil n^{2r-1} \rceil$, a Markov Kernel $P_{\hat{W}|\Theta^\top W}$ and a *compression algorithm* $\mathcal{A}_{\Theta,n}^* \colon \mathcal{Z}^n \to \mathbb{R}^d$, defined as $\mathcal{A}_{\Theta,n}^*(S_n) \triangleq \tilde{\mathcal{A}}(\Theta^\top \mathcal{A}(S_n)) = \hat{W}$, such that

$$\left|\mathbb{E}_{P_{S_n,W} P_{\hat{W}|\Theta^\top W}} \left[\text{gen}(S_n, W) - \text{gen}(S_n, \Theta\hat{W})\right]\right| = \mathcal{O}\left(n^{-r}\right),$$

and

$$\mathsf{CMI}(\mu, \mathcal{A}_{\Theta,n}^*) = o(n).$$

Having shown this, then applying Theorem 5 completes the proof.

In the second case, we show that for every $r \in \mathbb{R}$, there exist a projection matrix $\Theta \in \mathbb{R}^{D \times d}$ with $d = \lceil r \log(n) \rceil$, a Markov Kernel $P_{\hat{W}|\Theta^\top W}$ and a *compression algorithm* $\mathcal{A}^*_{\Theta,n} \colon \mathcal{Z}^n \to \mathbb{R}^d$, defined as $\mathcal{A}^*_{\Theta,n}(S_n) \triangleq \tilde{\mathcal{A}}(\Theta^\top \mathcal{A}(S_n)) = \hat{W}$, such that

$$\mathbb{E}_{P_{S_n,W} P_{\hat{W}|\Theta^\top W}} \left[ \text{gen}(S_n, W) - \text{gen}(S_n, \Theta \hat{W}) \right] = \mathcal{O}\left( n^{-r} \right),$$

and

$$\mathsf{CMI}(\mu, \mathcal{A}^*_{\Theta,n}) = o(n).$$

Having shown this, again applying Theorem 5 completes the proof.

Hence, it remains to show the existence of such projection matrices $\Theta \in \mathbb{R}^{D \times d}$, Markov Kernels $P_{\hat{W}|\Theta^\top W}$ and compression algorithms $\mathcal{A}^*_{\Theta,n} \colon \mathcal{Z}^n \to \mathbb{R}^d$, for each of the above cases.

### G.4.1 Case i.

Consider the construction of $\text{JL}(d, c_w, \nu)$, described in Appendix F.1. It is shown in equation 26 that

$$\mathsf{CMI}^\Theta(\tilde{\mathbf{S}}, \hat{\mathcal{A}}) = \leq d \log\left( \frac{c_w + \nu}{\nu} \right).$$

Hence, for any fixed $\Theta$,

$$\mathsf{CMI}^\Theta(\mu, \hat{\mathcal{A}}) = \leq d \log\left( \frac{c_w + \nu}{\nu} \right).$$

Now, let

$$\Delta(W, \Theta \hat{W}; S_n) := \text{gen}(S_n, W) - \text{gen}(S_n, \Theta \hat{W}).$$

We show that for any $r < 1$, letting

$$d = n^{2r-1}, \quad , c_w = 1.1, \quad , \nu = 0.4,$$

results in

$$\mathfrak{E}_1 \triangleq \mathbb{E}_\Theta \left[ \left| \mathbb{E}_{\hat{W}, W, S_n} \left[ \Delta(W, \Theta \hat{W}; S_n) \right] \right| \right] = \mathcal{O}\left( \frac{1}{n^r} \right). \tag{54}$$

Having shown this, it's easy to see that there exists a $\Theta$, for which simultaneously

$$\left| \mathbb{E}_{\hat{W}, W, S_n} \left[ \Delta(W, \Theta \hat{W}; S_n) \right] \right| = \mathcal{O}\left( \frac{1}{n^r} \right),$$

and

$$\mathsf{CMI}^\Theta(\mu, \hat{\mathcal{A}}) = o(n).$$

Fix this matrix $\Theta \in \mathbb{R}^{D \times d}$ and the Markov Kernel $P_{\hat{W}|\Theta^\top W}$ induced by that. Choosing the overall algorithms as $\mathcal{A}^*_{\Theta,n} \colon \mathcal{Z}^n \to \mathbb{R}^d$ completes the proof.

Hence, it remains to show that equation 54 holds. By equation 28, we have

$$\Delta(W, \Theta \hat{W}; S_n) = -\langle W, \bar{Z} \rangle + \langle \hat{W}, \Theta^\top \bar{Z} \rangle,$$

where $\bar{Z} \triangleq \mathbb{E}_{Z \sim \mu}[Z] - \frac{1}{n} \sum_{i=1}^n Z_i$. Recall that $\hat{W} = U + V_\nu$, where $\mathbb{E}_{V_\nu}[V_\nu] = 0$. Hence,

$$\begin{aligned}
\mathbb{E}_\Theta \left[ \left| \mathbb{E}_{\hat{W}, W, S_n} \left[ \Delta(W, \Theta \hat{W}; S_n) \right] \right| \right] &= \mathbb{E}_\Theta \left[ \left| \mathbb{E}_{W, S_n} \left[ -\langle W, \bar{Z} \rangle + \langle U, \Theta^\top \bar{Z} \rangle \right] \right| \right] \\
&\leq \mathbb{E}_{\Theta, W, S_n} \left[ \left| -\langle W, \bar{Z} \rangle + \langle U, \Theta^\top \bar{Z} \rangle \right| \right] \\
&\leq \mathbb{E}_{S_n, W} \mathbb{E}_\Theta \left[ \left| -\langle w, \bar{Z} \rangle + \langle U, \Theta^\top \bar{Z} \rangle \right| \right].
\end{aligned}$$

Combining above equation with equation 44 for $\phi(\bar{Z}) = \bar{Z}$ gives

$$\mathbb{E}_\Theta \left[ \left| \mathbb{E}_{\hat{W}, W, S_n} \left[ \Delta(W, \Theta \hat{W}; S_n) \right] \right| \right] \leq e^{-0.21 d (1 - c_w^2)^2} + \mathbb{E}_{S_n} \left[ \|\bar{Z}\| \right] \mathcal{O}\left( \frac{1}{\sqrt{d}} \right).$$

Next, we know by equation 35 that $\mathbb{E}[\|\bar{Z}\|] \leq \mathbb{E}[\|\bar{Z}^2\|]^{1/2} \leq \frac{2}{\sqrt{n}}$. Hence,

$$\mathbb{E}_\Theta \left[ \left| \mathbb{E}_{\hat{W}, W, S_n} \left[ \Delta(W, \Theta \hat{W}; S_n) \right] \right| \right] \leq e^{-0.21 d (1 - c_w^2)^2} + \mathcal{O}\left( \frac{1}{\sqrt{dn}} \right),$$

The proof is completed by letting

$$d = n^{2r-1}, \quad c_w = 1.1, \quad \nu = 0.4.$$

### G.4.2 Case ii.

Consider the construction of $\text{JL}(d, c_w, \nu)$, described in Appendix F.1. It is shown in equation 26 that

$$\text{CMI}^{\Theta}(\tilde{\mathbf{S}}, \hat{\mathcal{A}}) = \leq d \log \left( \frac{c_w + \nu}{\nu} \right).$$

Hence, for any fixed $\Theta$,

$$\text{CMI}^{\Theta}(\mu, \hat{\mathcal{A}}) = \leq d \log \left( \frac{c_w + \nu}{\nu} \right). \tag{55}$$

Furthermore, it is shown in equation 36 that

$$\mathbb{E}_{P_{S_n, W} P_{\Theta} P_{\hat{W} | \Theta^\top W}} \left[ \text{gen}(S_n, W) - \text{gen}(S_n, \Theta \hat{W}) \right] \leq \frac{2}{\sqrt{n}} \left( 1 + \frac{2}{d} \right)^{1/4} e^{-\frac{0.21}{4} d(c_w^2 - 1)^2}.$$

Hence, there exists at least one $\Theta$ for which

$$\mathbb{E}_{P_{S_n, W} P_{\hat{W} | \Theta^\top W}} \left[ \text{gen}(S_n, W) - \text{gen}(S_n, \Theta \hat{W}) \right] \leq \frac{2}{\sqrt{n}} \left( 1 + \frac{2}{d} \right)^{1/4} e^{-\frac{0.21}{4} d(c_w^2 - 1)^2}. \tag{56}$$

Choose this matrix $\Theta \in \mathbb{R}^{D \times d}$ and the Markov Kernel $P_{\hat{W} | \Theta^\top W}$ induced by that. Call the overall algorithms as $\mathcal{A}_{\Theta, n}^* \colon \mathcal{Z}^n \to \mathbb{R}^d$, with the choices

$$d = 500 r \log(n), \quad c_w = 1.1, \quad \nu = 0.4.$$

Plugging these constants in equation 55 and equation 56 completes the proof.

### G.5 Proof of Theorem 9

We first provide the proof of Theorem 9. The proof for the comment after the theorem, i.e., to show equation 14 instead of equation 13, then follows similarly to the below proof, in a similar manner shown in the Case ii part of the proof of Theorem 8.

To prove Theorem 9, we follow the Case i part of the proof of Theorem 8, with a slight modification: $\bar{Z}$ is replaced by $Z$, which results in convergence rates roughly $\sqrt{n}$ larger than the current ones. For the sake of completeness, we provide the proof.

Let

$$\Delta \mathcal{L}(W, \Theta \hat{W}) := \mathcal{R}(W) - \mathcal{R}(\Theta \hat{W}).$$

Following similarly to the Case i part of the proof of Theorem 8, it is sufficient to show that for any $r < 1/2$, letting

$$d = n^{2r}, \quad , c_w = 1.1, \quad , \nu = 0.4,$$

results in

$$\mathfrak{E}_1 \triangleq \mathbb{E}_{\Theta} \left[ \left| \mathbb{E}_{\hat{W}, W} \left[ \Delta \mathcal{L}(W, \Theta \hat{W}) \right] \right| \right] = \mathcal{O} \left( \frac{1}{n^r} \right). \tag{57}$$

Hence, it remains to show that equation 57 holds. We have

$$\Delta \mathcal{L}(W, \Theta \hat{W}) = -\mathbb{E}_{Z \sim \mu} \left[ \langle W, Z \rangle + \langle \Theta \hat{W}, Z \rangle \right]$$

$$= -\mathbb{E}_{Z \sim \mu} \left[ \langle W, Z \rangle + \langle \hat{W}, \Theta^\top Z \rangle \right]$$

$$= -\langle W, \mathbb{E}_{Z \sim \mu}[Z] \rangle + \langle \hat{W}, \Theta^\top \mathbb{E}_{Z \sim \mu}[Z] \rangle.$$

Denote $\tilde{z} \triangleq \mathbb{E}_{Z \sim \mu}[Z]$. Hence, since $\hat{W} = U + V_\nu$, where $\mathbb{E}_{V_\nu}[V_\nu] = 0$, we have

$$\mathbb{E}_{\Theta} \left[ \left| \mathbb{E}_{\hat{W}, W} \left[ \Delta \mathcal{L}(W, \Theta \hat{W}) \right] \right| \right] = \mathbb{E}_{\Theta} \left[ \left| \mathbb{E}_W \left[ -\langle W, \tilde{z} \rangle + \langle U, \Theta^\top \tilde{z} \rangle \right] \right| \right]$$

$$\leq \mathbb{E}_{\Theta, W} \left[ \left| \langle W, \tilde{z} \rangle - \langle U, \Theta^\top \tilde{z} \rangle \right| \right].$$

Combining the above equation with equation 44, and by replacing $\phi(\bar{Z})$ by $\tilde{z}$, gives

$$\mathbb{E}_{\Theta} \left[ \left| \mathbb{E}_{\hat{W}, W} \left[ \Delta \mathcal{L}(W, \Theta \hat{W}) \right] \right| \right] \leq e^{-0.21 d(c_w^2 - 1)^2} + \mathcal{O} \left( \frac{1}{\sqrt{d}} \right).$$

The proof is completed by letting

$$d = n^{2r}, \quad c_w = 1.1, \quad \nu = 0.4.$$

## G.6 Proof of Lemma 2

Consider the the $\mathrm{JL}(d, c_w, \nu)$ transformation described in Appendix F.1 with some $d \in \mathbb{N}^+$, $c_w \in \left[1, \sqrt{5/4}\right)$, and $\nu \in (0, 1]$. Recall that $\hat{W} = U + V_\nu$, where $V_\nu$ be a random variable that takes value uniformly over $\mathcal{B}_d(\nu)$ and

$$
U := \begin{cases} \Theta^\top w, & \text{if } \|\Theta^\top w\| \leq c_w, \\ \mathbf{0}_d, & \text{otherwise.} \end{cases}
$$

Let $\mathcal{E}$ be the event that $\|\Theta^\top w\| > c_w$ and denote by $\mathcal{E}^c$ the complementary event of $\mathcal{E}$. We have

$$
\begin{aligned}
\mathbb{E}_{\Theta, V_\nu}\left[\left\|\Theta \hat{W}\right\|^2\right] &\overset{(a)}{\geq} \mathbb{E}_\Theta\left[\|\Theta U\|^2\right] - \mathbb{E}_{\Theta, V_\nu}\left[\|\Theta V_\nu\|^2\right] \\
&\overset{(b)}{=} \mathbb{E}_\Theta\left[\|\Theta U\|^2\right] - \frac{D}{d}\mathbb{E}_{V_\nu}\left[\|V_\nu\|^2\right] \\
&\overset{(c)}{\geq} \mathbb{E}_\Theta\left[\|\Theta U\|^2\right] - \frac{D\nu^2}{d} \\
&\overset{(d)}{=} \mathbb{E}_\Theta\left[\left\|\Theta\Theta^\top w\right\|^2 \mathbb{1}\{\mathcal{E}^c\}\right] - \frac{D\nu^2}{d} \\
&= \mathbb{E}_\Theta\left[\left\|\Theta\Theta^\top w\right\|^2\right] - \mathbb{E}_\Theta\left[\left\|\Theta\Theta^\top w\right\|^2 \mathbb{1}\{\mathcal{E}\}\right] - \frac{D\nu^2}{d} \\
&\overset{(e)}{\geq} \mathbb{E}_\Theta\left[\left\|\Theta\Theta^\top w\right\|^2\right] - \mathbb{E}_\Theta\left[\left\|\Theta\Theta^\top w\right\|^4\right]^{1/2} \mathbb{E}_\Theta\left[\mathbb{1}\{\mathcal{E}\}\right]^{1/2} - \frac{D\nu^2}{d} \\
&\overset{(f)}{\geq} \mathbb{E}_\Theta\left[\left\|\Theta\Theta^\top w\right\|^2\right] - \mathbb{E}_\Theta\left[\left\|\Theta\Theta^\top w\right\|^4\right]^{1/2} e^{-0.1d(c_w^2-1)^2} - \frac{D\nu^2}{d} \\
&\overset{(g)}{=} \left(\frac{D + d + 1}{d}\right)\|w\|^2 - \sqrt{\frac{(D+d+3)(D+d+5)(d+2)}{d^3}}\|w\|^2 e^{-0.1d(c_w^2-1)^2} - \frac{D\nu^2}{d},
\end{aligned}
$$

where

- $(a)$ follows by the triangle inequality,
- $(b)$ follows by noting that each element of $\Theta$ is i.i.d. with distribution $\mathcal{N}(0, 1/d)$,
- $(c)$ holds since $V_\nu \in \mathcal{B}_d(\nu)$,
- $(d)$ is derived by the definition of $U$ and $\mathcal{E}$,
- $(e)$ follows using Cauchy-Schwarz inequality,
- $(f)$ is derived in equation 32,
- and $(g)$ followed by following relations

$$
\begin{aligned}
\mathbb{E}_\Theta\left[\left\|\Theta\Theta^\top w\right\|^2\right] &= \left(\frac{D + d + 1}{d}\right)\|w\|^2, \\
\mathbb{E}_\Theta\left[\left\|\Theta\Theta^\top w\right\|^4\right] &= \frac{(D+d+3)(D+d+5)(d+2)}{d^3}\|w\|^4,
\end{aligned}
$$

shown below.

*Proof of norm two.* Note that $\mathbb{E}_\Theta\left[\left\|\Theta\Theta^\top w\right\|^2\right]$ scales with $\|w\|$. Hence, it suffices to assume that $\|w\| = 1$. Next, first we show that $\mathbb{E}_\Theta\left[\left\|\Theta\Theta^\top w\right\|^2\right]$ is the same for any $w$ with $\|w\| = 1$.

For any $w \in \mathbb{R}^D$, there exists an orthonormal matrix $Q \in \mathbb{R}^{D \times D}$ such that $QQ^\top = I_D$ and $Qw = e_1 \triangleq [1, 0, 0, \cdots, 0]^\top$. This matrix can be constructed by letting the first row as $w^\top$, and choosing the other

rows orthogonal to $w^\top$. Next, by letting $\Theta' = Q\Theta$, we can write

$$
\begin{aligned}
\left\| \Theta\Theta^\top w \right\|^2 &= w^\top \Theta\Theta^\top \Theta\Theta^\top w \\
&= e_1^\top Q\Theta\Theta^\top \Theta\Theta^\top Q^\top e_1 \\
&= e_1^\top \Theta'\Theta'^\top \Theta'\Theta'^\top e_1 \\
&= \left\| \Theta'\Theta'^\top e_1 \right\|^2.
\end{aligned}
$$

The result follows by noting that $\mathbb{E}\left[ \left\| \Theta'\Theta'^\top e_1 \right\|^2 \right] = \mathbb{E}\left[ \left\| \Theta\Theta^\top e_1 \right\|^2 \right]$, since the distribution of $\Theta$ is rotationally invariant.

Hence, it is sufficient to compute $\mathbb{E}\left[ \left\| \Theta\Theta^\top e_1 \right\|^2 \right]$. Denote the elements of $\Theta$ by $\theta_{i,j}$, where $i \in [D]$, $j \in [d]$. Then, simple algebra gives

$$
\mathbb{E}\left[ \left\| \Theta\Theta^\top e_1 \right\|^2 \right] = \mathbb{E}\left[ \sum_{i\in[D]} \sum_{j,j'\in[d]^2} \theta_{i,j}\theta_{i',j'}\theta_{1,j}\theta_{1,j'} \right].
$$

We know that for $\theta \sim \mathcal{N}(0, 1/d)$,

$$
\mathbb{E}\left[ \theta^m \right] = 0 \text{ for odd } m, \quad \mathbb{E}\left[ \theta^2 \right] = \frac{1}{d}, \quad \mathbb{E}\left[ \theta^4 \right] = \frac{3}{d^2}.
$$

Then, it suffices to consider terms in the expansions that are non-zero, i.e. the terms where only even norms of each random variable appear. We consider all such cases:

1. $i \neq 1$: $D - 1$ choices

    1.1. $j = j'$: $d$ choices and and the expectation of each term equals $\frac{1}{d^2}$.

2. $i = 1$: 1 choice

    2.1. $j = j'$: $d$ choices and and the expectation of each term equals $\frac{3}{d^2}$.

    2.2. $j \neq j'$: $d(d - 1)$ choices and and the expectation of each term equals $\frac{1}{d^2}$.

Summing all terms and factorizing properly gives

$$
\begin{aligned}
\mathbb{E}\left[ \left\| \Theta\Theta^\top e_1 \right\|^2 \right] &= \mathbb{E}\left[ \sum_{i\in[D]} \sum_{j,j'\in[d]^2} \theta_{i,j}\theta_{i',j'}\theta_{1,j}\theta_{1,j'} \right] \\
&= \frac{D + d + 1}{d}.
\end{aligned}
$$

$\square$

*Proof of norm four.* Note that $\mathbb{E}_\Theta\left[ \left\| \Theta\Theta^\top w \right\|^4 \right]$ scales with $\|w\|$. Hence, it suffices to assume that $\|w\| = 1$. Next, similar to the proof of norm two, it can be shown that $\mathbb{E}_\Theta\left[ \left\| \Theta\Theta^\top w \right\|^4 \right]$ is the same for any $w$ with $\|w\| = 1$. Hence, it is sufficient to compute $\mathbb{E}\left[ \left\| \Theta\Theta^\top e_1 \right\|^4 \right]$. Denote the elements of $\Theta$ by $\theta_{i,j}$, where $i \in [D]$, $j \in [d]$. Then, simple algebra gives

$$
\mathbb{E}\left[ \left\| \Theta\Theta^\top e_1 \right\|^4 \right] = \mathbb{E}\left[ \sum_{i,i'\in[D]^2} \sum_{j_1,j_2,j_1',j_2'\in[d]^4} \theta_{i,j_1}\theta_{i,j_2}\theta_{i',j_1'}\theta_{i',j_2'}\theta_{1,j_1}\theta_{1,j_2}\theta_{1,j_1'}\theta_{1,j_2'} \right].
$$

We know that for $\theta \sim \mathcal{N}(0, 1/d)$,

$$
\mathbb{E}\left[ \theta^m \right] = 0 \text{ for odd } m, \quad \mathbb{E}\left[ \theta^2 \right] = \frac{1}{d}, \quad \mathbb{E}\left[ \theta^4 \right] = \frac{3}{d^2}, \quad \mathbb{E}\left[ \theta^6 \right] = \frac{15}{d^3}, \quad \mathbb{E}\left[ \theta^8 \right] = \frac{105}{d^4}.
$$

Then, it suffices to consider terms in the expansions that are non-zero, i.e. the terms where only even norms of each random variable appear. We consider all such cases:

1. $i = i' \neq 1$: $D - 1$ choices

  1.1. $j_1 = j_2 = j_1' = j_2'$: $d$ choices, and the expectation of each term equals $\frac{9}{d^4}$.
  1.2. Two of $(j_1, j_2, j_1', j_2')$ are the same, and two others as well, with a different value: $3d(d-1)$ choices, and the expectation of each term equals $\frac{1}{d^4}$.

  Hence, the sum of the expectation of the terms for this case equals:
  $$3d^{-3}(D-1)(d+2).$$

2. $i, i' \neq 1$ and $i \neq i'$: $(D-1)(D-2)$ choices

  2.1. $j_1 = j_2 = j_1' = j_2'$: $d$ choices, and the expectation of each term equals $\frac{3}{d^4}$.
  2.2. $j_1 = j_2$ and different from $j_1' = j_2'$: $d(d-1)$ choices and the expectation of each term equals $\frac{1}{d^4}$.

  Hence, the sum of the expectation of the terms for this case equals:
  $$d^{-3}(D-1)(D-2)(d+2).$$

3. $i = 1$ and $i' \neq 1$ or $i' = 1$ and $i \neq 1$: $2(D-1)$ choices

  3.1. $j_1 = j_2 = j_1' = j_2'$: $d$ choices, and the expectation of each term equals $\frac{15}{d^4}$.
  3.2. $j_1 = j_2$ and different from $j_1' = j_2'$: $d(d-1)$ choices and the expectation of each term equals $\frac{3}{d^4}$.
  3.3. $j_1$ different from $j_1' = j_2' = j_2$: $d(d-1)$ choices and the expectation of each term equals $\frac{3}{d^4}$.
  3.4. $j_2$ different from $j_1' = j_2' = j_1$: $d(d-1)$ choices and the expectation of each term equals $\frac{3}{d^4}$.
  3.5. $j_1 \neq j_2$ and both different from $j_1' = j_2'$: $d(d-1)(d-2)$ choices and the expectation of each term equals $\frac{1}{d^4}$.

  Hence, the sum of the expectation of the terms for this case equals:
  $$2d^{-3}(D-1)(15 + 9(d-1) + (d-1)(d-2) = 2d^{-3}(D-1)(d+2)(d+4).$$

4. $i = i' = 1$: 1 choice

  4.1. $j_1 = j_2 = j_1' = j_2'$: $d$ choices, and the expectation of each term equals $\frac{105}{d^4}$.
  4.2. Exactly three of the indices among $(j_1, j_2, j_1', j_2')$ are the same: $4d(d-1)$ choices and the expectation of each term equals $\frac{15}{d^4}$.
  4.3. Two of $(j_1, j_2, j_1', j_2')$ are the same, and two others as well, with a different value: $3d(d-1)$ choices, and the expectation of each term equals $\frac{9}{d^4}$.
  4.4. There are exactly two same indices among $(j_1, j_2, j_1', j_2')$: $6d(d-1)(d-2)$ choices and the expectation of each term equals $\frac{3}{d^4}$.
  4.5. All indices among $(j_1, j_2, j_1', j_2')$ are different: $d(d-1)(d-2)(d-3)$ choices and the expectation of each term equals $\frac{1}{d^4}$.

  Hence, the sum of the expectation of the terms for this case equals:
  $$d^{-3}\left(105 + 60(d-1) + 27(d-1) + 18(d-1)(d-2) + (d-1)(d-2)(d-3)\right)$$
  $$= d^{-3}(d+2)(d+4)(d+6).$$

Finally, summing all terms and factorizing properly gives

$$\mathbb{E}\left[\left\|\Theta\Theta^\top e_1\right\|^4\right] = \mathbb{E}\left[\sum_{i,i' \in [D]^2} \sum_{j_1, j_2, j_1', j_2' \in [d]^4} \theta_{i,j_1}\theta_{i,j_2}\theta_{i',j_1'}\theta_{i',j_2'}\theta_{1,j_1}\theta_{1,j_2}\theta_{1,j_1'}\theta_{1,j_2'}\right]$$
$$= \frac{(D+d+3)(D+d+5)(d+2)}{d^3}.$$

$\square$

## G.7    Proof of Lemma 5

To prove this lemma, we show the below stronger result:

$$\sum_{i\in[n]} \left| \mathbb{E}_{W,\tilde{\mathbf{S}}_{i,[2]},J_i,\mathcal{Q}} \left[ \mathbb{1}\left\{ \mathcal{Q}(W, Z_{i,J_i^c}, \mu) = \text{`in'} \right\} - \mathbb{1}\left\{ \mathcal{Q}(W, Z_{i,J_i}, \mu) = \text{`in'} \right\} \right] \right| = o(n),$$

which results also

$$\mathbb{E}_{W,\tilde{\mathbf{S}},\mathbf{J},\mathcal{Q}} \left[ \sum_{i\in[n]} \left( \mathbb{1}\left\{ \mathcal{Q}(W, Z_{i,J_i^c}, \mu) = \text{`not in'} \right\} - \mathbb{1}\left\{ \mathcal{Q}(W, Z_{i,J_i}, \mu) = \text{`not in'} \right\} \right) \right] = o(n).$$

For a given $(W, Z_{i,j})$, denote

$$P_{\mathcal{Q}}\left( \mathcal{Q}(W, Z_{i,j}, \mu) = \text{`in'} \right) = \mathbb{E}_{\mathcal{Q}}\left[ \mathbb{1}\left\{ \mathcal{Q}(W, Z_{i,j}, \mu) = \text{`in'} \right\} \right] \triangleq p\left( W, Z_{i,j} \right),$$

where the probability and expectation with respect to $\mathcal{Q}$ refer to the stochasticity of the adversary. Note that $p\left( W, Z_{i,j} \right)$ is a measurable function of $(W, Z_{i,j})$.

For $r \in \{0, 1, \ldots, 2^n - 1\}$, denote its binary representation as $r = (b_{r,1}, \ldots, b_{r,n})$, where $b_{r,i} \in \{0, 1\}$. Now, consider $2^n$ *auxiliary* estimators, indexed by $r \in \{0, 1, \ldots, 2^n - 1\}$ and defined as follows. The estimator $r$, for the $i$-th sample, by having access to $(W, Z_{i,0}, Z_{i,1})$ estimates $J_i$ as

$$\hat{J}_i = \begin{cases} 0, & \text{with probability } \frac{1 + (-1)^{b_{r,i}} p\left( W, Z_{i,0} \right) - (-1)^{b_{r,i}} p\left( W, Z_{i,1} \right)}{2}, \\ 1, & \text{with probability } \frac{1 - (-1)^{b_{r,i}} p\left( W, Z_{i,0} \right) + (-1)^{b_{r,i}} p\left( W, Z_{i,1} \right)}{2}. \end{cases}$$

Note that each of these estimators makes its estimations only by having access to $(W, Z_{i,0}, Z_{i,1})$.

Define the Hamming distance $d_H : \{0, 1\}^n \times \{0, 1\}^n \to [n]$ between binary vectors $\mathbf{J}$ and $\hat{\mathbf{J}}$ as

$$d_H\left( \mathbf{J}, \hat{\mathbf{J}} \right) = \sum_{i\in[n]} \mathbb{1}\{J_i \neq \hat{J}_i\}.$$

We now compute the expectation of $d_H(\mathbf{J}, \hat{\mathbf{J}})$ for the r-th estimator, *i.e.,* $\mathbb{E}_{W,\tilde{\mathbf{S}},\mathbf{J},\hat{\mathbf{J}}}\left[ d_H(\mathbf{J}, \hat{\mathbf{J}}) \right]$. Note that due to the symmetry of $\tilde{\mathbf{S}}$, we can only consider the case where $\mathbf{J} = (1, 1, \ldots, 1) := \mathbf{1}_n$.

$$
\begin{aligned}
\mathbb{E}_{W,\tilde{\mathbf{S}},\mathbf{J},\hat{\mathbf{J}}}\left[ d_H(\mathbf{J}, \hat{\mathbf{J}}) \right] &= \mathbb{E}_{W,\tilde{\mathbf{S}},\hat{\mathbf{J}}|\mathbf{J}=\mathbf{1}_n}\left[ d_H(\mathbf{1}_n, \hat{\mathbf{J}}) \right] \\
&= \sum_{i\in[n]} \mathbb{E}_{W,\tilde{\mathbf{S}}_{i,[2]},\hat{J}_i|J_i=1}\left[ d_H(1, \hat{J}_i) \right] \\
&= \sum_{i\in[n]} \mathbb{E}_{W,\tilde{\mathbf{S}}_{i,[2]}|J_i=1}\left[ \frac{1 + (-1)^{b_{r,i}} p\left( W, Z_{i,0} \right) - (-1)^{b_{r,i}} p\left( W, Z_{i,1} \right)}{2} \right] \\
&= \frac{n}{2} + \frac{1}{2} \sum_{i\in[n]} (-1)^{b_{r,i}} \mathbb{E}_{W,\tilde{\mathbf{S}}_{i,[2]}|J_i=1}\left[ p\left( W, Z_{i,0} \right) - p\left( W, Z_{i,1} \right) \right] \\
&= \frac{n}{2} + \frac{1}{2} \sum_{i\in[n]} (-1)^{b_{r,i}} \mathbb{E}_{W,\tilde{\mathbf{S}}_{i,[2]},J_i}\left[ p\left( W, Z_{i,J_i^c} \right) - p\left( W, Z_{i,J_i} \right) \right] \\
&= \frac{n}{2} + \frac{1}{2} \sum_{i\in[n]} (-1)^{b_{r,i}} \mathbb{E}_{W,\tilde{\mathbf{S}}_{i,[2]},J_i,\mathcal{Q}}\left[ \mathbb{1}\left\{ \mathcal{Q}(W, Z_{i,J_i^c}, \mu) = \text{`in'} \right\} \right. \\
&\qquad\qquad\qquad\qquad\qquad\qquad\qquad\qquad \left. - \mathbb{1}\left\{ \mathcal{Q}(W, Z_{i,J_i}, \mu) = \text{`in'} \right\} \right].
\end{aligned}
$$

Then, there exists an estimator $r^*$, for which

$$\mathbb{E}_{W,\tilde{\mathbf{S}},\mathbf{J},\hat{\mathbf{J}}}\left[ d_H(\mathbf{J}, \hat{\mathbf{J}}) \right] = \frac{n}{2} - \frac{1}{2} \sum_{i\in[n]} \left| \mathbb{E}_{W,\tilde{\mathbf{S}}_{i,[2]},J_i,\mathcal{Q}}\left[ \mathbb{1}\left\{ \mathcal{Q}(W, Z_{i,J_i^c}, \mu) = \text{`in'} \right\} - \mathbb{1}\left\{ \mathcal{Q}(W, Z_{i,J_i}, \mu) = \text{`in'} \right\} \right] \right|.$$

Now, suppose by contradiction that

$$\sum_{i\in[n]} \left| \mathbb{E}_{W,\tilde{\mathbf{S}}_{i,[2]},J_i,\mathcal{Q}}\left[ \mathbb{1}\left\{ \mathcal{Q}(W, Z_{i,J_i^c}, \mu) = \text{`in'} \right\} - \mathbb{1}\left\{ \mathcal{Q}(W, Z_{i,J_i}, \mu) = \text{`in'} \right\} \right] \right|,$$

is not $o(n)$. This means that there exists some $b_1 \in \mathbb{R}+$ and a sequence $\{a_i\}_{i\in\mathbb{N}}$ such that $\lim_{i\to\infty} a_i = \infty$ and limiting $n$ to this subsequence, we have

$$\sum_{i\in[n]} \left| \mathbb{E}_{W,\tilde{\mathbf{S}}_{i,[2]},J_i,\mathcal{Q}} \left[ \mathbb{1}\left\{ \mathcal{Q}(W,Z_{i,J_i^c},\mu) = \text{`in'}\right\} - \mathbb{1}\left\{ \mathcal{Q}(W,Z_{i,J_i},\mu) = \text{`in'}\right\} \right] \right| \geq 2b_1 n.$$

Without loss of generality, we can assume that $b_1 \in (0,1/4)$. Then, for the estimator $r^*$,

$$\mathbb{E}_{W,\tilde{\mathbf{S}},\mathbf{J},\hat{\mathbf{J}}}\left[ d_H(\mathbf{J},\hat{\mathbf{J}}) \right] \leq \frac{n(1-2b_1)}{2}.$$

Next, we use Fano's inequality with approximate recovery [59, Theorem 2]. Let $t = \frac{1}{n}\left\lfloor \frac{(1-b_1)n}{2}\right\rfloor$ and denote

$$P_{e_t} \triangleq \mathbb{P}\left( d_H\left(\mathbf{J},\hat{\mathbf{J}}\right) > nt \right),$$

$$N_{\hat{\mathbf{j}}} \triangleq \sum_{\mathbf{j}\in\{0,1\}^n} \mathbb{1}\left\{ d_H(\mathbf{j},\hat{\mathbf{j}}) \leq nt \right\}.$$

It is easy to not that $N_{\hat{\mathbf{j}}}$ is the same for all $\hat{\mathbf{j}} \in \{0,1\}^n$. Hence, the maximum over $\hat{\mathbf{j}}$ of $N_{\hat{\mathbf{j}}}$ is equal to $N_{\mathbf{1}_n}$. With these notations, we have

$$o(n) \overset{(a)}{\geq} \mathsf{I}(\mathbf{J};W|\tilde{\mathbf{S}})$$

$$\overset{(b)}{=} \mathsf{I}(\mathbf{J};W,\hat{\mathbf{J}}|\tilde{\mathbf{S}})$$

$$\overset{(c)}{\geq} \mathsf{I}(\mathbf{J};\hat{\mathbf{J}}|\tilde{\mathbf{S}})$$

$$\overset{(d)}{\geq} \mathsf{I}(\mathbf{J};\hat{\mathbf{J}})$$

$$\overset{(e)}{\geq} (1-P_{e_t})\log\left(\frac{2^n}{N_{\mathbf{1}_n}}\right) - \log(2)$$

$$\overset{(f)}{\geq} n(1-P_{e_t})(1-h_b(t)) - (1-P_{e_t})\log(3) - \log(2),$$

where $(a)$ is by construction of $W$ and as shown in the proof of Theorem 3, $(b)$ is derived since $\hat{\mathbf{J}}$ is a function of $(W,\tilde{\mathbf{S}})$, $(c)$ is derived due to positivity of the mutual information, $(d)$ is derived due to the below relations

$$\mathsf{I}(\mathbf{J};\hat{\mathbf{J}}|\tilde{\mathbf{S}}) = H(\mathbf{J}) - H(\mathbf{J}|\hat{\mathbf{J}},\tilde{\mathbf{S}}) \geq H(\mathbf{J}) - H(\mathbf{J}|\hat{\mathbf{J}}) = \mathsf{I}(\mathbf{J};\hat{\mathbf{J}}),$$

$(e)$ is derived using [59, Theorem 2], and $(f)$ is derived using the claim, proved later below, that $N_{\mathbf{1}_n} \leq c_3 2^{nh_b(t)}$ for some constant $c \in \mathbb{R}_+$ and for $n$ sufficiently large.

Note that $t = \frac{1}{n}\left\lfloor \frac{(1-b_1)n}{2}\right\rfloor < 1/2$ and as $n \to \infty$, $1-h_b(t)$ converges to the constant $1-h_b\left(\frac{1-b_1}{2}\right) > 0$. Hence, if we show that for sufficiently large $n$, $1-P_{e_t} > 1-b_2$, for some constant $b_2 \in (0,1)$, the contradiction is achieved. Since the left-hand side is of order $o(n)$, which is greater than the right-hand side, which is $\Omega(n)$, and the proof is complete.

Hence, it remains to show for $n$ sufficiently larg **i)** $N_{\mathbf{1}_n} \leq c_3 2^{nh_b(t)}$ for some constant $c \in \mathbb{R}_+$ and **ii)** $P_{e_t} < b_2$, for some constant $b_2 \in (0,1)$.

**Proof of Claim i)** This is shown in equation 49.

**Proof of Claim ii)** Using Markov's inequality, we have

$$P_{e_t} \triangleq \mathbb{P}\left( d_H\left(\mathbf{J},\hat{\mathbf{J}}\right) > nt \right)$$

$$\leq \frac{\mathbb{E}_{W,\tilde{\mathbf{S}},\mathbf{J},\hat{\mathbf{J}}}\left[ d_H(\mathbf{J},\hat{\mathbf{J}}) \right]}{nt}$$

$$\leq \frac{(1-2b_1)}{2t}$$

$$\leq \frac{1-2b_1}{1-b_1+1/n}$$

$$=1 - \frac{b_1 - \frac{1}{n}}{1 - b_1 + 1/n}$$

$$\leq b_2,$$

for some constant $b_2 \in (1/2, 1)$ and $n$ sufficiently large (or $a_i$ sufficiently large).

This completes the proof of the lemma.

# H    Proofs of Appendix D: Random subspace training algorithms

## H.1    Proof of Lemma 3

**Part i.** For $a = 0$,

$$g_{a,p}(x) = \frac{1}{\sqrt{2\pi}} e^{-\frac{x^2}{2}},$$

which is a standard Gaussian distribution. Hence, $h(g_{a,p}(x)) = \log(\sqrt{2\pi e})$ and $f(a, p) = 0$.

**Part ii.** The relation $f(a, p) = f(-a, p)$ is trivial since by the symmetry of the distribution $g_{a,p}$. To show the increasing behavior with respect to $a$, consider $0 \leq a' < a$ and some $p \in [0, 1]$. We show $f(a', p) < f(a, p)$. For $a > 0$, let

$$X_1 = Y_1 + Ja, \quad X_2 = \frac{1}{a} X_1 = \frac{1}{a} Y_1 + J,$$

where $Y_1 \sim \mathcal{N}(0, 1)$ is independent of $J \sim \text{Bern}(p)$. Then, it is easy to verify that

$$\mathsf{I}(X_2; J) = \mathsf{I}(X_1; J) = f(a, p). \tag{58}$$

Now let $\sigma \triangleq \sqrt{\left(\frac{a}{a'}\right)^2 - 1}$ and define

$$X_3 = X_2 + \frac{1}{a} Y_2 = \frac{1}{a}(Y_1 + Y_2) + J, \tag{59}$$

where $Y_2 \sim \mathcal{N}\left(0, \sigma^2\right)$ is independent of other random variables. Note that $Y_3 \triangleq \frac{a'(Y_1 + Y_2)}{a}$ is independent of $J$ and distributed according to $\mathcal{N}(0, 1)$. Hence, we can write

$$X_3 = \frac{1}{a'} Y_3 + J. \tag{60}$$

Now, we have

$$f(a, p) \overset{(a)}{=} \mathsf{I}(X_2, J)$$
$$\overset{(b)}{<} \mathsf{I}(X_3; J)$$
$$\overset{(c)}{=} f(a', p),$$

where $(a)$ follows from equation 58, $(b)$ from equation 59 and the strong data processing inequality, and $(c)$ from equation 60. This completes the proof of the strictly increasing behavior with respect to $a$ in the range $[0, \infty)$.

**Part iii.** Denote $Q_1(x) := \frac{1}{\sqrt{2\pi}} e^{-\frac{x^2}{2}}$ and $Q_2(x) := \frac{1}{\sqrt{2\pi}} e^{-\frac{(x-a)^2}{2}}$. Note that $g_{a,p}(x) = pQ_1(x) + (1 - p)Q_2(x)$. Hence, $h(g_{a,p}(x)) = -p\mathbb{E}_{Q_1}[\log(g_{a,p}(x))] - (1 - p)\mathbb{E}_{Q_2}[\log(g_{a,p}(x))]$. Now, considering the limit to infinity, we have

$$\lim_{a \to \infty} h(g_{a,p}(x)) = -p\mathbb{E}_{Q_1}[\log(pQ_1(x))] - (1 - p)\lim_{a \to \infty} \mathbb{E}_{Q_2}[\log((1 - p)Q_2(x))]$$

$$= -p\log(p) - (1 - p)\log(1 - p) - p\mathbb{E}_{Q_1}[\log(Q_1(x))] - (1 - p)\lim_{a \to \infty} \mathbb{E}_{Q_2}[\log(Q_2(x))]$$

$$\overset{(a)}{=} \log(2)h_b(p) + \frac{1}{2}\log(2\pi e),$$

where $(a)$ is deduced by noting that both $Q_1$ and $Q_2$ are Gaussian distributions with variance 1 and hence, their differential entropy is equal to $\frac{1}{2}\log(2\pi e)$.

This concludes that $\lim_{a \to \infty} f(a, p) = h_b(p)$.

**Part iv.** $f(a, p) = f(a, 1 - p)$ is trivial since by the symmetry of the distribution $g_{a,p}$.

To show the strictly increasing behavior with respect to $p$, consider $0 \le p_1 < p_2 \le 1/2$. Let

$$X_1 = Y + J_1 a,$$

where $Y \sim \mathcal{N}(0, 1)$ is independent of $J_1 \sim \text{Bern}(p_1)$. Then, due to Part ii,

$$I(X_1; J_1) = f(a, p_1) = h(g_{a,p}(x)) - \log(\sqrt{2\pi e}). \tag{61}$$

Moreover, note that

$$h(X_1) = h(g_{a,p_1}(x)) = h(g_{a,1-p_1}(x)). \tag{62}$$

Let $Z \sim \text{Bern}(q)$ be independent of other random variables for some $q \in (0, 1)$ that will be determined later. Let

$$X_2 \triangleq Y + V a,$$

where $V = |J_1 - Z|$. Note that $V \sim \text{Bern}(p_1 q + (1 - p_1)(1 - q))$ is independent of $Y$.

Now, on the one hand, we have

$$h(X_2|V) = h(Y) = h(X_1|J_1). \tag{63}$$

On the other hand,

$$
\begin{aligned}
h(X_2) &\overset{(a)}{>} h(X_2|Z) \\
&= h(X_2|Z = 0) q + h(X_2|Z = 1)(1 - q) \\
&= h(Y + |J_1|a) q + h(Y + |J_1 - 1|a)(1 - q) \\
&\overset{(b)}{=} h(Y + J_1 a) q + h(Y + J_1' a)(1 - q) \\
&\overset{(c)}{=} h(g_{a,p_1}(x)) q + h(g_{a,1-p_1}(x))(1 - q) \\
&\overset{(d)}{=} h(g_{a,p_1}(x))) \\
&\overset{(e)}{=} h(X_1),
\end{aligned}
$$

where $(a)$ is derived by strong data processing inequality and since $p_1 \in [0, 1/2)$ and $q \in (0, 1)$, $(b)$ is derived for $J' \sim \text{Bern}(1 - p_1)$ independent of $Y$, and steps $(c)$, $(d)$, $(e)$ are derived using equation 62.

Hence, combining equation 61, equation 63, and equation 63, we have

$$
\begin{aligned}
f(a, p_1) &= I(X_1; J_1) \\
&< I(X_2; V) \\
&= f(a, p_1 q + (1 - p_1)(1 - q)).
\end{aligned}
$$

The proof completes by find a $q \in [0, 1]$ such that $p_1 q + (1 - p_1)(1 - q) = p_2$. To show that such $q$ exist, first denote $e_{p_1}(q) := p_1 q + (1 - p_1)(1 - q)$. Now, note that $e_{p_1}(1) = p_1 < p_2$ and $e_{p_1}(0) = 1 - p_1 > \frac{1}{2} \ge p_2$. Hence, there exists a $q^* \in (0, 1)$ such that $e_p(q^*) = p_2$. This completes the proof of this part.

## H.2  Proof of Theorem 10

Recall that

$$V_t \triangleq \{i_{t,1}, \ldots, i_{t,b}\},$$

is the set of sample indices chosen at time $t \in [T]$, chosen independently of any other random variables. Hence,

$$\text{gen}(\mu, \mathcal{A}^{(d)}) = \mathbb{E}_{\mathbf{V}} \left[ \text{gen}(\mu, \mathcal{A}^{(d)}_{\mathbf{V}}) \right],$$

where $\mathcal{A}^{(d)}_{\mathbf{V}}$ is the algorithm $\mathcal{A}^{(d)}$ where the batch indices $\mathbf{V} = (V_1, \ldots, V_T)$ are used.

The proof consists of bounding each of the conditional mutual information terms

$$\mathsf{CMI}^{\Theta}_{\mathbf{V},i,\mathbf{J}_{-i}}(\tilde{\mathbf{S}}, W') \triangleq \mathsf{I}^{\tilde{\mathbf{S}},\mathbf{J}_{-i},\Theta}(\mathcal{A}^{(d)}_{\mathbf{V}}(\tilde{\mathbf{S}}_{\mathbf{J}}, \Theta); J_i), \qquad i \in [n],$$

and then using the bound 15 of Corollary 2, with $\hat{\mathcal{A}}^{(d)}_{\mathbf{V}} = \mathcal{A}^{(d)}_{\mathbf{V}}$ and $\epsilon = 0$.

It is sufficient then to show that for a fixed $\mathbf{V}$ and every fixed $i \in [n]$, we have that

$$\mathsf{CMI}^{\Theta}_{\mathbf{V},\mathbf{J}_{-i},i}(\tilde{\mathbf{S}}, W') \leq \sum_{t:\ i \in V_t} \mathbb{E}_{p_{t,i},\Delta_{t,i}}\left[f\left(\frac{\eta_t}{b\sigma_t}\Delta_{t,i}, p_{t,i}\right)\right], \tag{64}$$

where

$$\Delta_{t,i} \triangleq \left\| \nabla_{w'}\ell\left(\Theta W'_{t-1}, Z_{i,0}\right) - \nabla_{w'}\ell\left(\Theta W'_{t-1}, Z_{i,1}\right) \right\|,$$

$$p_{t,i} \triangleq \mathbb{P}\left(J_i = 0 \big| \tilde{\mathbf{S}}, \Theta, \mathbf{V}, \mathbf{J}_{-i}, W'_{t-1}, \{W'_r, W'_{r-1}: r < t, i \in V_r\}\right).$$

For a fixed $i \in [n]$, if $\{t: i \in V_t\}$ is an empty set, then the final model is independent of $J_i$ and hence $\mathsf{CMI}^{\Theta}_{\mathbf{V},i,\mathbf{J}_{-i}}(\tilde{\mathbf{S}}, W') = 0$, which completes the proof. Now, assume that this set is not empty. For ease of notation, suppose that

$$\{t: i \in V_t\} = \{t_1, \ldots, t_M\},$$

where $1 \leq t_1 < t_2 < \cdots < t_M \leq T$.

Then, for a fixed $\mathbf{V}$,

$$\mathsf{CMI}^{\Theta}_{\mathbf{V},\mathbf{J}_{-i},i}(\tilde{\mathbf{S}}, W') \triangleq \mathsf{I}^{\tilde{\mathbf{S}},\mathbf{J}_{-i},\Theta}(\mathcal{A}^{(d)}_{\mathbf{V}}(\tilde{\mathbf{S}}_{\mathbf{J}}, \Theta); J_i)$$

$$= \mathsf{I}^{\tilde{\mathbf{S}},\mathbf{J}_{-i},\Theta,\mathbf{V}}(W'_T; J_i)$$

$$\overset{(a)}{\leq} \mathsf{I}^{\tilde{\mathbf{S}},\mathbf{J}_{-i},\Theta,\mathbf{V}}(W'_{t_M}, W'_{t_M-1}, W'_{t_{M-1}}, W'_{t_{M-1}-1}, \cdots, W'_{t_1}, W'_{t_1-1}; J_i)$$

$$\overset{(b)}{=} \sum_{m \in [M]} \mathsf{I}^{\tilde{\mathbf{S}},\mathbf{J}_{-i},\Theta,\mathbf{V}}(W'_{t_m}, W'_{t_m-1}; J_i | W'_{t_{m-1}}, W'_{t_{m-1}-1}, \cdots, W'_{t_1}, W'_{t_1-1})$$

$$\overset{(c)}{=} \sum_{m \in [M]} \mathsf{I}^{\tilde{\mathbf{S}},\mathbf{J}_{-i},\Theta,\mathbf{V}}(W'_{t_m}; J_i | W'_{t_m-1}, W'_{t_{m-1}}, W'_{t_{m-1}-1}, \cdots, W'_{t_1}, W'_{t_1-1}),$$

where $(a)$ holds since by the data processing inequality $\mathsf{I}^{\tilde{\mathbf{S}},\mathbf{J}_{-i},\Theta,\mathbf{V}}(W'_T; J_i) \leq \mathsf{I}^{\tilde{\mathbf{S}},\mathbf{J}_{-i},\Theta,\mathbf{V}}(W'_{t_M}; J_i)$ and $\mathsf{I}^{\tilde{\mathbf{S}},\mathbf{J}_{-i},\Theta,\mathbf{V}}(W'_{t_M}; J_i) \leq \mathsf{I}^{\tilde{\mathbf{S}},\mathbf{J}_{-i},\Theta,\mathbf{V}}(W'_{t_M}, W'_{t_M-1}, W'_{t_{M-1}}, W'_{t_{M-1}-1}, \cdots, W'_{t_1}, W'_{t_1-1}; J_i)$ by the non-negativity of the mutual information, $(b)$ is derived using the chain rule for the mutual information and by using the convention that when $m = 1$, the conditioning part $\{W'_{t_{m-1}}, W'_{t_{m-1}-1}, \cdots, W'_{t_1}, W'_{t_1-1}\}$ is an empty set, and $(c)$ is derived since $\mathsf{I}^{\tilde{\mathbf{S}},\mathbf{J}_{-i},\Theta,\mathbf{V}}(W'_{t_m-1}; J_i |, W'_{t_{m-1}}, W'_{t_{m-1}-1}, \cdots, W'_{t_1}, W'_{t_1-1}) = 0$.

Consider a fixed value of $(W'_{t_m-1}, W'_{t_{m-1}}, W'_{t_{m-1}-1}, \cdots, W'_{t_1}, W'_{t_1-1})$ and let

$$\mathcal{F}_m \triangleq \left\{ \tilde{\mathbf{S}}, \Theta, \mathbf{V}, \mathbf{J}_{-i}, W'_{t_m-1}, W'_{t_{m-1}}, W'_{t_{m-1}-1}, \cdots, W'_{t_1}, W'_{t_1-1} \right\}.$$

Note that

$$p_{t_m,i} \triangleq \mathbb{P}\left(J_i = 0 \big| \tilde{\mathbf{S}}, \Theta, \mathbf{V}, \mathbf{J}_{-i}, W'_{t_m-1}, \{W'_r, W'_{r-1}: r < t_m, i \in V_r\}\right)$$

$$= \mathbb{P}\left(J_i = 0 \big| \mathcal{F}_m\right).$$

Hence, it is sufficient to show that

$$\mathsf{I}^{\mathcal{F}_m}(W'_{t_m}; J_i) \leq f\left(\frac{\eta_{t_m}}{b\sigma_{t_m}}\Delta_{t_m,i}, p_{t_m,i}\right). \tag{65}$$

Recall that

$$W'_{t_m} = \mathrm{Proj}\left\{W'_{t_m-1} - \eta_{t_m}\nabla_{w'}\widehat{\mathcal{R}}(V_{t_m}, \Theta W'_{t_m-1}) + \sigma_{t_m}\varepsilon_{t_m}\right\},$$

where $\widehat{\mathcal{R}}(V_{t_m}, W) \triangleq \frac{1}{b}\sum_{i' \in V_{t_m}} \ell\left(Z_{i',J_{i'}}, W\right)$. Denote

$$\widehat{\mathcal{R}}_{-i}(V_{t_m}, W) \triangleq \frac{1}{b} \sum_{\substack{i':\\ i' \in V_{t_m}, i' \neq i}} \ell\left(Z_{i',J_{i'}}, W\right).$$

Furthermore, denote

$$\tilde{W}_{t_m} \triangleq W'_{t_m-1} - \eta_{t_m} \nabla_{w'} \widehat{\mathcal{R}}(V_{t_m}, \Theta W'_{t_m-1}) + \sigma_{t_m} \varepsilon_{t_m}$$
$$= W'_{t_m-1} - \eta_{t_m} \nabla_{w'} \widehat{\mathcal{R}}_{-i}(V_{t_m}, \Theta W'_{t_m-1}) - \frac{\eta_{t_m}}{b} \nabla_{w'} \ell \left(Z_{i,J_i}, \Theta W'_{t_m-1}\right) + \sigma_{t_m} \varepsilon_{t_m},$$

where the last line holds since by assumption $i \in V_{t_m}$.

Using the data processing inequality, we have that

$$\mathsf{I}^{\mathcal{F}_m}(W'_{t_m}; J_i) \leq \mathsf{I}^{\mathcal{F}_m}(\tilde{W}_{t_m}; J_i).$$

Hence, it is sufficient to show that

$$\mathsf{I}^{\mathcal{F}_m}(\tilde{W}_{t_m}; J_i) \leq f\left(\frac{\eta_{t_m}}{b\sigma_{t_m}} \Delta_{t_m,i}, p_{t_m,i}\right). \tag{66}$$

Note that

$$\mathsf{I}^{\mathcal{F}_m}(\tilde{W}_{t_m}; J_i) = \mathsf{I}^{\mathcal{F}_m}(\tilde{W}_{t_m}/\sigma_{t_m}; J_i) = h^{\mathcal{F}_m}(\tilde{W}_{t_m}/\sigma_{t_m}) - h^{\mathcal{F}_m}(\tilde{W}_{t_m}/\sigma_{t_m} \big| J_i). \tag{67}$$

To compute each of the two terms in right-side of equation 67, first we derive the marginal and conditional distributions of $\frac{1}{\sigma_{t_m}} \tilde{W}_{t_m}$.

- Given $\mathcal{F}_m$ and given $J_i = 0$,

  $$\frac{1}{\sigma_{t_m}} \tilde{W}_{t_m} = \frac{1}{\sigma_{t_m}} W'_{t_m-1} - \frac{\eta_{t_m}}{\sigma_{t_m}} \nabla_{w'} \widehat{\mathcal{R}}_{-i}(V_{t_m}, \Theta W'_{t_m-1}) - \frac{\eta_{t_m}}{b\sigma_{t_m}} \nabla_{w'} \ell \left(Z_{i,0}, \Theta W'_{t_m-1}\right) + \varepsilon_{t_m}.$$

  Hence, given $\mathcal{F}_m$ and given $J_i = 0$, $\frac{1}{\sigma_{t_m}} \tilde{W}_{t_m}$ is distributed as

  $$\frac{1}{\sigma_{t_m}} \tilde{W}_{t_m} \sim \tilde{P}_0 \triangleq \mathcal{N}(\mu_0, \mathrm{I}_d), \tag{68}$$

  where

  $$\mu_0 \triangleq \frac{1}{\sigma_{t_m}} W'_{t_m-1} - \frac{\eta_{t_m}}{\sigma_{t_m}} \nabla_{w'} \widehat{\mathcal{R}}_{-i}(V_{t_m}, \Theta W'_{t_m-1}) - \frac{\eta_{t_m}}{b\sigma_{t_m}} \nabla_{w'} \ell \left(Z_{i,0}, \Theta W'_{t_m-1}\right).$$

- Similarly, given $\mathcal{F}_m$ and given $J_i = 1$, $\frac{1}{\sigma_{t_m}} \tilde{W}_{t_m}$ is distributed as

  $$\frac{1}{\sigma_{t_m}} \tilde{W}_{t_m} \sim \tilde{P}_1 \triangleq \mathcal{N}(\mu_1, \mathrm{I}_d), \tag{69}$$

  where

  $$\mu_1 \triangleq \frac{1}{\sigma_{t_m}} W'_{t_m-1} - \frac{\eta_{t_m}}{\sigma_{t_m}} \nabla_{w'} \widehat{\mathcal{R}}_{-i}(V_{t_m}, \Theta W'_{t_m-1}) - \frac{\eta_{t_m}}{b\sigma_{t_m}} \nabla_{w'} \ell \left(Z_{i,1}, \Theta W'_{t_m-1}\right).$$

- Lastly, since $\mathbb{P}\left(J_i = 0 \big| \mathcal{F}_m\right) = p_{t_m,i}$, then given $\mathcal{F}_m$, $\frac{1}{\sigma_{t_m}} \tilde{W}_{t_m}$ is distributed as

  $$\frac{1}{\sigma_{t_m}} \tilde{W}_{t_m} \sim \tilde{P} \triangleq p_{t_m,i} \tilde{P}_0 + (1 - p_{t_m,i}) \tilde{P}_1$$
  $$= p_{t_m,i} \mathcal{N}(\mu_0, \mathrm{I}_d) + (1 - p_{t_m,i}) \mathcal{N}(\mu_1, \mathrm{I}_d).$$

Now, we compute each of the two terms of $h^{\mathcal{F}_m}(\tilde{W}_{t_m}/\sigma_{t_m})$ and $h^{\mathcal{F}_m}(\tilde{W}_{t_m}/\sigma_{t_m} | J_i)$:

- The term $h^{\mathcal{F}_m}(\tilde{W}_{t_m}/\sigma_{t_m})$ equals the differential entropy $h(\tilde{P})$. Since the differential entropy is invariant under the shift and since also the Gaussian distributions $\tilde{P}_0$ and $\tilde{P}_1$ are invariant under the rotation, $h(\tilde{P})$ is equal to the entropy of the distribution $\tilde{Q}$, defined as

  $$\mathbf{Q} \triangleq p_{t_m,i} \mathcal{N}(\mathbf{0}_d, \mathrm{I}_d) + (1 - p_{t_m,i}) \mathcal{N}(\mathbf{a}_d, \mathrm{I}_d),$$

  where

  $$\mathbf{a}_d = \left(\frac{\eta_{t_m}}{b\sigma_{t_m}} \mu, 0, 0, \cdots, 0\right) \in \mathbb{R}^d,$$

and

$$\mu \triangleq \frac{b\sigma_{t_m}}{\eta_{t_m}} \|\mu_1 - \mu_0\|$$
$$= \left\| \nabla_{w'} \ell \left( \Theta W'_{t_m-1}, Z_{i,0} \right) - \nabla_{w'} \ell \left( \Theta W'_{t_m-1}, Z_{i,1} \right) \right\|$$
$$= \Delta_{t_m,i}.$$

Note that $\|\mathbf{a}_d\| = \|\mu_1 - \mu_0\|$.

Furthermore, we can write

$$\mathbf{Q} = Q_1 \otimes Q_2 \otimes \cdots Q_d, \tag{70}$$

where

$$Q_1 = p_{t_m,i} \mathcal{N}(0,1) + (1 - p_{t_m,i}) \mathcal{N} \left( \frac{\eta_{t_m}}{b\sigma_{t_m}} \Delta_{t_m,i}, 1 \right),$$

and for $r \in \{2, 3, \ldots, d\}$,

$$Q_i = \mathcal{N}(0,1).$$

Hence,

$$h^{\mathcal{F}_m}(\tilde{W}_{t_m}/\sigma_{t_m}) = h(\tilde{P})$$
$$= h(\mathbf{Q})$$
$$\overset{(a)}{=} \sum_{r \in [d]} h(Q_r)$$
$$\overset{(b)}{=} h(Q_1) + (d-1)\log(\sqrt{2\pi e})$$
$$\overset{(c)}{=} h\left( g_{a_1, p_{t_m,i}}(x) \right) + (d-1)\log(\sqrt{2\pi e})$$
$$\overset{(d)}{=} f\left( \frac{\eta_{t_m}}{b\sigma_{t_m}} \Delta_{t_m,i}, p_{t_m,i} \right) + d\log(\sqrt{2\pi e}), \tag{71}$$

where $(a)$ is derived by equation 70, $(b)$ holds since the distributions $Q_2, \ldots, Q_d$ are scalar standard Gaussian distributions, $(c)$ is derived for $a_1 \triangleq \frac{\eta_{t_m}}{b\sigma_{t_m}} \Delta_{t_m,i}$ and by the definition of $g_{a,p}(\cdot)$ in 19, and $(d)$ by the definition of $f(a,p)$ in 18.

- To compute $h^{\mathcal{F}_m}(\tilde{W}_{t_m}/\sigma_{t_m}|J_i)$, note that for each value of $J_i$, due to equation 68 and equation 69, the conditional distribution of $\frac{1}{\sigma_{t_m}} \tilde{W}_{t_m}$ is a multivariate Gaussian distribution with covariance $\mathrm{I}_d$. Hence,

$$h^{\mathcal{F}_m}(\tilde{W}_{t_m}/\sigma_{t_m}|J_i) = d\log(\sqrt{2\pi e}). \tag{72}$$

Combining equation 71 and equation 72 gives equation 66 which completes the proof.

### H.3 Proof of Theorem 13

Recall that

$$W'_t = \mathrm{Proj}\left\{ W'_{t-1} - \eta_t \nabla_{w'} \widehat{\mathcal{R}}(V_t, \Theta W'_{t-1}) + \sigma_t \varepsilon_t \right\},$$

where $\widehat{\mathcal{R}}(V_t, W) \triangleq \frac{1}{b} \sum_{i \in V_t} \ell(Z_{i,J_i}, W)$.

In the proof, to define the lossy compression algorithm $P_{\hat{W}|W',\Theta,S}$ of Corollary 2, we introduce auxiliary optimization iterations $\left\{ \hat{W}_t \right\}_{t \in [T]}$, as follows. Let $\hat{W}_0 = W'_0$, and for $t \in [T]$, let

$$\hat{W}_t = \mathrm{Proj}\left\{ \hat{W}_{t-1} - \eta_t \nabla_{\hat{w}} \widehat{\mathcal{R}}(V_t, \Theta \hat{W}_{t-1}) + \sigma_t \varepsilon_t + \nu_t \varepsilon'_t \right\}, \tag{73}$$

where $\varepsilon'_t \sim \mathcal{N}(\mathbf{0}_d, \mathbf{I}_d)$ is an additional noise, independent of all other random variables.

In the following Lemma, proved in Appendix H.4, we show that, this choice of $P_{\hat{W}|W',\Theta,S,\mathbf{V}}$ satisfies the distortion term equation 16:

$$\mathbb{E}_{P_S P_\Theta P_{\mathbf{V}} P_{W'_T|S,\Theta,\mathbf{V}} P_{\hat{W}_T|W'_T,S,\Theta,\mathbf{V}}} \left[ \mathrm{gen}(S, \Theta W'_T) - \mathrm{gen}(S, \Theta \hat{W}_T) \right] \leq \epsilon,$$

for

$$\epsilon := \frac{2\sqrt{2}\mathfrak{L}\,\Gamma((d+1)/2)}{\Gamma(d/2)} \sum_{t \in [T]} \alpha^{T-t} \nu_t.$$

**Lemma 6.** *The following inequalities holds:*

$$\left\| \hat{W}_t - W_t' \right\| \le \sum_{r \in [t]} \alpha^{t-r} \nu_r \left\| \varepsilon_r' \right\|,$$

*and*

$$\mathbb{E}_{P_S P_\Theta P_{\mathbf{V}} P_{W_T'|S,\Theta,\mathbf{V}} P_{\hat{W}_T|W_T',S,\Theta,\mathbf{V}}} \left[ \mathrm{gen}(S, \Theta W_T') - \mathrm{gen}(S, \Theta \hat{W}_T) \right] \le \frac{2\sqrt{2}\mathfrak{L}\,\Gamma((d+1)/2)}{\Gamma(d/2)} \sum_{t \in [T]} \alpha^{T-t} \nu_t.$$

Hence, it is sufficient to show that

$$\mathbb{E}_{P_S P_\Theta P_{\mathbf{V}} P_{\hat{W}_T|S,\Theta,\mathbf{V}}} \left[ \mathrm{gen}(S, \Theta \hat{W}_T) \right]$$

$$\le \frac{C\sqrt{2}}{n} \sum_{i \in [n]} \mathbb{E}_{\tilde{\mathbf{S}},\Theta,\mathbf{V},\mathbf{J}_{-i}} \left[ \sqrt{ \sum_{t \,:\, i \in V_t} A_{t,i} \mathbb{E}_{\hat{p}_{t,i},\hat{\Delta}_{t,i}} \left[ f\left( \frac{\eta_t}{b\sqrt{\sigma_t^2 + \nu_t^2}} \hat{\Delta}_{t,i}, \hat{p}_{t,i} \right) \right] } \right].$$

Note that the iterations defined in equation 73 are equivalent in distribution to the following iterations:

$$\hat{W}_t = \mathrm{Proj}\left\{ \hat{W}_{t-1} - \eta_t \nabla_{\hat{w}} \widehat{\mathcal{R}}(V_t, \Theta \hat{W}_{t-1}) + \hat{\sigma}_t \tilde{\varepsilon}_t \right\},$$

where $\tilde{\varepsilon}_t \sim \mathcal{N}(\mathbf{0}_d, \mathbf{I}_d)$ is independent of all other random variables and

$$\hat{\sigma}_t \triangleq \sqrt{\sigma_t^2 + \nu_t^2}.$$

Similar to the proof of Theorem 10, and by using Corollary 2, it is sufficient to show that for a fixed $\mathbf{V}$ and every fixed $i \in [n]$, we have that

$$\mathsf{CMI}_{\mathbf{V},\mathbf{J}_{-i},i}^{\Theta}(\tilde{\mathbf{S}}, \hat{W}_T) \le \sum_{t \,:\, i \in V_t} A_{t,i} \mathbb{E}_{\hat{p}_{t,i},\hat{\Delta}_{t,i}} \left[ f\left( \frac{\eta_t}{b\hat{\sigma}_t} \hat{\Delta}_{t,i}, \hat{p}_{t,i} \right) \right],$$

where

$$\mathsf{CMI}_{\mathbf{V},\mathbf{J}_{-i},i}^{\Theta}(\tilde{\mathbf{S}}, \hat{W}_T) \triangleq I^{\tilde{\mathbf{S}},\mathbf{J}_{-i},\Theta,\mathbf{V}}(\hat{W}_T; J_i),$$

$$\hat{\Delta}_{t,i} \triangleq \left\| \nabla_{w'}\ell\left( \Theta \hat{W}_{t-1}, Z_{i,0} \right) - \nabla_{w'}\ell\left( \Theta \hat{W}_{t-1}, Z_{i,1} \right) \right\|,$$

$$\hat{p}_{t,i} \triangleq \mathbb{P}\left( J_i = 0 \big| \tilde{\mathbf{S}}, \Theta, \mathbf{V}, \mathbf{J}_{-i}, \hat{W}_{t-1} \right),$$

$$A_{t,i} := \prod_{r \in [t+1:T] \,:\, i \notin V_r} q_r.$$

For a fixed $i \in [n]$, if $\{t \,:\, i \in V_t\}$ is an empty set, then the final model is independent of $J_i$ and hence $\mathsf{CMI}_{\mathbf{V},i,\mathbf{J}_{-i}}^{\Theta}(\tilde{\mathbf{S}}, \hat{W}_T) = 0$, which completes the proof. Now, assume that this set is not empty. For ease of notation, suppose that

$$\{t \,:\, i \in V_t\} = \{t_1, \ldots, t_M\},$$

where $1 \le t_1 < t_2 < \cdots < t_M \le T$.

We show by induction on $m \in [M]$ that, we have

$$\mathsf{CMI}_{\mathbf{V},\mathbf{J}_{-i},i}^{\Theta}(\tilde{\mathbf{S}}, \hat{W}_{t_m}) \le \sum_{k \le m} A_{t_k,i}^{t_m} \mathbb{E}_{\hat{p}_{t_k,i},\hat{\Delta}_{t_k,i}} \left[ f\left( \frac{\eta_{t_k}}{b\hat{\sigma}_{t_k}} \hat{\Delta}_{t_k,i}, \hat{p}_{t_k,i} \right) \right], \tag{74}$$

where $\hat{\Delta}_{t,i}$ and $\hat{p}_{t,i}$ are defined as above and

$$A_{t,i}^{t'} := \prod_{r \in [t+1:t'] \,:\, i \notin V_r} q_r,$$

with the convention that $A_{t,i}^{t'} = 1$ for $t' \le t$.

Once this claim is shown, then we have

$$\mathsf{I}^{\tilde{\mathbf{S}},\mathbf{J}_{-i},\Theta,\mathbf{V}}(\hat{W}_T; J_i) \overset{(a)}{\leq} \left( \prod_{r=t_M+1}^{T} q_r \right) \mathsf{I}^{\tilde{\mathbf{S}},\mathbf{J}_{-i},\Theta,\mathbf{V}}(\hat{W}_{t_M}; J_i)$$

$$\overset{(b)}{\leq} \left( \prod_{r=t_M+1}^{T} q_r \right) \sum_{k \leq M} A_{t_k,i}^{t_M} \mathbb{E}_{\hat{p}_{t_k,i}, \hat{\Delta}_{t_k,i}} \left[ f \left( \frac{\eta_{t_k}}{b\hat{\sigma}_{t_k}} \hat{\Delta}_{t_k,i}, \hat{p}_{t_k,i} \right) \right]$$

$$\overset{(b)}{=} \sum_{k \leq M} A_{t_k,i} \mathbb{E}_{\hat{p}_{t_k,i}, \hat{\Delta}_{t_k,i}} \left[ f \left( \frac{\eta_{t_k}}{b\hat{\sigma}_{t_k}} \hat{\Delta}_{t_k,i}, \hat{p}_{t_k,i} \right) \right],$$

where

- $(a)$ is achieved by repeated using of [74, Lemma 4],
- $(b)$ is derived using equation 74,
- and $(c)$ holds by definitions of $A_{t_k,i} = A_{t_k,i}^{T}$ and $A_{t_k,i}^{t_M}$.

Hence, it remains to show that equation 74 holds by induction.

Consider the base of the induction $m = 1$. Note that $A_{t_1,i}^{t_1} = 1$. Hence, the result follows using the proof of Theorem 10; more precisely using equation 64 with $W' \to \hat{W}_{t_1}$, $\Delta_{t,i} \to \hat{\Delta}_{t,i}$, $p_{t,i} \to \hat{p}_{t,i}$, and $\sigma_t \to \hat{\sigma}_t$.

Now, suppose that the result holds for $m = N \leq M - 1$, *i.e.,*

$$\mathsf{CMI}_{\mathbf{V},\mathbf{J}_{-i},i}^{\Theta}(\tilde{\mathbf{S}}, \hat{W}_N) \leq \sum_{r \in [N]} A_{t_r,i}^{t_N} \mathbb{E}_{\hat{p}_{t_r,i}, \hat{\Delta}_{t_k,i}} \left[ f \left( \frac{\eta_{t_r}}{b\hat{\sigma}_{t_r}} \hat{\Delta}_{t_r,i}, \hat{p}_{t_r,i} \right) \right], \tag{75}$$

where

$$A_{t_r,i}^{t_N} := \prod_{t \in [t_r+1:t_N]\,:\, i \notin V_t} q_t.$$

We show that it also holds for $m = N + 1 \leq M$.

We have

$$\mathsf{I}^{\tilde{\mathbf{S}},\mathbf{J}_{-i},\Theta,\mathbf{V}}(\hat{W}_{t_{N+1}}; J_i) \leq \mathsf{I}^{\tilde{\mathbf{S}},\mathbf{J}_{-i},\Theta,\mathbf{V}}(\hat{W}_{t_{N+1}}, \hat{W}_{t_{N+1}-1}; J_i)$$

$$= \mathsf{I}^{\tilde{\mathbf{S}},\mathbf{J}_{-i},\Theta,\mathbf{V}}(\hat{W}_{t_{N+1}}; J_i | \hat{W}_{t_{N+1}-1}) + \mathsf{I}^{\tilde{\mathbf{S}},\mathbf{J}_{-i},\Theta,\mathbf{V}}(\hat{W}_{t_{N+1}-1}; J_i)$$

$$\overset{(a)}{\leq} \mathbb{E}_{\hat{p}_{t_{N+1},i}} \left[ f \left( \frac{\eta_{t_{N+1}}}{b\hat{\sigma}_{t_{N+1}}} \hat{\Delta}_{t_{N+1},i}, \hat{p}_{t_{N+1},i} \right) \right] + \mathsf{I}^{\tilde{\mathbf{S}},\mathbf{J}_{-i},\Theta,\mathbf{V}}(\hat{W}_{t_{N+1}-1}; J_i)$$

$$\overset{(b)}{\leq} \mathbb{E}_{\hat{p}_{t_{N+1},i}} \left[ f \left( \frac{\eta_{t_{N+1}}}{b\hat{\sigma}_{t_{N+1}}} \hat{\Delta}_{t_{N+1},i}, \hat{p}_{t_{N+1},i} \right) \right]$$

$$+ \left( \prod_{r=t_N+1}^{t_{N+1}-1} q_r \right) \mathsf{I}^{\tilde{\mathbf{S}},\mathbf{J}_{-i},\Theta,\mathbf{V}}(\hat{W}_{t_N}; J_i)$$

$$\overset{(c)}{\leq} \mathbb{E}_{\hat{p}_{t_{N+1},i}} \left[ f \left( \frac{\eta_{t_{N+1}}}{b\hat{\sigma}_{t_{N+1}}} \hat{\Delta}_{t_{N+1},i}, \hat{p}_{t_{N+1},i} \right) \right]$$

$$+ \left( \prod_{r=t_N+1}^{t_{N+1}-1} q_r \right) \sum_{r \in [N]} A_{t_r,i}^{t_N} \mathbb{E}_{\hat{p}_{t_r,i}} \left[ f \left( \frac{\eta_{t_r}}{b\hat{\sigma}_{t_r}} \hat{\Delta}_{t_r,i}, \hat{p}_{t_r,i} \right) \right]$$

$$\overset{(d)}{=} \sum_{r \in [N+1]} A_{t_r,i}^{t_{N+1}} \mathbb{E}_{\hat{p}_{t_r,i}} \left[ f \left( \frac{\eta_{t_r}}{b\hat{\sigma}_{t_r}} \hat{\Delta}_{t_r,i}, \hat{p}_{t_r,i} \right) \right],$$

where

- $(a)$ is derived using the proof of Theorem 10; more precisely using equation 65 with $W'_{t_m} \to \hat{W}_{t_{N+1}}$, $\Delta_{t_m,i} \to \hat{\Delta}_{t_{N+1},i}$, $p_{t_m,i} \to \hat{p}_{t_{N+1},i}$, $\sigma_{t_m} \to \hat{\sigma}_{t_{N+1}}$, and by considering

$$\mathcal{F}_m \to \left\{ \tilde{\mathbf{S}}, \Theta, \mathbf{V}, \mathbf{J}_{-i}, W'_{t_{N+1}-1} \right\}.$$

- $(b)$ is derived by repeated using of [74, Lemma 4],
- $(c)$ holds by the assumption of the induction 75,
- and $(d)$ by definition of $A_{t,i}^{t_N}$ and $A_{t,i}^{t_{N+1}}$.

This completes the proof of the theorem.

## H.4 Proof of Lemma 6

To prove the result, we show first what

$$\left\| \hat{W}_t - W_t' \right\| \leq \sum_{r \in [t]} \alpha^{t-r} \nu_r \left\| \varepsilon_r' \right\|, \tag{76}$$

using induction over $t \in [T]$. Then, using the Lipschitzness property of the loss function, we have that

$$\mathbb{E}_{P_S P_\Theta P_\mathbf{V} P_{W_T'|S,\Theta,\mathbf{V}} P_{\hat{W}_T|W_T',S,\Theta,\mathbf{V}}} \left[ \mathrm{gen}(S, \Theta W_T') - \mathrm{gen}(S, \Theta \hat{W}_T) \right] \leq 2 \mathfrak{L} \mathbb{E} \left[ \sum_{r \in [T]} \alpha^{T-r} \nu_r \left\| \varepsilon_r' \right\| \right]$$

$$\overset{(a)}{=} \frac{2\sqrt{2} \mathfrak{L} \, \Gamma((d+1)/2)}{\Gamma(d/2)} \sum_{r \in [T]} \alpha^{T-r} \nu_r,$$

where $(a)$ is obtained using the fact that if $\mathfrak{Z} \sim \mathcal{N}(0, \mathrm{I}_d)$, then $\|\mathfrak{Z}\|$ has a chi-distribution, whose mean is equal to $\sqrt{2} \frac{\Gamma((d+1)/2)}{\Gamma(d/2)}$.

For $t = 1$,

$$\left\| \hat{W}_1 - W_1' \right\| \overset{(a)}{\leq} \left\| \left( W_0' - \eta_1 \nabla_{w'} \widehat{\mathcal{R}}(V_1, \Theta W_0') + \sigma_1 \varepsilon_1 + \nu_1 \varepsilon_1' \right) - \left( W_0' - \eta_1 \nabla_{w'} \widehat{\mathcal{R}}(V_1, \Theta W_0') + \sigma_1 \varepsilon_1 \right) \right\|$$

$$= \nu_1 \left\| \varepsilon_1' \right\|,$$

where $(a)$ is derived since for any $w_1', w_2' \in \mathbb{R}^d$, $\left\| \mathrm{Proj}\left\{ w_1' \right\} - \mathrm{Proj}\left\{ w_2' \right\} \right\| \leq \left\| w_1' - w_2' \right\|$, by the contraction property of the projection. This shows the base of the induction.

Suppose that equation 76 holds for $t = t'$. Now, we show that it also holds for $t = t' + 1$.

$$\left\| \hat{W}_{t'+1} - W_{t'+1}' \right\| \leq \left\| \left( \hat{W}_{t'} - \eta_{t'+1} \nabla_{w'} \widehat{\mathcal{R}}(V_{t'+1}, \Theta \hat{W}_{t'}) + \sigma_{t'+1} \varepsilon_{t'+1} + \nu_{t'+1} \varepsilon_{t'+1}' \right) \right.$$

$$\left. - \left( W_{t'}' - \eta_{t'+1} \nabla_{w'} \widehat{\mathcal{R}}(V_{t'+1}, \Theta W_{t'}') + \sigma_{t'+1} \varepsilon_{t'+1} \right) \right\|$$

$$= \left\| \left( \hat{W}_{t'} - \eta_{t'+1} \nabla_{w'} \widehat{\mathcal{R}}(V_{t'+1}, \Theta \hat{W}_{t'}) \right) - \left( W_{t'}' - \eta_{t'+1} \nabla_{w'} \widehat{\mathcal{R}}(V_{t'+1}, \Theta W_{t'}') \right) \right.$$

$$\left. + \nu_{t'+1} \varepsilon_{t'+1}' \right\|$$

$$\overset{(a)}{\leq} \left\| \left( \hat{W}_{t'} - \eta_{t'+1} \nabla_{w'} \widehat{\mathcal{R}}(V_{t'+1}, \Theta \hat{W}_{t'}) \right) - \left( W_{t'}' - \eta_{t'+1} \nabla_{w'} \widehat{\mathcal{R}}(V_{t'+1}, \Theta W_{t'}') \right) \right\|$$

$$+ \left\| \nu_{t'+1} \varepsilon_{t'+1}' \right\|$$

$$\overset{(b)}{\leq} \alpha \left\| \hat{W}_{t'} - W_{t'}' \right\| + \nu_{t'+1} \left\| \varepsilon_{t'+1}' \right\|$$

$$\overset{(c)}{\leq} \alpha \sum_{r \in [t']} \alpha^{t'-r} \nu_r \left\| \varepsilon_r' \right\| + \nu_{t'+1} \left\| \varepsilon_{t'+1}' \right\|$$

$$= \sum_{r \in [t'+1]} \alpha^{(t'+1)-r} \nu_r \left\| \varepsilon_r' \right\|,$$

where $(a)$ is derived using the triangle inequality, $(b)$ using the contractility assumption, and $(c)$ using the assumption of the induction. This completes the proof of the induction and the proof of the lemma.

