# OpenReview forum: "Tighter CMI-Based Generalization Bounds via Stochastic Projection and Quantization"
_NeurIPS.cc/2025/Conference — NeurIPS 2025 oral_

### Official Review · Reviewer_UwxD · 2025-06-26

**Clarity:** 2
**Significance:** 3
**Originality:** 2
**Rating:** 4
**Confidence:** 3

**Summary:**

This paper aims at improving the well-established conditional mutual information (CMI) generalization bounds. The main idea is to apply the CMI bounds to a compressed parameter that has been obtained via stochastic projection. In this context, the obtained generalization bounds is the sum of two terms: (1) a loss compression constant, measuring how the generalization errors of the compressed and uncompressed learned parameters differ, and (2) a CMI bound on a lower-dimensional space, on which the parameter has been projected. In order to show that this strategy yields tighter generalization bounds, the authors apply it to several counter-examples of the classical CMI bounds, that have been recently proposed by Livni (2023) and Attias et al. (2024). In these classes of simple optimization problems, the authors obtain better convergence and sample complexity, hence, beating the proposed counter-examples. The last part of the paper is dedicated to the application of lossy compression and stochastic projection to the problem of memorization, ie, the ability of an adversary to guess which data points have been used in training. For one of the problem classes studied above, it is shown that these techniques allow to construct algorithms that are robust to memorization and, yet, have similar generalization error as their uncompressed counterpart.

**Questions:**

- In most of the counter-examples, the constructed data distribution seems to depend on $n$, which seems restrictive compared to the practice. Are there counter-examples for which it is not the case and where you new bound are tighter? In particular, an example in the style of Section 4.1 would be more convincing with data distributions that do not depend on $n$.
- In section 5, you find alternate learning algorithms with comparable generalization error but which do not memorize the dataset. Is it also possible to show the same alternate algorithm have similar empirical risk, ie, do they have similar overall performance (population risk)? Indeed, this is waht practitioners really care about.
- Is it possible to demonstrate that the new bounds are tightest on a more complex problem class than the ones that are studied in the main text? The same quesion holds for memorization in Section 5.

**Ethical Concerns:**

["NO or VERY MINOR ethics concerns only"]

**Final Justification:**

During the rebuttal, most of the points raised in the review were addressed by the authors. In particular there were discussions about the counter-examples involving data distributions that depend on the sample size. I understand that this is something that has been previously introduced in the literature and it is not the aim of this paper to address this particular point.

I still have a positive opinion on the paper and I will keep my score.

**Limitations:**

The limitations could be discussed more precisely. For instance, adding a conclusion paragraph to summarize the limitations of the work would be helpful.

**Quality:**

3

**Strengths And Weaknesses:**

**Strengths:**

- The proposed approach is very general. While stochastic projection has already been used in mutual information bounds, the authors successfully apply it the CMI framework and combine it with lossy compression.
- The new framework addresses the counter-examples of the CMI bounds found in earlier studies. Therefore, the proposed bounds are arguable tighter.
- In addition to providing tighter generalization bounds, the same techniques (stochastic projection and lossy compression) can be used to address the vulnerability to memorization, at least in simple examples.
- The literature review in the introduction is quite complete.


**Weaknesses:**

- Despite addressing several known counter examples, the interest of the new bounds is only assessed on very simple settings (linear, quandratic).
- In the abstract, applications to randomsubspace algorithms, SGD and SGLD are mentioned but these are not discussed in the main text.
- The main paper ends abruptly, in particular there is no conclusion to discuss the potential limitations and future works.

**Other remarks**

- I think some notations have not been introduced, for instance: $M_1(Z)$. Also $A^{(d)}$ has not been defined in theorem 1, is it a typo?
- When the data distribution $\mu$ depends on $n$, maybe it should be denoted $\mu_n$, to make it explicit.

---

> ### Author Rebuttal · Authors · 2025-07-30
>
> Thank you for your comments and positive rating.
>
> **Weaknesses**
>
> > "Despite addressing several known counter examples, the interest of the new bounds is only assessed on very simple settings (linear, quandratic)."
>
> We appreciate the opportunity to clarify the motivation and contribution of our paper. Our work focuses on stochastic convex optimization (SCO), a commonly used framework in theoretical machine learning, in particular for rigorously analyzing generalization in overparameterized models. SCO encompasses classical algorithms such as (stochastic) gradient descent, and is not limited to linear or quadratic losses.
>
> Following [42, 43], we considered linear/quadratic losses in order to derive fully explicit and tight bounds. These setups are simple yet nontrivial, and allow for rigorous derivations that reveal concrete insights into generalization, compressibility, and memorization. This strategy allows us to precisely characterize certain phenomena and illustrate our main theorems, which are of the form: *there exists an instance in the class of SCO such that the generalization bound exhibits a specific behavior.* These examples should therefore be considered as concrete instances where our analysis and discussion apply, rather than as restrictions of the general SCO framework.
>
> That said, linear losses are not the only ones we considered: in Section 4.4, we go beyond them by analyzing generalized linear models, which form a larger, nonlinear class. Given that computations and proofs are already nontrivial in the setings we cover, exploring richer nonlinear instances remains an interesting yet challenging research direction.
>
> > "In the abstract, applications to random subspace algorithms, SGD and SGLD are mentioned but these are not discussed in the main text."
>
> Thank you for pointing this out. We will remove the last sentence of the abstract to clarify the main text's content, but we will still refer to these applications at the end of our introduction.
>
> > "The main paper ends abruptly"
>
> We will add the following paragraph as a conclusion section at the end of our paper.
>
> *"In this work, we revisit recent limitations identified in conditional mutual information-based generalization bounds. By incorporating stochastic projections and lossy compression mechanisms into the CMI framework, we derive bounds that remain informative in stochastic convex optimization, thereby offering a new perspective on the results in [42,43]. Our approach also provides a constructive resolution to the memorization phenomenon described in [43], by showing that for any algorithm and data distribution, one can construct an alternative model that does not trace training data while achieving comparable generalization.*
>
> ***Limitations and open questions.** Like prior work on information-theoretic bounds, our analysis applies to stochastic convex optimization. A natural, open question is whether and how these results can be extended to more general learning settings. Another key direction is to translate our theoretical findings into actionable design principles for learning algorithms with controlled generalization and compressibility."*
>
> **Other remarks**
>
> > "I think some notations have not been introduced, for instance: $M_1(Z)$."
>
> Thank you for noting this: $M_1(Z)$ denotes the set of probability measures on $Z$. We will precise it in Definition 3.
>
> > "Also $A^{(d)}$ has not been defined in theorem 1, is it a typo?"
>
> This is indeed a typo: the correct notation is $\mathcal{A}$.
>
> > "When the data distribution $\mu$ depends on $n$, maybe it should be denoted $\mu_n$"
>
> For readability, we chose to omit this dependence in the notation. We acknowledge this may cause some confusion and will thus clarify this choice at the beginning of Section 3.
>
> **Questions**
>
> > the constructed data distribution seems to depend on $n$, which seems restrictive
>
> We agree that this is a relevant question, and note that this restriction is not specific to our work. To the best of our knowledge, there is no counter-example in the literature where the data distribution and dimensionality do not depend on $n$. This dependence seems crucial in existing approaches, including in Attias et al. (2024), where counter-examples involve (C\)MI terms that grow with $D$. Intuitively, the higher $D$, the more likely one finds a coordinate that makes the training and test data distinguishable. Therefore, to obtain bounds that shrinks with $n$ increasing, it is typically necessary for $D$ to grow faster than $n$.
>
> > Is it also possible to show the same alternate algorithm have similar empirical risk
>
> We can indeed provide additional guarantees on the population risk of the projected-quantized model in Theorem 7. Based on your feedback, we were able to show that for any $r< 1/2$, there exists an auxiliary model $\Theta \hat{W}$ that cannot trace data and such that
> $$|\mathbb{E}\_{P_{S_n,W}P_{\hat{W} | \Theta^T W}}[\mathcal{L}(W) - \mathcal{L}(\Theta \hat{W})] | \leq \mathcal{O}(n^{-r}),$$
> which is achieved with $d=n^{2r}$.
>
> Alternatively, for any $r\in[R]$, we can guarantee the existence of an auxiliary model $\Theta \hat{W}$ that cannot trace data and satisfies
> $$\mathbb{E}\_{P_{S_n,W}P_{\hat{W} | \Theta^T W}}[\mathcal{L}(W) - \mathcal{L}(\Theta \hat{W})] \leq \mathcal{O}(n^{-r}),$$
> achieved for $d=r\log(n)$.
>
> The proof closely follows that of Theorem 7, with a slight modification: $\bar{Z}$ is replaced by $Z$, which results in convergence rates roughly $\sqrt{n}$ larger than the current ones. For example, in the first case, this strategy yields
> $$\mathbb{E}\_{\Theta}\big[\big|\mathbb{E}\_{\hat{W}, W, S_n}[\mathcal{L}(W)-\mathcal{L}(\Theta \hat{W})] \big|\big] \leq e^{-0.21 d(1-c_w^2)^2} + \mathcal{O}(d^{-1/2}),$$
> which is in $\mathcal{O}(n^{-r})$ for $d=n^{2r}$. Furthermore, since $r<1/2$,
> $$\mathsf{CMI}^{\Theta}({\mu},\hat{\mathcal{A}}) \leq d \log\left(\frac{c_w+\nu}{\nu}\right) = o(n)$$
> which completes the proof.
>
> We thank the reviewer for raising this interesting question, which has helped us extending our results. We will incorporate the above discussion in the paper.
>
> > "Is it possible to demonstrate that the new bounds are tightest on a more complex problem class?"
>
> This is a natural and interesting question. Currently, we have not formally established that our bounds are tightest for more complex problem classes beyond those studied in the main text, and we view this as a direction for future work.
>
> That said, despite its simple characterization, we emphasize that stochastic convex optimization already captures relevant aspects for overparameterized models. It also enables nontrivial analytical derivations and encompasses standard algorithms in modern ML (e.g., stochastic gradient descent and accelerated variants). These explain its popularity in theoretical ML research. We refer to [42, 43] for detailed explanations and references.
>
> Regarding memorization (Section 5), we also focused on SCO to allow clear theoretical analysis and to align our setting with that of [43], since our primary goal was to demonstrate the value of incorporating compression in their analysis. Extending both analyses ([43] and ours) to more complex function classes, such as non-convex losses or deep networks, would indeed be a valuable extension.

---

> > ### Comment · Reviewer_UwxD · 2025-08-02
> > **Thank you for your answer**
> >
> > Thank you for your detailed answer, which clarifies some of my concerns, especially regarding the considered problem classes.
> >
> > I am still curious about the fact that the constructed data distributions depend on the sample size. Do you think this can be related to practical scenarios?
> > Additionally, are there results (like lower bounds) showing that this is necessary? If that is the case, does it mean that the theory does not apply when the data distribution is fixed and we let the sample size increase?
> >
> > Thank you again for your answer.

---

> > > ### Author Response · Authors · 2025-08-02
> > >
> > > Thank you for your feedback on our rebuttal and the follow-up question.
> > >
> > > Please note that:
> > >
> > > - The condition that the dimension grows with the sample size '$n$' as $D > n^4 \log n $ was introduced by Attias et al. [2024], not us. We are not aware of any work that has shown that this is actually a necessary condition for classic CMI bounds to fail.  What is shown in Attias et al. [2024] is that this is a sufficient condition for classic CMI bounds to fail for the convex optimization problem instances that are carefully constructed therein. It would indeed be interesting to find (if any) counter-examples with fixed $D$ for which classic CMI bounds fail. We will add a few lines of discussion on this aspect in the paper.
> > >  - Regarding the practicality of the condition that $D$ grows with '$n$':  this condition is indeed not common in practical scenarios.  However, one could interpret the result of Attias et al. [2024] differently, in a manner that perhaps makes it more useful in practice:  In a sense, that result says that if $D$ is fixed but is so chosen so large that it exceeds $n^4 \log n$, then classic CMI bounds are guaranteed to be non-meaningful in that case. (Our results, which do not contradict Attias et al., show that even in such a case one can still find a suitable projection and generalization-preserving lossy compression which, when applied to the original model, yields a projected-compressed model for which the application of the CMI framework is still meaningful).
> > >
> > > - As for the lower bounds on the generalization, this is a domain that has attracted less attention (than upper bounds) in ML research; and actually very few lower bounds exist that are tailored for specific cases or algorithms. In that case, they are usually used to prove impossibility results, as in Livni [42] (which has provided a lower bound on an upper bound). Lower bounds that apply in general seldom exist, and this area has so far attracted no attention, perhaps because it is somehow assumed that the generalization can be arbitrarily small or even zero (e.g., by simply overfitting to the training data).
> > >
> > > Also, just for a better clarity, we would also like to make it clearer that:
> > > - Our generalization bounds results apply in general, irrespective of whether the dimension (data distribution) increases with the sample size or is set to be fixed.
> > > - They are generally tighter than classic CMI bounds, in both cases (when the dimension increases with the sample size and when it is fixed). This is because the classic CMI bound can be obtained as a special case of our results, by setting $\Theta$ to be the identity matrix and the distortion $\epsilon$ is set to be zero.
> > > - In the specific case in which the dimension increases with the sample size as in the examples of Attias et al. [2024] and Livni [2023], the classic CMI bounds fail as shown therein, while our results are shown (in our present paper) to successfully capture the behavior of the generalization error.
> > >
> > > We hope this clarifies the issue. If not, please let us know and we will hope to elaborate further.

---

> > > > ### Comment · Reviewer_UwxD · 2025-08-03
> > > > **Thank you for your answer**
> > > >
> > > > Thank you for further clarifying this point and add some discussion in the paper.
> > > >
> > > > For the moment, I will keep my score, pending discussion with the other reviewers.

---

> > > > > ### Author Response · Authors · 2025-08-04
> > > > >
> > > > > Thank you for your feedback; we are happy that our response has clarified the issue.

---

### Official Review · Reviewer_9s6T · 2025-06-30

**Clarity:** 3
**Significance:** 3
**Originality:** 3
**Rating:** 5
**Confidence:** 4

**Summary:**

This paper presents tighter conditional mutual information (CMI) bounds by using random projection and lossy compression. The motivation is to address known limitations of previous CMI-based generalization bounds in certain stochastic convex optimization (SCO) problems. Specifically, previous CMI bounds fail to explain the learnability of some SCO settings because the CMI term scales explicitly with the dimension $D$ of the hypothesis space, while the actual generalization error in these problems can be dimension-independent. As a result, in overparameterized regimes (e.g., $D>n$), such CMI bounds do not vanish as the sample size n increases.

To overcome this issue, the authors propose compressing the hypothesis through random projection and lossy encoding. This leads to new generalization bounds where the error is upper bounded by the CMI of the compressed model, plus an additional term quantifying the generalization gap between the compressed and original models. Although the CMI term still depends on the compressed dimension $d$, this dimension corresponds to a subspace of the original hypothesis space and can be chosen arbitrarily, e.g., $d=1,\Theta(\sqrt{n}), n^{1-r}$, to ensure that the bound vanishes with n. Consequently, the proposed CMI bounds successfully capture the learnability of SCO problems that elude explanation under standard CMI analysis.

Moreover, the authors apply their bounds to the memorization game and demonstrate that the compressed models achieve strong generalization without exhibiting memorization.

**Questions:**

1. While I generally enjoyed reading this paper and believe it deserves acceptance, I do have some concerns regarding the claim about the "inherent limitation" of the CMI framework. Specifically, is it possible to directly bound the generalization error of $\mathcal{P}_{cvx}^{(D)}$ without relying on the CMI framework? My concern is that bounding the $\epsilon$ term in your result seems to require a level of effort comparable to directly bounding the generalization error itself. If this is indeed the case, then although the dimensional dependence of the CMI term is now controllable via compression, the practical utility of your CMI bound remains unclear without an explicit control over $\epsilon$, which again seems to require similar efforts on bounding the generalization gap. In this sense, is it really accurate to claim (as in Line 54) that "the aforementioned limitations are not inherent to the CMI framework"?

Put differently, while this work successfully mitigates the dimensional dependency of the CMI term, the framework still relies on the additional control of the generalization gap between the compressed and uncompressed models. Since this requires nontrivial analysis, one might argue that the fundamental limitation of CMI, as a generalization measure, still persists, albeit in a more manageable form.

2. Regarding Theorem 4, would it be possible to provide the order-wise behavior of the actual generalization error in a concrete example? This would help clarify whether the $\mathcal{O}(1/n^{1/4})$  bound is sufficiently tight and informative as a generalization measure in practice.

3. Please clarify how Theorem 7 supports the claim that the auxiliary model $\Theta\hat{W}$ is not vulnerable to memorization. In particular, part (ii) of the theorem seems unrelated to the auxiliary model, and it is unclear how the conclusion follows from the stated result.

**Ethical Concerns:**

["NO or VERY MINOR ethics concerns only"]

**Final Justification:**

I recommend accepting this paper without reservation.

**Limitations:**

Yes, the authors mention certain limitations in the checklist. I would like to raise an additional point: the proposed CMI bounds are primarily designed for parameterized models. In contrast to works such as [2–4], the framework does not naturally extend to non-parameterized models, which may be considered a potential limitation.

**Quality:**

4

**Strengths And Weaknesses:**

**Strengths:**

1. The main strength of this paper is straightforward: the proposed CMI bounds successfully address the counterexamples raised in previous works on SCO, where earlier CMI-based generalization bounds fail to vanish.

2. The paper provides an insightful result that, for a learning algorithm using compression, memorization might not be necessary to achieve strong generalization performance.

**Weaknesses:**

1. Regarding Theorem 7, it is unclear why the auxiliary (compressed) model is considered not vulnerable to memorization.  In particular, part (ii) of the theorem appears unrelated to the auxiliary model $\Theta\hat{W}$.

2. The following are not significant weaknesses but rather suggestions for improvement:

The clarity of some claims, particularly in the introduction, could be improved. For example, the first listed contribution states: "we use them to establish a new CMI-based bound that is generally tighter than existing ones." However, "existing ones" are not specified, and Section 3 does not formally justify the claim that the proposed bounds are generally tighter than all previous CMI variants before discussing the counterexamples. Notably, several other CMI-based bounds, such as chained CMI bounds [1], evaluation-based bounds [2–4], are not shown to fail on the counterexamples from Attias et al. (2024) and Livni (2023). While Theorem 1 is clearly tighter than classic hypothesis-based CMI bounds, and while techniques from [1–4] may be extendable to Theorem 1, the introduction should clarify its scope to avoid overstating generality.

[1] Hafez-Kolahi, Hassan, et al. "Conditioning and processing: Techniques to improve information-theoretic generalization bounds."NeurIPS 2020.

[2] Harutyunyan, Hrayr, et al. "Information-theoretic generalization bounds for black-box learning algorithms." NeurIPS 2021.

[3] Hellström, Fredrik, and Giuseppe Durisi. "A new family of generalization bounds using samplewise evaluated CMI." NeurIPS 2022.

[4] Wang, Ziqiao, and Yongyi Mao. "Tighter information-theoretic generalization bounds from supersamples." ICML 2023.

In addition, below are some additional relevant references:

The use of (input-output) mutual information to evaluate memorization has been studied from a different angle in:

[5] Brown, Gavin, et al. "When is memorization of irrelevant training data necessary for high-accuracy learning?." STOC 2021.

The idea of applying model compression to obtain improved information-theoretic generalization bounds has also been explored in:

[6] Bu, Yuheng, et al. "Information-theoretic understanding of population risk improvement with model compression." AAAI 2020.

Tighter variants of CMI bound in the context of some CLB problems are also studied in:

[7] Wang, Ziqiao, and Yongyi Mao. "Sample-conditioned hypothesis stability sharpens information-theoretic generalization bounds." NeurIPS 2023.

---

> ### Author Rebuttal · Authors · 2025-07-30
>
> We thank the reviewer for the positive rating and feedback. We are glad that they enjoyed reading the paper and answer their questions below.
>
> **Question 1.** This is a good and valid point: one could indeed question whether the fact that our approach yields bounds with the desired behavior truly implies that the limitations are not inherent to the CMI framework, especially given the technical work required to make CMI successful, as the reviewer noted.
>
> Nonetheless, we would like to maintain this claim for two main reasons:
> * First, while the machinary of suitable stochastic projection and lossy compression is indeed essential (notably, the way we control simultaneously the projection dimension and incurred distortion in the generalization error of the projected-quantized surrogate model), our final bounds are still within the CMI framework (not from another one).
> * Second, although seemingly so, the core analysis is not excessively more complex. In particular, when controlling the distortion $\epsilon$ induced by the compression, we do not need to account for the statistical dependencies between the original model $W$ and the data $S$, which simplifies the bounding steps. We elaborate on this point below.
>
> The main difficulty in bounding the generalization error of stochastic learning algorithms is essentially due to the dependence of $W$ on $S$. Otherwise, bounding the generalization error would reduce to a classical concentration problem, e.g., for a fixed model. A common approach to overcome this difficulty is the change-of-measure trick, which comes at the expense of, e.g., a KL-divergence term. MI and CMI bounds fall into this category of approaches.
>
> Our general framework consists of two main steps: i) we apply projection-quantization to the original model and study the induced distortion, then ii) we study the generalization error of the projected-quantized model.
>
> *Step i:* We consider the expected difference of the generalization errors for $W$ and $\Theta \hat{W}$. Since we construct $\hat{W}$ suitably from $\Theta^\top W$, **we do not need to account for the dependence between $W$ and $S$** at this stage. Instead, the analysis involves developing expected concentration bounds by taking expectations over three sources of randomness: **the projection matrix** ($\Theta$), **quantization noise** ($V_{\nu}$, see l. 1076), and the **empirical deviation of the distribution of $S$ from $\mu$**. As shown in the proofs, the analysis of the distortion term often **reduces to classical concentration problems with well-known (near)-tight bounds**.
>
> *Step ii:* We then consider the joint distribution of $(S,\Theta \hat{W})$, and leverage CMI to establish generalization bounds. Projection-quantization reduces the dependence between the model and training data.
>
> In short, our approach decomposes the generalization error into two components, where the first can often be bounded near-optimally using classical concentration techniques.
>
> That said, we agree with the reviewer that the statement that our claim "the aforementioned limitations are not inherent to the CMI framework" may be questioned by readers, so we will add this discussion to clarify this point accordingly.
>
>
> **Question 2.** Theorem 4 is primarily given to show that there are no counterexamples to our extended CMI bound of Theorem 1, not even within the broader class of generalized linear stochastic optimization. Regarding the order of the bound in Theorem 4, we acknowledge that it is likely suboptimal and conjecture that a refined analysis could achieve a rate of $\sqrt{\log(n)/n}$. The intuition is as follows.
>
> As stated in our response to Question 1, analyzing the distortion requires bounding some expectations over the randomness on the projection-quantization and the dataset. However, the current distortion analysis in the proof of Theorem 4 only averages over the projection-quantization randomness and considers the worst-case scenario among all data samples. We think that a more refined analysis, which incorporates a concentration bound related to the deviation of empirical distribution of $S$ from $\mu$, could lead to a better rate. We will include this discussion in the paper.
>
> **Question 3.** This is actually a typo: $W$ should read $\Theta \hat{W}$. Thank you for pointing it out.
>
> **On the extension to other variants of CMI:** Thank you for these suggestions and for pointing us these additional references. We agree that the wording in our introduction lacks precision, and we understand it may overstate the scope of our results. As you correctly note, our intention was to emphasize that Theorem 1 improves upon standard CMI bounds. We did not mean to imply that it dominates all the proposed variants, especially since, to the best of our knowledge, those approaches have not been explicitly shown to fail on the counterexamples of Attias et al. (2024) and Livni (2023).
>
> We will thus revise the introduction accordingly. More precisely, we will clearly state that our bound improves on standard hypothesis-based CMI bounds in a provable sense; acknowledge that other CMI variants, such as those you referred in [1–4] are not directly addressed in our paper.
>
> **Limitations:** Indeed, the CMI framework applies to parameterized models. We will add this potential limitation in the checklist.

---

> > ### Comment · Reviewer_9s6T · 2025-08-05
> > **Thank you for the reply**
> >
> > I appreciate the authors’ detailed response and, at this point, intend to maintain my positive evaluation.

---

### Official Review · Reviewer_KsyD · 2025-06-30

**Clarity:** 3
**Significance:** 3
**Originality:** 4
**Rating:** 5
**Confidence:** 3

**Summary:**

This paper presents a new type of conditional mutual information-based (CMI-based) generalization error bound based on passing a learned, randomized (even conditional on the data) hypothesis $W \\in \\mathbb{R}^D$ through a low-dimensional projection (via a linear mapping $\Theta^T$ from $\\mathbb{R}^D$ to $\\mathbb{R}^d$, passing the result through a possibly randomized procedure $\\tilde{\\mathcal{A}}$ which again yields a parameter in $\\mathbb{R}^d$, and then projecting back to $\\mathbb{R}^D$ via the linear mapping $\\Theta$. The authors refer to their procedure as "quantization". The first main result shows a generalization error bound for a quantized hypothesis which has on its RHS the disintegrated CMI of the parameter produced (prior to projecting back to $\\mathbb{R}^D$) with a sequence of binary indices (indicating which elements from a double sample of size $2n$ were included in the training set of size $n$), all conditional on the realization of the double sample and the realization of $\\Theta$ (which itself can be random). Importantly, the quantized hypothesis has the property that it upper bounds the generalization error of the original hypothesis up to an additive $\\varepsilon$, allowing us (up to $\\varepsilon$) extend the generalization error bound for the quantized hypothesis to a generalization error bound for the original hypothesis.

Several counter-examples from recent prior works have shown apparently strong limitations of CMI-based generalization error bounds, essentially showing that in high-dimensional settings ($D$ of order $\\Omega(n^4)$ or larger) that are potentially of interest, previous CMI-based generalization bounds become useless. For these settings, which include convex Lipschitz bounded stochastic convex optimization (SCO) problems and strongly convex Lipschitz SCO problems, the authors show that their new bounds provide meaningful generalization error guarantees. The authors also give results for a rather general, potentially non-convex setting that they call "generalized linear stochastic optimization problems". Finally, they give results related to memorization, showing that discretization can fight memorization.

**Questions:**

1. In lines 234–235, I wish for more explanation for the claim that the authors dimension reduction technique being lossy implies that they additionally need lossy compression of the algorithm to get the desired result. Can you explain more here? (just a few sentences should be fine; don't go overdo it!)

2. Does my interpretation of the results of Section 5 (Memorization) match up with your own, that is, (see $\\textcolor{blue}{\\bigstar}$ from Strengths and Weaknesses), memorization is not necessary for generalization?

3. On Line 275: "unlike the CMI bound of [11]" should be "unlike the MI bound if [11]"?

4. In Section 5, in Theorem 7, in point **ii)**, I was unsure of what algorithm (original vs discretized) the authors had in mind when they wrote "an adversary which $(m, q, \\xi)$-memorizes the data". Which is it, original or discretized?

**Ethical Concerns:**

["NO or VERY MINOR ethics concerns only"]

**Final Justification:**

I am satisfied with the authors' responses to me. Initially, I had some concerns due to points raised by Reviewer RmaZ regarding order of quantifiers, etc. I can see that this reviewer is satisfied with the authors' response and is in fact more positive on this work now. That is good enough for me (I initially did not have these concerns); I appreciate the nuance here (sorry for the vagueness of this statement).  This is a solid paper which I would like to see accepted.

**Limitations:**

I think it's fine. This work is mostly about circumventing limitations, so it would be a bit much to heavily emphasize limitations of the authors' circumventing of limitations.

**Quality:**

4

**Strengths And Weaknesses:**

## Pros

Overall, the paper is written very clearly until just before Section 5 (Memorization). Given all of the complexity of what the authors are doing (with many variables, stochastic kernels, etc.), the authors did a great job in making the mathematical presentation understandable. Also, the English descriptions to give interpretation to the various results helped a lot.

I really like the idea of this work. It is quite natural to consider using some type of lossy, noisy model compression in order to get much-improved generalization error bounds, but this of course can only be useful if the compression is able to faithfully describe the original model of interest. In this case, the authors measure this "faithfulness" according to difference in generalization error. They succeeded in using a standard idea of linearly mapping to a low-dimensional space, along with some additional ingredients, to get a CMI-based generalization error bound that applies to the original hypothesis (not just the compressed model). This is nice. Also, the authors noticed that some previous negative results for CMI-based generalization error bounds seem to heavily rely on having dimension of order $n^4$ or larger, which suggests trying to "take care of the dimension" via a low-dimensional mapping.

At first, one might say that the impact of this work is narrow due to its focus (in Section 4) of circumventing limitations of CMI bounds, limitations that were shown for high-dimensional settings. However, these limitations can be viewed as a sign of a problem with the bounds as used thus far, and the authors "knocking out" these limitations is a sign of a better path, specifically, a better way to witness the power of CMI bounds.

To be honest, I did read all of Section 4, and it was clear to me. The discussion from the authors was good. I don't really have any major comments here. One quibble (which I keep here rather than under "Cons" since it's minor) is that in lines 234–235, I wish for more explanation for the claim that the authors dimension reduction technique being lossy implies that they additionally need lossy compression of the algorithm to get the desired result. I'll repeat this as a question in the Questions section.

I appreciate the authors mentioning technical challenges in the main text and hinting at proof techniques. I'm guessing this paper would have worked better as a COLT paper, where the authors could take the space (in the main text) to explain more about how they specifically addresses technical issues (like getting Theorem 4, which sounds very interesting, but due to time I could not look at the appendix to see how the authors pulled off this result).

The results on memorization seem interesting. I did not have as good of a grasp (but still a reasonably ok grasp) on this section and felt the paper already was strong enough without this section, so I place somewhat less weight on this section in my review. Regarding the results on memorization, if I understand correctly, we can view Theorems 6 and 7 as being a critique on using memorization as a perspective on generalization. This is because an original algorithm might memorize (in some meaningful way, i.e, $m = \Omega(n)$ and suitably small $\xi$ and suitably large $q$), whereas a discretized algorithm can fail to memorize; the latter then can be shown to have good generalization properties, and those properties transfer to the original algorithm. Hence,  $\\textcolor{blue}{\\bigstar}$ memorization is not necessary for generalization.


## Cons

I have to say that Section 5, Memorization, is very interesting. However, packing it all into 2 pages, including the sophisticated Recall Game, giving an appreciation for what a suitable benchmark performance level should be, etc. just feels like too much. As a result, this section suffers. For example, in the paragraph immediately after Lemma 1, there is a confusing sentence regarding "every one of the following requirements taking separately". I interpret the authors as saying that for any one of the requirements, there exists an adversary that satisfies the requirement (given the authors' emphasis on "separately"), but I am still not sure and this could have been presented better. Assuming my interpretation is correct, then point **i)** is trivial since an adversary that always outputs $0$ is $0$-sound, and point **ii)** is trivial since an adversary that always outputs $1$ will recall $m = n$ samples with probability $q = 1$. And then points **iii)** looks the same as condition **i)** from the statement of Lemma 1. I guess the only thing worth spending space on is clarifying the interpretation of **iv)**.

Also in Section 5, in Theorem 7, in point **ii)**, I was unsure of what algorithm (original vs discretized) the authors had in mind when they wrote "an adversary which $(m, q, \\xi)$-memorizes the data".



## Minor comments

On line 96, $j = 1, 2$ should be changed to $j = 0, 1$

In Theorem 1, it may be helpful to explicitly clarify that $W \\mid \\mathbf{\\tilde{S}}$ means that $W$ is trained on the entire double sample $\\mathbf{\\tilde{S}}$ as this is the only time in the paper (as far as I can tell) that you do this. Other places all train on $n$-samples rather than a $2n$-sample.

In the statement of Theorem 1, first math display, the notation $\\mathcal{A}^{(d)}$ was never defined. I infer that it is the algorithm's hypothesis obtained via "quantization" (stochastic projection and lossy compression).

Line 275: "unlike the CMI bound of [11]" should be "unlike the MI bound if [11]"? (I didn't think [11] considers CMI bounds as those were unveiled in a concurrent work by different authors)

---

> ### Author Rebuttal · Authors · 2025-07-30
>
> Thank you for your careful reading and positive rating. We are grateful for your detailed comments in the 'Pros' section, and your appreciation of our results and discussion.
>
> > "Section 5, Memorization, is very interesting. However, packing it all into 2 pages ... just feels like too much."
>
> We acknowledge that Section 5 is dense, and we have revised it to improve readability. Based on your comment, we made the discussion after Lemma 1 more concise and to the point (see details below). Additionally, we have shortened Section 4.4 and move part of its content to the appendix. This allows us to add more background on the recall game. We hope these changes improve clarity, and we would be happy to receive any further suggestions you may have on this section.
>
> > "after Lemma 1, there is a confusing sentence"
>
> The reviewer's interpretation is correct, and we agree that sentence could be much clearer. We also agree that the first three cases are trivial. These were included for two reasons: to provide simple examples of the recall game that help readers build intuition; to emphasize that considering the soundness or recall criterion alone is not sufficient. For instance, it is possible to achieve either $\xi = 0$ or $q = 1$, or both small $\xi$ and large $q$ when $m=o(n)$. Therefore, an adversary is considered good if it achieves a small $\xi$ or large $q$ simultaneously, with a non-negligible value of $m$.
>
> Regarding  (iii), stating $\xi \geq q$ is "achievable" is highlighted because Theorems 6 and 7 show that when $m$ is not $o(n)$ and $\mathsf{CMI}(\mu,{\mathcal{A}_n}) = o(n)$, no adversary can outperform a "dummy" adversary.
>
> Based on your feedback, we will thus rephrase lines 333-339 as follows: "In particular, Lemma 1 shows that even a dummy adversary can  $(m,q,\xi)$-trace the data in several cases: when $\xi$ is small, when $q$ is large, when $\xi \geq q$, or when $\xi$ is small $q$ is large, provided that $m=o(n)$ and there exists a subset $\mathcal{B}\subseteq \mathcal{W}$ such that $\mathbb{P}\left(W \in \mathcal{B}\right)=1-\xi$. Hence, $(m,q,\xi)$-memorization does not always translate ... "
>
> **Minor comments:** Thank you for your thorough comments: we have corrected the typos in l.96 and 275, and we provide further clarifications on your other points below.
>
> > "In Theorem 1, it may be helpful to explicitly clarify that $W | \tilde{\mathbf{S}}$ means that $W$ is trained on the entire double sample $\tilde{\mathbf{S}}$"
>
> This notation actually refers to a random variable drawn from the conditional distribution $P_{W | \tilde{\mathbf{S}}} = \mathbb{E}_{P_{\mathbf{J}}}[P_{W |\tilde{\mathbf{S}}_\mathbf{J}}]$ (as in l.1055). Therefore, $W$ is in fact trained on $n$ samples (not $2n$). We thank you for pointing out this ambiguity, and will clarify it in the statement of Theorem 1.
>
> > "In the statement of Theorem 1, first math display, the notation $\mathcal{A}^{(d)}$ was never defined."
>
> While your interpretation of $\mathcal{A}^{(d)}$ is correct (see the definition in Section B, l.941), it should not appear in Theorem 1: the correct notation here is $\mathcal{A}$. We have now fixed it. We emphasize that this typo does not affect our results and conclusions.
>
> **Questions:**
>
> 1. Thank you for pointing this out, we realize that our writing was ambiguous. There are two sources of distortion in our framework:
> (i) the (stochastic) projection from a high- to low-dimensional space, which is inherently lossy,
> (ii) a second, intentional lossy compression step applied *after* projection.
>
>    The reason for step (ii) is that, in order to control the distortion introduced by step (i) (projection), we follow a standard approach inspired by lossy source coding. This helps us obtain quantized representations that are easier to analyze and leads to tighter bounds. This procedure is analogous to $\epsilon$-net covering.
>
>    We will revise l.234-235 as follows: *"Since this dimension reduction technique is lossy, controlling the induced distortion is essential. To do so, we introduce an additional lossy compression step by adding independent noise in the lower-dimensional space. This approach is inspired by lossy source coding and allows us to obtain tighter bounds on the quantized, projected model."*
>
> 2. Your interpretation aligns with ours. Our results show that an adversary's ability to memorize or trace training data from the learning model does not necessarily imply poor generalization: one can construct a quantized, projected representation of the model with similar generalization error that does not memorize the data.
>
> 3. Indeed, we have fixed the typo, thank you!
>
> 4. Thank you for your careful reading: we meant the discretized model, so $W$ should be replaced by $\Theta \hat{W}$ in Theorem 7(ii). We have corrected this typo.

---

> > ### Comment · Reviewer_KsyD · 2025-08-02
> > **thanks**
> >
> > Thanks for the clarifications. There is no way I could have guessed what the intended definition of $W \\mid \\mathbf{\\tilde{S}}$ was. Thus far, my evaluation of this work is the same as before. However, I see some important points were raised by some reviews. In particular, I will closely follow the discussion with Reviewer RmaZ.

---

> > > ### Author Response · Authors · 2025-08-04
> > >
> > > Thank you for engaging in the discussion during the rebuttal, which we have appreciated; and also for the positive feedback on our paper.

---

### Official Review · Reviewer_6JF3 · 2025-07-03

**Clarity:** 3
**Significance:** 2
**Originality:** 3
**Rating:** 5
**Confidence:** 3

**Summary:**

This paper proposes new generalization error bound for learning algorithm.
It is based on Conditional Mutual Information (CMI) that are made tighter by the joint use of stochastic projection (of model parameters) and the concept of lossy compression.
The bound is shown to be non-vacuous in specific problem instances for which MI and CMI bound were vacuous.
The paper also investigates the use of this new bound to discuss memorization (and, in appendix, bounds for SG(L)D).

**Questions:**

To what extent can the bounds be evaluated (e.g., plotted) and used to design new algorithms?

**Ethical Concerns:**

["NO or VERY MINOR ethics concerns only"]

**Final Justification:**

This paper proposes a novel way to derive tight generalization bounds.
Provided improvements to the exposition are made as discussed with the reviewers, it will be a very strong paper.

**Limitations:**

yes

**Quality:**

3

**Strengths And Weaknesses:**

The proposed compressed-projected-CMI approach is, to my knowledge, novel.
It provides non-vacuous bound for cases identified as pathological for previous MI-based bounding approaches, advancing the current knowledge on theoretical results.
The paper also reads well and the results seem sound.
Illustrations could have been helpful to help understanding the main concepts (MI, CMI, J, projection and Θ, lossy compression).

Remarks

- the paper might have been cut, there is no conclusion

Local comments

- l22: "intrinsic-dimension-" -> "intrinsic-dimension"
- l53: "_also_ used in [22], [45]", [22] is already mentioned just before
- l83: "A: Z -> W" -> "A: Z^n -> W"
- l96: "j=1,2" -> "j=0,1"

---

> ### Author Rebuttal · Authors · 2025-07-30
>
> Thank you for your feedback and positive evaluation: we appreciate you find our approach novel and sound. We have corrected the identified typos.
>
> > "Illustrations could have been helpful to help understanding the main concepts (MI, CMI, J, projection and Θ, lossy compression)."
>
> We agree that illustrations could help clarify these concepts, and observe that such visual aids are generally lacking in the existing literature. If the paper is accepted we will add a diagram that depicts pictorially our approach, i.e., stochastic projection onto a smaller dimension space followed by lossy compression that preserves the generalization; and, on top of all the CMI framework.
>
> > "there is no conclusion"
>
> We will add the following paragraph as a conclusion section at the end of our paper.
>
> *"In this work, we revisit recent limitations identified in conditional mutual information-based generalization bounds. By incorporating stochastic projections and lossy compression mechanisms into the CMI framework, we derive bounds that remain informative in stochastic convex optimization, thereby offering a new perspective on the results in [42,43]. Our approach also provides a constructive resolution to the memorization phenomenon described in [43], by showing that for any algorithm and data distribution, one can construct an alternative model that does not trace training data while achieving comparable generalization.*
>
> ***Limitations and open questions.** Like prior work on information-theoretic bounds, our analysis applies to stochastic convex optimization. A natural, open question is whether and how these results can be extended to more general learning settings. Another key direction is to translate our theoretical findings into actionable design principles for learning algorithms with controlled generalization and compressibility."*
>
> > "To what extent can the bounds be evaluated (e.g., plotted) and used to design new algorithms?"
>
> Please note that our CMI-based generalization bounds are primarily intended as diagnostic tools. A direct major utility of them is to uncover which properties of learning algorithms lead to generalization. But they can of course also be used in the construction of suitable regularizes for new algorithms that are generalization-aware (similar to in [45]). On this latter aspect, we mention that some estimation strategies have been proposed in prior art that can be applied within our framework. For example, the CMI term can be upper-bounded by the KL divergence between the conditional distribution of the parameters given the data and a prior [38] or, for countable hypothesis spaces, by a deterministic (but potentially loose) quantity [§4.1, 10] (see our Section D.2).

---

> > ### Comment · Reviewer_6JF3 · 2025-08-05
> >
> > Thank you for the answers and discussions, including with other reviewers and particularly Reviewer RmaZ. I will keep my score for now but keep the possibility to reconsider in case Reviewer RmaZ would find the answers unsatisfying.

---

### Official Review · Reviewer_RmaZ · 2025-07-03

**Clarity:** 3
**Significance:** 4
**Originality:** 4
**Rating:** 6
**Confidence:** 4

**Summary:**

This paper studies generalization bounds in stochastic convex optimization (SCO) using information-theoretic measures. While it is known that existing quantities like mutual information (MI) and conditional mutual information (CMI) do not guarantee optimal generalization, this paper proposes a new variant of CMI that achieves the optimal convergence rate. The key idea is to perform dimension reduction via a stochastic projection, such that the low-dimensional compressed model retains similar generalization guarantees to the high-dimensional model.

As an application, the paper examines the problem of memorization, defined in terms of membership inference or tracing attacks. Recent work (Attias et al., 2024) has shown that for any accurate, sample-efficient, overparameterized model in SCO, there exist distributions where an adversary can trace a constant fraction of the training set. The new bound presented in this paper appears to mitigate, to some extent, the memorization effect.

While this result does not contradict existing impossibility results (and the paper does not claim otherwise), it would be helpful if the authors clarified how their results should be interpreted in light of the related literature. Concrete discussion and comparison would improve the paper’s positioning and help avoid potential misunderstandings.

If these points are clarified and properly discussed, I would reconsider revising my score.

**Questions:**

Questions for the Authors:

What is the exact formal claim made in the paper? In particular, what are the precise order of quantifiers?

How should we reconcile your results with Attias et al. (2024), which claim:
For any learner, there exists a distribution such that there exists an attacker that can trace samples?

Does the success of your approach rely on randomness in the algorithm that is unknown to the adversary?

Wouldn't it be more accurate to say that the adversary traces the data (rather than "memorizes")? The learner memorizes, the adversary performs a tracing attack. Clarifying this terminology may help.

How is your paper related to "In Defense of Uniform Convergence: Generalization via Derandomization with an Application to Interpolating Predictors" (ICML 2020)?
The proposal in your paper reminds me of the idea of understanding generalization through a surrogate algorithm. In your case, the surrogate operates on the projected data. One could also consider a surrogate that works conditional on the projection, which would align more directly with the surrogate-via-conditioning approach advocated in that work.

Minor: The results at Attias et al. hold true for $d>n^2\log(n)$ and not $n^4$, but I assume it doesn’t make a difference.

**Ethical Concerns:**

["NO or VERY MINOR ethics concerns only"]

**Final Justification:**

POST REBUTTAL:

I believe the ideas in this paper are fundamental and could inspire future work on the role of compression in mitigating privacy issues, particularly in the overparameterized regime. The paper is also clearly written, though the comparison to prior work should be improved—it remains somewhat elusive in the broader context.
Therefore, I will raise my score to 6 (strong accept).

I encourage the authors to further clarify the presentation, especially regarding how and why the results do not contradict previous findings, as discussed in detail in the response. Clarifying this point would make the contribution more visible and valuable for future research.
Please also update the text to reflect that $D$ can grow as $n^2\log(n)$, rather than $n^4\log(n)$, since this strengthens the result. It would also be helpful to mention the exact rate at which $D$, should grow to enable memorization in the non-compressible model.

As additional suggestions, you could expand on the implications for differential privacy and sample compression schemes.
Another potentially interesting direction might be to frame your “random” version of the theorem in a game-theoretic language. In this view, the learner could be seen as the first player in a game, selecting a randomized strategy (i.e., the projection matrix) to share with the second player (the adversary)—without revealing the actual realization. Even under this partial information, the learner can still achieve their goal, highlighting the strength of randomization over determinism.

**Limitations:**

See the above.

**Quality:**

4

**Strengths And Weaknesses:**

Strengths:

The use of random projection to compress the model  based on training data drawn from a distribution is interesting. It shows that under these conditions, information-theoretic methods can achieve optimal convergence rates and mitigate memorization. Overall, the paper is well written, aside from a crucial point of interpretation discussed below.

Weaknesses:

Positioning the result:

In Attias et al. (2024), the order of quantifiers is: for any accurate, sample-efficient, overparameterized model in SCO, there exist distributions and adversaries such that a fraction of the training data can be traced (i.e., the model memorizes that data).

In contrast, the present paper claims that for any algorithm A that memorizes, there exists another algorithm A' (the compressed model) that does not memorize and achieves comparable generalization error. However, this appears to hold only for the specific distribution used to construct A', and does not rule out the possibility that a different distribution D' could be adversarially constructed for A'.

In other words, Attias et al. make a universal claim over learning algorithms, while this paper chooses a favorable algorithm tailored to a given distribution (using information from the distribution to perform compression). This effectively reverses the order of quantifiers, which avoids contradiction but results in a fundamentally different (and weaker) claim. This distinction should be discussed explicitly.

Another potential explanation, which I suspect is correct, is that the construction of the compressed model involves internal randomness that is unknown to the adversary. If so, the result may hold for randomized algorithms in a way that is compatible with prior impossibility results. Again, this is a subtle but essential point that should be addressed in the paper.

This issue is especially relevant given the abstract's phrasing:

"We show that for every learning algorithm and every data distribution, there exists an algorithm that does not memorize and which yields comparable generalization error."

While this statement is technically accurate, the difference in the game being played compared to prior work is non-obvious and potentially misleading. The main text (e.g., line 352) does soften the claim slightly:

"... allow to (partly) address this memorization issue, in a sense that will become clearer in the sequel."

This is more precise, but the relationship to prior results is still unclear. A detailed discussion of the interpretation of quantifiers (and perhaps the role of algorithmic randomness) would greatly strengthen the paper.

A similar concern arises in Section 4, which states that the proposed approach "resolves recently raised limitations of classic CMI bounds." While the method does provide a resolution in a certain sense, the conditions under which the bounds are derived are different. The phrase “stochastic projection along with our lossy compression resolves those issues…” seems insufficient. The dependence on the data distribution appears critical and should be made explicit.

Additional comment:
The current construction is existential: it shows that a compressed model with good generalization exists, but it does not provide an explicit or practical method to construct such a model.

---

> ### Author Rebuttal · Authors · 2025-07-30
>
> We thank the reviewer for the positive feedback and relevant suggestions. As requested, we clarify our contributions and how we think should be positioned with respect to prior work.
>
> For convenience, this is a summary of our formal claim and how it reconcilies with the impossibility results of Attias et al. (2024). Details will follow.
>
> Our proof of Theorem 7 actually allows two forms for the statement of the result: one that is explicit (the current one), and one that is implicit and can easily be implied by the very same proof. The two statements differ among them through the degree of algorithm randomness that is shared with the adversary and the order of quantifiers.
> * The current statement of Theorem 7 holds for $\Theta$ being deterministic and shared with the adversary. It asserts that for any learning algorithm $\mathcal{A}(S)=W$ and data distribution $\mu$, there exists a suitable projected-quantized model $\hat{\mathcal{A}}(S,\Theta)=\Theta \hat{W}$ for which no adversary would be able to trace the data. As correctly observed by the Reviewer, this statement indeed does not preclude the existence of data distributions and adversaries which would be able to trace that data from the output of the new algorithm $\hat{\mathcal{A}}(S,\Theta)=\Theta \hat{W}$. Note, however, that the impossibility result of Attias et al. does *not* apply to the new algorithm $\hat{\mathcal{A}}(S,\Theta)$ (see below); and, so, while our result of Theorem 7 in its current form does not rule out the existence of such distributions and adversaries for the new algorithm $\hat{\mathcal{A}}(S,\Theta)$, that of Attias et al. does not guarantee that existence either.
> * Easily implied by the very same proof, a second form of Theorem 7 holds for $\Theta$ being random and shared with the adversary (not deterministic as for the current statement!); and its statement is exactly that of the current one but with an added extra expectation over the projection matrix $\Theta$ in the LHS of the inequality (6) -- See the section **Randomized projections and quantization** below).   This alternate form of statement of Theorem 7 asserts that if for a given $W$ a random $\Theta$ is chosen and **revealed to the adversary**, then the projected-quantized model which uses that $\Theta$ guarantees that: (i) for **any data distribution**, **no adversary** can trace the data; and (ii) on average over $\Theta$, the corresponding generalization error is comparable to that of the original model $W$. This means that the order of quantifiers here is the same as for the impossibility result of Attias et al. (2024).
>
> We hasten to mention that  none of these two forms of statement contradicts the impossibility result of Attias et al. (2024). This is because the result of Attias et al. does *not* apply to our projected-quantized algorithm $\hat{\mathcal{A}}(S,\Theta)$. In particular, athough equally universal the seemingly contracting above alternate statement of Theorem 7 does actually *not* contradict the impossiblity result of Attias et al. (2024). More precisely, their main assumption on the boundednes of model's norm (used in the proof of their Theorem 4.1, e.g., right before equation (12) in the PMLR version of their paper) would require that $\mathbb{E}\left[\|\Theta \Theta^\top W\|^2\right]=\frac{D+d+1}{d}\|W\|^2 \leq 1$, which cannot be statisfied with a dimension $D$ that grows with $n$ as $O(n^4 \log n)$. The same argument holds for the current statement of Theorem 7.
>
> If the reviewer agrees with, we will keep the statement of Theorem 7 as is but add in the revised paper a discussion on the order of quantifiers and relation to Attias et al., as well as how a slightly modified form of the statement can be inferred from the very same proof but with a result that holds with the same universality as Attias et al. (2024) and, finally, how/why the result of Attias et al. (2024) applies to none of the two versions of statement.
>
> **On the formal claim, order of quantifiers, and disclosure of the randomness of the algorithm:** Thank you for taking the time to carefully read our paper and write thorough comments. Regarding the randomness, in addition to the standard randomness already present in the process of learning from samples, our method introduces two further sources of randomness: the stochastic projection matrix $\Theta$ and the quantization noise (the random variable $V_{\nu}$, see l.1076). Depending on whether these sources are fixed and shared with the decoder or not, Theorem 7 can be stated in slightly different ways, resulting in distinct levels of generality:
>
> - **Partially deterministic (deterministic projection and random quantization):** Theorem 7, as stated now, assumes a fixed projection matrix $\Theta$ that is known to the adversary, in addition to a random (quantization noise) $V_{\nu}$ which is unknown to the adversary. The randomness on $V_{\nu}$ is used merely to properly quantize the projected hypothesis space, as is common in rate-distortion theory literature. Alternatively, one could use a "fixed" quantization technique similar to $\epsilon$-net covering in order to map the projected models to the closest points on the $\epsilon$-net mesh. However, there would still exists some level of randomness that is unknown to the adversary (related to how such a mapping is performed as the adversary only observes the centroid points on the $\epsilon$-net). We note that even with common learning algorithms (i.e., in the absence of any stochastic projection and/or lossy compression, such as using standard mini-batch SGD), there are sources of randomness (related, e.g. to the choices of mini-batch size, initialization, and other choices) that are unknown to the adversary.
>
>   For this setting, we show that for any learning algorithm $\mathcal{A}(S)=W$ and data distribution $\mu$, there exists a suitable projected-quantized model $\hat{\mathcal{A}}(S,\Theta)=\Theta \hat{W}$ for which no adversary can trace the data. Such statement indeed does not preclude the existence of data distributions and adversaries which could trace some amount of the data by directly observing the output of the new algorithm $\hat{\mathcal{A}}(S,\Theta)=\Theta \hat{W}$. However, as we argue (see the argument in the above summary in the beginning of this response) the impossibility result of Attias et al. does *not* apply to this new algorithm $\hat{\mathcal{A}}(S,\Theta)$.
>
>
>
> - **Randomized projections and quantization:** In the actual proof of Theorem 7, we begin by establishing a version of the inequality (6) that holds in-expectation over $\Theta$. Namely, the same inequality (6) but with an extra expectation over $\Theta$ in the LHS of the inequality (and, actually, this is the exact argument that we then use to infer the existence of at (least) one value of $\Theta$ for which the inequality holds without the expectation).
>
> As a result, Theorem 7 also holds if $\Theta$ is randomly generated and revealed to the adversary (provided that one adds the mentioned extra expectation over $\Theta$ in the LHS of (6)).
> For this setting (i.e., for a given $W$, a random $\Theta$ chosen and **revealed to the adversary**) we get that the projected-quantized model which uses that $\Theta$ guarantees that: (i) for **any data distribution**, **no adversary** can trace the data; and (ii) on average over $\Theta$, the corresponding generalization error is comparable to that of the original model $W$.
>
>   This new statement makes the claim more general and somewhat similar to that of Attias et al. in its universality (which it does not contradict, as explained above), but at the expense of more randomness in designing the learning algorithm.
>
> We will add these discussions to avoid any confusion.
>
> **On the terminology usage "tracing vs. memorizing the data:"**  Indeed; it would be better to refer to the adversary as being able to "trace or not trace" the data instead of it memorizing or not that data. We will revise the document using this terminology, which is better.
>
> **On the relation to the prior art ICML 2020 paper "In Defense of Uniform Convergence: Generalization via Derandomization with an Application to Interpolating Predictors":** Thank you for pointing out to us the connection with this work, which we were not aware of. In that work, the generalization error of an algorithm is studied indirectly using a "surrogate algorithm" and by controlling the distortions induced for the population and empirical risks, in a way that is essentially similar to lossy compression in rate-distortion theory. Note that although there is indeed a high level connection, in our case: (i) the surrogate or lossy algorithm is achieved through suitable stochastic projection (onto a proper smaller dimension space) followed by quantization, not just lossy compression, (ii) the quantization is a very careful one, chosen in a manner that the  generalization error of the surrogate algorithm has comparable generalization error with the original algorithm, and (iii) we accomodate the CMI framework to the used stochastic projection and lossy compression. Nontheless it is indeed useful to the reader to reference this paper, which we will do; and discuss relationship to our work.
>
> **On the order of dimension:** In the latest version of the PMLR paper by Attias et al. (2024), the condition is written as $n^4\log(n)$. However, as the reviewer mentioned, this ordering does not affect our results.
>
> **Regarding the practical method for constructing the model:** The reviewer is right. However, it should be noted that similar high-probability bounds also hold for the used Johnson-Lindenstrauss projection. This means that if we take a random matrix $\Theta$ whose entries are chosen i.i.d. from the distribution $\mathcal{N}(0,1/d)$ then, with high probability, the new algorithm will have comprable generalization error to that of the original algorithm.

---

> > ### Author Response · Authors · 2025-08-05
> >
> > Dear Reviewer RmaZ,
> >
> > Again, thank you for your review and positive feedback on our paper, as well as the relevant comments and suggestions.
> >
> > We would appreciate if you could please take a look at our rebuttal response and let us know if that clarifies the points you raised.
> >
> > In advance, thank you for your great help with this.

---

> > > ### Comment · Reviewer_RmaZ · 2025-08-05
> > >
> > > Apologies for the late response.
> > > Thank you for the detailed reply - it clarified most of my concerns.
> > > I also find the second, "randomized" version of Theorem 7 particularly interesting.
> > > I think these clarifications really help convey the core contribution of the paper, especially since the precise distinction from Attias et al. (2024) is fascinating.
> > >
> > > There’s one point I’m still unsure about. Could you elaborate on the sentence:
> > > "More precisely, their main assumption on the boundedness of the model's norm..."
> > > This seems to be the key technical distinction, and I’d appreciate a clearer explanation to be confident about it.
> > >
> > > I will vote for acceptance of the paper.
> > >
> > > As a suggestion, it might be interesting to add a brief discussion on potential implications for differential privacy in stochastic convex optimization.

---

> > > > ### Author Response · Authors · 2025-08-06
> > > >
> > > > Thank you for your follow-up on the discussion; and your helpful and deep comments/suggestions during both the initial review and this rebuttal. We are glad to that our response has helped clarify the picture, especially in regard to the impossibility results by Attias et al.
> > > >
> > > > Your suggestion on the implications of our approach on differential privacy is indeed a good one; and we will elaborate on it and mention it in the form of an added Concluding Remarks section to the document. Our results may also have important consequences on the question of (impossibility of existence) of constant-size sample compression schemes in SCOs; that we will mention as well.
> > > >
> > > >
> > > > Hereafter, we elaborate on: (i) the role of the bounded model norm assumption $\\|W\\|^2 \leq 1$ that is central to the impossibility results of Attias et al. in their proofs (for so called Convex-Lipschitz-Bounded (CLB) instance class of SCOs); and, (ii) why/how violating that condition (which, as mentioned in our previous response, cannot be enforce for our projected-quantized model since $\mathbb E [\\|\Theta \Theta^T W\\|^2] = \frac{D+d+1}{d} \\|W\\|^2$ cannot be smaller than one when $D$ grows with $n$ as $\mathcal O(n^4 \log n)$) affects some of the key proof steps.
> > > >
> > > > All references (e.g., to equations, lemmas, theorems, and so on) are w.r.t. the PMLR version of Attias et al. (2024). Furthermore, for convenience we use the notation of our paper. (For example, the dimension of the initial model (before projection) is denoted by $D$ in our paper, while Attias et al. denote it by $d$ - This latter notation ($d$) is used in our paper for the (smaller) dimension after applying a suitable projection).
> > > >
> > > > Since what plays a crucial role for the bounded norm assumption being discussed is the stochastic projection, not the lossy compression, hereafter for simplicity we set $\epsilon=0$.
> > > >
> > > > In this case, in our approach we replace the model $W$ by $W' = \Theta \Theta^\top W$. Note that while $\mathbb{E}_{\Theta}[W']=W$, we have $\mathbb{E}\left[\\|W'\\|^2\right]=\frac{D+d+1}{d}\\|W\\|^2$. That is, the variance of $W'$ increases with the dimension $D$; and can be arbitrary large (e.g, when $D$ grows as $\mathcal{O}(n^4 \log n)$)! Intuitively, this is what prevents an adversary from guessing correctly whether a sample has (or not) been used for training; and which makes some key proof steps of Attias et al. fail when applied to the model $W'=\Theta \Theta^\top W$. Explanation is as follows.
> > > >
> > > > - Consider the last inequality of page 19 of Attias et al. [2024], whose result is used later on in the proof to bound the term $\mathbb{E}[|\mathcal{I}|]$ -- the set $\mathcal I$ being the subset of columns of supersample such that one of the samples has a large correlation with the
> > > > output of the algorithm and the other one has small correlation with the output of the algorithm. When working with the model $W'$ the RHS of this inequality needs to be replaced by the $D$-dependent quantity $8\varepsilon^4 n^2 \frac{D+1}{d} +2\varepsilon^2$, which increases with $D$, e.g., as $D=\Omega(n^4)$ and $d=o(n)$. This has to be contrasted with the actual bound $8\varepsilon^4 n^2 +2\varepsilon^2$ when the bounded model's norm assumption holds. Thus, one important issue is that, this quantity now being now non-negligible, the LHS of (9) can no longer be lower-bounded by the RHS of the inequality (9).
> > > >
> > > > - Another proof step of Attias et al. that would fail when applied to the model $W'$ is in upper-bounding the probability that the correlation between the model output and the heldout samples is significant in Eq. (11). Such error event, denoted as $\mathcal{G}^c$ in Attias et al., is shown therein to have a probability that vanishes with $n$ as $\mathcal{O}(1/n^2)$. For the model $W'$, that bounding step does not work as it yields $\mathbb{P}(\mathcal{G}^c) \lessapprox \mathcal{O}\left(n e^{-\frac{\beta^2}{n^2}}\right)$. Clearly this bounding step is now vacuous!, which is a direct consequence of the variance of $W'$ being increasing with $D$; and, so, with $n$.
> > > >
> > > > - Another probability of error that does not vanish when considering the model $W'$ is that of the event $\mathcal{E}^c$. Precisely, in this case the denominator in the LHS of  (12) depends on $D$, for exactly the same reason as in the previous point; and, because of this one cannot get the desired $\mathcal{\Omega}(1/\epsilon^2)$.
> > > >
> > > > We hope this clarifies more the role of our introduced stochastic projection, especially in relationship with Attias et al. proof. If deemed needed/useful, please let us know; and we will be happy to elaborate more.

---

> > > > > ### Comment · Reviewer_RmaZ · 2025-08-07
> > > > >
> > > > > Thanks again for your detailed response.
> > > > >
> > > > > I believe the ideas in this paper are fundamental and could inspire future work on the role of compression in mitigating privacy issues, particularly in the overparameterized regime. The paper is also clearly written, though the comparison to prior work should be improved—it remains somewhat elusive in the broader context.
> > > > > Therefore, I will raise my score to 6 (strong accept).
> > > > >
> > > > > I encourage the authors to further clarify the presentation, especially regarding how and why the results do not contradict previous findings, as discussed in detail in the response. Clarifying this point would make the contribution more visible and valuable for future research.
> > > > > Please also update the text to reflect that $D$ can grow as $n^2\log(n)$, rather than $n^4\log(n)$, since this strengthens the result. It would also be helpful to mention the exact rate at which $D$, should grow to enable memorization in the non-compressible model.
> > > > >
> > > > > As additional suggestions, you could expand on the implications for differential privacy and sample compression schemes.
> > > > > Another potentially interesting direction might be to frame your “random” version of the theorem in a game-theoretic language. In this view, the learner could be seen as the first player in a game, selecting a randomized strategy (i.e., the projection matrix) to share with the second player (the adversary)—without revealing the actual realization. Even under this partial information, the learner can still achieve their goal, highlighting the strength of randomization over determinism.

---

> > > > > > ### Author Response · Authors · 2025-08-07
> > > > > >
> > > > > > Thank you so much! We were really glad to receive such a deep and high-standard quality review; and it was pleasure to continue the constructive discussion during the rebuttal.
> > > > > >
> > > > > > Definitely, we will improve the exposition taking into account all the aspects that we discussed, including relation to prior art, how/why our results do not contradict Attias et al. previous findings, choice of the dimension $D$, the implications for differential privacy, sample compression schemes and the "random" version of Theorem 7 in a game theoretic language, which indeed is very interesting.

---

### Author Response · Authors · 2025-08-08

Dear Reviewers,

As ourselves serving as reviewers for other papers we know how difficult it can be to find time to reserve for deep, honest and constructive review.

All your reviews have been as such; and we really appreciated the quality of your initial reviews, as well as your entire engagement and constructive comments during the rebuttal period; and, for all of that, we would like to sincerely thank you all!

Authors.

---

### Decision · Program_Chairs · 2025-09-17

**Decision:**

Accept (oral)

**Comment:**

This paper establishes new algorithm-dependent generalization bounds for stochastic convex optimization, based on a variant of conditional mutual information (CMI) involving stochastic projections.  The new bounds are applicable to natural scenarios where previous CMI bounds are known to be inapplicable.  They further use these bounds to explore issues concerning avoiding the need for "data memorization" for certain learning problems, contrasting with recent work on the subject.

It is now well-understood that techniques for algorithm-dependent analysis are necessary for understanding generalization in overparameterized regimes.  CMI is a natural framework for approaching such algorithm-dependent analysis.  However, previous attempts to use CMI to understand generalization yielded negative results in high-dimensional problems.  This new definition breaks through such limitations, showing this variant of CMI can establish generalization guarantees for stochastic convex optimization in such scenarios, thus demonstrating this as a promising technique for algorithm-dependent analysis.

Reviewers all agree that the results are novel and important, shedding light on topics currently of high interest to the community.

Reviewers praise the clarity of writing in the paper, with only a few spots as exceptions.

The discussion between authors and reviewers (RmaZ) was enlightening, concerning why the new results on avoiding data memorization do not contradict prior works showing necessity of data memorization, and the authors have indicated they will expand upon this explanation in the final version of the paper.